# Next-generation EDIpack: A Lanczos-based package for quantum impurity models featuring general broken-symmetry phases, flexible bath topologies and multi-platform interoperability

Lorenzo Crippa[1,2,3], Igor Krivenko[1], Samuele Giuli[4], Gabriele Bellomia[4], Alexander Kowalski[3], Francesco Petocchi[5], Alberto Scazzola[6], Markus Wallerberger[7], Giacomo Mazza[8], Luca de Medici[9], Giorgio Sangiovanni[2,3], Massimo Capone[4,10] and Adriano Amaricci[10]

**1** I. Institut für Theoretische Physik, University of Hamburg, Hamburg, Germany
**2** Würzburg-Dresden Cluster of Excellence ct.qmat, Dresden, Germany
**3** Institut für Theoretische Physik und Astrophysik, Universität Würzburg, Würzburg, Germany
**4** SISSA, Scuola Internazionale Superiore di Studi Avanzati, Trieste, Italy
**5** Department of Quantum Matter Physics, University of Geneva, Geneva, Switzerland
**6** Department of Electronics and Telecommunications, Politecnico di Torino, Torino, Italy
**7** Institute of Solid State Physics, TU Wien, Vienna, Austria
**8** Department of Physics "E. Fermi", University of Pisa, Pisa, Italy
**9** LPEM, ESPCI Paris, PSL Research University, CNRS, Sorbonne Université, Paris, France
**10** CNR-IOM, Istituto Officina dei Materiali, Consiglio Nazionale delle Ricerche, Trieste, Italy

EDIpack group: edipack@cnr.iom.it

## Abstract

We present a next-generation version of EDIpack, a flexible, high-performance numerical library using Lanczos-based exact diagonalization to solve generic quantum impurity problems, such as those introduced in Dynamical Mean-Field Theory to describe extended strongly correlated materials. This new release efficiently solves impurity problems allowing for different broken-symmetry solutions, including superconductivity, featuring local spin-orbit coupling and/or electron-phonon coupling. It provides quick access to dynamical correlation functions on the entire complex frequency plane at zero and low-temperatures. The modular architecture of the software not only provides Fortran APIs but also includes bindings to C/C++, interfaces with Python and Julia or with TRIQS and w2dynamics research platforms, thus ensuring unprecedented level of inter-operability. The outlook includes further extensions to study quantum materials and cold atoms quantum simulators, as well as quantum information applications.

# 1    Introduction and Motivation

Quantum impurity models play a central role in the study of strongly correlated electron systems. They describe the coupling of a small number of localized degrees of freedom to an extended non-interacting environment [1, 2]. Originally introduced to describe diluted magnetic impurities in metallic hosts, single-impurity models have played a central role in the field of quantum many-body systems through the decades, starting from the full understanding of the Kondo problem [3–5] and subsequently proving crucial to the understanding of a wide range of quantum many-body phenomena [6–9]. More recently, their scope has expanded beyond the realm of condensed matter, influencing the understanding of other areas such as quantum information [10–15] or ultra-cold atomic systems [16–20].

At the heart of quantum impurity models is the interaction between a localized "impurity" and a surrounding bath of itinerant particles, usually represented as a conduction band of electrons [2]. In this context, the impurity can be defined more generally, so as to include orbital and spin local degrees of freedom as well as a finite number of lattice sites, i.e. quantum clusters, immersed in a suitable environment. This setup lies at the core of several quantum embedding methods, including Density-Matrix Embedding Theory (DMET) [21, 22] and further extensions [23, 24], the Gutzwiller Approximation (GA) [25, 26], Quantum Cluster methods [27–30] or Dynamical Mean-Field Theory (DMFT) [7–9], capturing many essential aspects of strong correlation, including screening [31–34], decoherence and entanglement in a computationally tractable way [13, 15].

In several of these embedding schemes, most notably DMFT, the complex physics of the lattice systems gets reduced to an effective self-consistent quantum impurity problem. By accounting for local quantum fluctuations, DMFT successfully describes the most important physical features of correlated materials [7, 9], including Mott insulators [35–45], heavy fermion compounds [46–52] and unconventional superconductors [13, 43, 53–61].

Accurately and efficiently solving a quantum impurity problem and computing the Green's functions, which are central in the DMFT framework, nonetheless remains a critical challenge, particularly for multi-orbital systems, those at low temperatures or endowed with lower symmetry. To this end, many advanced numerical techniques have been developed over the years, incorporating cutting-edge strategies to reduce computational loads and leverage state-of-the-art algorithms [62, 63]. These range from continuous-time quantum Monte Carlo methods [58, 64–67], to numerical renormalization group approaches [42, 68–70] and density-matrix renormalization group [42, 68, 69, 71] or Configuration Interaction [72–76] solvers. In this context, the Exact Diagonalization (ED) approach plays a significant and

distinctive role [54, 77–82].

Several open-source ED software packages have recently been made available within the condensed matter community to address quantum many-body problems with an emphasis on efficiency and flexibility. Examples include `EDLib` [83] and the newer `XDiag` [84], which focus on generic fermionic systems; `Pomerol` [85] designed for fermion-boson models and interfaced with TRIQS [63]; HΦ [86, 87] for fermions coupled to spins; and general-purpose spin-model solvers such as `QuSpin` [88,89], all contributing to the rich landscape of ED-based techniques.

In this work we present the improved and renewed EDIpack: a flexible, high-performance numerical library designed to address these challenges, providing an ED solver for generic quantum impurity problems. The library builds on the foundations of its predecessor [82], featuring massively parallel Lanczos-based algorithms [90–92] while including several significant extensions and improvements. The library now supports, within a unified framework, both zero and finite temperature calculations [79, 82] for a wide set of impurity models, making it suitable for addressing a broad class of problems, from spin-orbit coupling in quantum materials to multi-orbital superconductivity. It also provides direct access to numerically exact dynamical correlation functions across the entire frequency complex plane, thus enabling calculations of spectral functions, local susceptibilities, and other response properties.

Additionally, EDIpack offers access to the Fock space of the impurity system and thus, for instance, the possibility to evaluate the impurity reduced density matrix, enabling quantum information-inspired analyses providing insights into the entanglement properties, subsystem purity, and quantum correlations of interacting electron systems [13, 15]. In the quantum embedding context, these features are increasingly recognized as essential to understand the emergent behavior of correlated matter, including quantum criticality and superconductivity.

This new EDIpack version also includes support for systems with inequivalent impurities. This is a critical feature in DMFT or, more generally, in quantum embedding methods, to study hetero-structures, super-lattices, and disordered systems, where the local environment varies significantly across different sites.

With the growing importance of quantum impurity models in the study of correlated materials, standardization and interoperability across software suites have become critical necessities. EDIpack has been expressly designed following this guiding principle. While the library itself is implemented using modern Fortran constructs, it also provides an extensive set of C-bindings that enable integration with other programming languages, be they compiled (C, C++), JIT-compiled (Julia) or interpreted (Python). Especially notable in this sense is the Python API, EDIpack2py, which beyond providing a high-level interface for defining and solving impurity problems, it also allows integration with established scientific libraries, such as NumPy and SciPy, facilitating data analysis and rapid prototyping.

A key development feature in EDIpack, unlocked by the introduction of the Python API, is the possibility to directly interface to both TRIQS [63] (Toolbox for Research on Interacting Quantum Systems) and w2dynamics [67] software suites, which are two of the most important and widely used computational frameworks in the field. EDIpack slots in as an alternative solver to the native Quantum Monte Carlo implementations [64, 66], extending the parameter (and especially temperature) range reachable by the simulation while minimizing the coding requirements for the end user. At the same time, this integration provides EDIpack users with access to the extensive data handling and analysis tools in the TRIQS ecosystem. Overall, interoperability enhances the collective reach and reliability of these platforms, improving the reproducibility and cross-validation of numerical results, with the ultimate goal of achieving standardization for quantum impurity solvers

along the guiding principles of initiatives such as FAIRmat (see [fairmat-nfdi.eu/fairmat](fairmat-nfdi.eu/fairmat)).

In this paper, we present the mathematical framework, details of the implementation, and key features of EDIpack, along with several illustrative applications. We aim to demonstrate how this flexible, high-performance solver can be used to tackle a wide range of challenging problems. The rest of this work is organized as follows. In Sec. 2 we briefly illustrate the overall structure of EDIpack together with its dependencies, configuration, and installation procedures. In Sec. 3 we present the quantum impurity problem and review the software implementation in detail. In Sec. 4 we thoroughly discuss the C-binding interface that is at the heart of the interoperability capabilities of EDIpack. In this section we also discuss 4 interface layers: i) the Python API, ii) the TRIQS interface, iii) the w2dynamics interface and iv) an experimental, yet operative, Julia API. In Sec. 5 we discuss the functionalities and the capabilities of EDIpack and all the discussed extensions with a series of illustrative examples. Finally, in Sec. 6 we present some concluding remarks and considerations.

## 2 Installation

The configuration and installation of EDIpack is handled by CMake, which ensures cross-platform compatibility and dependency resolution. The software builds into two distinct libraries. The main one is a static Fortran library named `libedipack.a`, which alongside the compiled modules, wraps the EDIpack software. A second dynamic library, `libedipack_cbindings.so`, together with the associated header file `edipack_cbindings.h`, enables interoperability with other programming languages.

### 2.1 Structure

EDIpack is a modular library organized into two main components. At its core lies the ED solver for the quantum single-impurity Hamiltonian, constituting the primary computational engine. Building on this, the library includes EDIpack2ineq, an extension for handling multiple inequivalent impurity problems at once. Finally, a Fortran-C/C++ interface is provided for seamless integration with external software and the development of additional APIs.

- **EDIpack.** This module forms the foundation of the library, implementing a Lanczos-based solver for general quantum impurity problems. It supports systems with a wide range of symmetries leveraging conservation of different quantum numbers, multi-orbital models, and even coupling to local phonons. The EDIpack solver is structured hierarchically and modularly, with different sections of the library communicating through a shared memory layer. The top-level module, `EDIPACK`, provides access to the core Fortran API, exposing key procedures for initialization, execution, and finalization, while abstracting the underlying data structures. A detailed overview of this part of the library is provided in Sec. 3.

  - ↪ *EDIpack2ineq.* This extension, developed using suitable Fortran interfaces, enables the treatment of multiple, independent quantum impurities. It is particularly useful in DMFT applications involving unit cells with inequivalent atomic sites or systems with broken translational symmetry, such as heterostructures, large supercells, or disordered materials. This module provides flexible memory management and supports the simultaneous solution of multiple impurity problems, as discussed in Sec. 3.14.

- **EDIpack C-bindings.** For enhanced interoperability, EDIpack includes a dedicated module implementing Fortran-C/C++ bindings for key library procedures. This module relies on the `ISO_C_BINDING` capabilities available in modern Fortran, allowing direct translation of Fortran functions to C. To ensure a straightforward user experience, only the functions and variables directly exposed to the user are included in this binding, shielding developers from the complexity of the library's internal architecture. This interface is intended to facilitate integration with third-party software and support the development of custom APIs.

## 2.2 Dependencies

EDIpack directly depends on two external libraries.

1. **SciFortran**: an open-source Fortran library that provides support for mathematical and scientific software development, available at [github.com/SciFortran](github.com/SciFortran).

2. **MPI**: a distributed memory parallel communication layer with support for modern Fortran compilers.

**SciFortran** provides a solid development platform enabling access to many algorithms and functions, including standard linear algebra operations and high-performance Lanczos-based algorithms. This greatly reduces code clutter and development time. The use of distributed memory parallel environment is required to access scalable parallel diagonalization algorithms, which speed up calculations for large systems. Nonetheless, thanks to careful design, EDIpack can operate serially even without initializing the MPI framework.

## 2.3 Build and Install

### 2.3.1 Compilation from source

The software can be installed from source as follows. The code package can be retrieved directly from its GitHub repository, for instance using

```
git clone https://github.com/edipack/EDIpack
```

Then, assuming to be in the software root directory (`$cd EDIpack`), a conventional out-of-source building can be performed as follows:

```
mkdir build
cd build
cmake ..
make -j
make install
```

The CMake configuration can be further tuned using the following variables:

| Option | Scope | Value (default in green) |
|--------|-------|--------------------------|
| -D**CMAKE_INSTALL_PREFIX** | Install prefix override | None/User-defined path |
| -D**LONG_PREFIX** | Install directory (if | ∼/opt/edipack/\ |
| | install prefix unset) | TAG/\ (*e.g. 5.3.3*) |
| | | PLAT/\ (*gnu/intel/...*) |
| | | BRANCH (*branch name*) |
| | | ∼/opt/edipack/custom |
| -D**USE_MPI** | MPI support | True/False |
| -D**WITH_INEQ** | Multi-impurities support | True/False |
| -D**VERBOSE** | Verbose CMake output | True/False |
| -D**BUILD_TYPE** | Compilation flags | RELEASE/TESTING/DEBUG |

The default target builds and installs both the main library and the C-bindings. Separate build targets for each component are available. A recap message is printed at the end of the CMake configuration step to guide the user through the compilation, installation and OS loading process.

### 2.3.2 Anaconda

Installation is also available through Anaconda packages for Linux and macOS systems, within a virtual environment containing Python ($\geq 3.10$). The Conda package installation procedure for a virtual environment called `myenv` reads

```
conda create -n myenv
conda activate myenv
conda install -c conda-forge -c edipack edipack
```

and installs a bundle of SciFortran and EDIpack libraries together with specific `pkg-config` configuration files, which can be used to retrieve compilation and linking flags. In order to compile Fortran/C++ programs, the `compilers` conda package will need to be installed.

## 2.4 Environment Variables

In order to avoid possible conflicts or the requirement for administrative privileges, the results of the building step are installed by default in user's `HOME` directory. As a consequence, the environment variables holding the library and include paths will need to be updated by the user. We offer different ways to perform this action:

1. A CMake-generated configuration file for an environment module which allows users to load and unload the library at any time. This is preferred solution for HPC systems.

2. A CMake-generated bash script to be sourced (once or permanently) in any shell session to add EDIpack library to the default environment.

3. A CMake-generated `pkg-config` file to be added in the `PKG_CONFIG_PATH` itself.

A recap message with all instructions is automatically generated at the end of the installation procedure.

# 3 Implementation

In this section we introduce the generic quantum impurity problem and present a detailed overview of the EDIpack library implementation.

## 3.1 The quantum impurity problem

We consider a quantum impurity problem described by the Hamiltonian:

$$\hat{H} = \hat{H}_{\text{imp}} + \hat{H}_{\text{bath}} + \hat{H}_{\text{hyb}} + \hat{H}_{\text{ph}} + \hat{H}_{\text{e}-\text{ph}}$$

which characterizes a multi-orbital quantum impurity coupled to an electronic bath and local phonons (e.g. Holstein modes). For now, we assume no specific symmetries.

### 3.1.1 Impurity Hamiltonian

The impurity Hamiltonian is split into a quadratic and an interacting part,

$$\hat{H}_{\text{imp}} = \hat{H}_{\text{imp}}^0 + \hat{H}_{\text{imp}}^{\text{int}}. \tag{1}$$

The first term has the form

$$\hat{H}_{\text{imp}}^0 = \sum_{\alpha\beta\sigma\sigma'} h^0_{\alpha\beta\sigma\sigma'} d^\dagger_{\alpha\sigma} d_{\beta\sigma'} \tag{2}$$

and represents the non-interacting part, with $d_{\alpha\sigma}$ ($d^\dagger_{\alpha\sigma}$) being the annihilation (creation) operators for impurity electrons in orbital $\alpha = 1, \ldots, N_\alpha$, where $N_\alpha$ is the number of orbitals, and spin $\sigma = \uparrow, \downarrow$. The internal structure of this quadratic part is captured by the matrix $h^0_{\alpha\beta\sigma\sigma'}$, which can include orbital-dependent on-site energies or amplitudes, spin-orbit coupling or other single-particle terms. In EDIpack the matrix $\hat{h}_0$ must be set by the user before the solver is initialized using the function `ed_set_Hloc`. The latter accepts complex arrays of rank-2 or 4 as only argument. In a quantum embedding framework, e.g. DMFT, the matrix $\hat{h}_0$ corresponds to the local part of the non-interacting Hamiltonian. Some examples of use can be found in Sec. 5.

The interaction part of the impurity Hamiltonian, $\hat{H}^{\text{int}}$ can, in principle, contain any set of two-body operators:

$$\hat{H}_{\text{imp}}^{\text{int}} = \frac{1}{2} \sum_{ijkl} d^\dagger_i d^\dagger_j U_{ijkl} d_l d_k. \tag{3}$$

where the symbols $\{ijkl\}$ collect both orbital and spin indices, e.g. $i = \alpha\sigma$. However, we typically adopt a generic formulation of the local multi-orbital Hubbard-Kanamori interaction [93]:

$$
\begin{aligned}
\hat{H}_{\text{imp}}^{\text{int}} = {} & U \sum_\alpha n_{\alpha\uparrow} n_{\alpha\downarrow} + U' \sum_{\alpha\neq\beta} n_{\alpha\uparrow} n_{\beta\downarrow} + (U' - J) \sum_{\alpha<\beta,\sigma} n_{\alpha\sigma} n_{\beta\sigma} \\
& - J_X \sum_{\alpha\neq\beta} d^\dagger_{\alpha\uparrow} d_{\alpha\downarrow} d^\dagger_{\beta\downarrow} d_{\beta\uparrow} + J_P \sum_{\alpha\neq\beta} d^\dagger_{\alpha\uparrow} d^\dagger_{\alpha\downarrow} d_{\beta\downarrow} d_{\beta\uparrow},
\end{aligned}
\tag{4}
$$

where we introduced the occupation number operator $n_{\alpha\sigma} = d^\dagger_{\alpha\sigma} d_{\alpha\sigma}$. The first three terms represent the density-density part of the interaction, where $U$ is the local intra-orbital Coulomb repulsion, $U'$ the inter-orbital one and $J$ the Hund's coupling [93–98]. The last two terms are, respectively, the spin-exchange ($J_X$) and the pair-hopping ($J_P$). In the three-orbital case $N_\alpha = 3$ a fully symmetric $SU(3)_{\text{orbital}} \otimes SU(2)_{\text{spin}} \otimes U(1)_{\text{charge}}$ form of the interaction is obtained by setting $U' = U - 2J$ and $J_X = J_P = J$ [93]. Alternative choices that preserve part of the combined symmetry group can be made for other numbers of orbitals [93].

### 3.1.2 Bath and Hybridization

The coupling between the impurity and the bath is described by

$$\hat{H}_{\text{bath}} = \sum_p \sum_{\alpha\beta\sigma\sigma'} h^p_{\alpha\beta\sigma\sigma'} a^\dagger_{p\alpha\sigma} a_{p\beta\sigma'},$$

$$\hat{H}_{\text{hyb}} = \sum_p \sum_{\alpha\beta\sigma\sigma'} V^p_{\alpha\beta\sigma\sigma'} d^\dagger_{\alpha\sigma} a_{p\beta\sigma'} + H.c., \tag{5}$$

where $p = 1, \ldots, N_{\text{bath}}$ indexes the bath elements. The operators $a_{p\alpha\sigma}$ $(a^\dagger_{p\alpha\sigma})$ correspond to the destruction (creation) of bath electrons with index $p$, orbital $\alpha$ and spin $\sigma$. Any bath element can be composed of several electronic levels according to the chosen bath topology (see Sec. 3.6). The properties of each bath element are described by the matrix $h^p_{\alpha\beta\sigma\sigma'}$ while the amplitude $V^p_{\alpha\beta\sigma\sigma'}$ describes the coupling with the impurity. The bath parametrization is handled in EDIpack using reverse communication strategy as will be discussed later in Sec. 3.6.

### 3.1.3 Electron-Phonon coupling

Finally, we include a local electron-phonon coupling on the impurity site described by the Hamiltonian terms:

$$\hat{H}_{\text{ph}} = \sum_m \omega_{0m} b^\dagger_m b_m,$$

$$\hat{H}_{\text{e}-\text{ph}} = \sum_m [\hat{O}_m - A_m](b_m + b^\dagger_m), \tag{6}$$

where $m = 1, \ldots, M$ indexes the number of local phonon modes, and $b_m$ $(b^\dagger_m)$ are the destruction (creation) operators for a phonon with frequency $\omega_{0m}$. Each phonon mode is coupled to a bilinear fermionic operator $\hat{O}_m = \sum_{\alpha\beta,\sigma} g^m_{\alpha\beta} d^\dagger_{\alpha,\sigma} d_{\beta,\sigma}$ on the impurity, while $A_m$ is a constant acting as a displacement field. For single-orbital impurities, the relevant case is $\hat{O}_1 = g_1 n_{1d} = g_1 \sum_\sigma d^\dagger_{1\sigma} d_{1\sigma}$, which defines the Holstein model [99]. For two-orbital models we included also the two Jahn-Teller modes [100], for which $\hat{O}_2 = g_2(n_{1d} - n_{2d})$ [101] and $\hat{O}_3 = g_3 \sum_\sigma (d^\dagger_{1\sigma} d_{2\sigma} + H.c.)$.

For each phonon mode, one has to introduce a cut-off in the phonon number to avoid unbounded growth of the phonons Fock space. The cutoff has to be chosen in a problem dependent way, also considering the value of the electron-phonon coupling [102, 103] and its interplay with the electron-electron interaction [104–110]. The larger the number of excited phonons in the ground state, the larger the cutoff. For practical calculations a cutoff of about 20 phonons is usually sufficient to achieve converged results, except for the deep polaronic regime [102].

In many cases, one can reduce the number of excited phonons in the ground state, and consequently the required cutoff, by appropriately choosing the shift $A_m$. In practice one can use $A_m = \langle \hat{O}_m \rangle$ and redefine a shifted bosonic fields $\tilde{b}^{(\dagger)}_m = b^{(\dagger)}_m + \frac{\langle \hat{O}_m \rangle}{\omega_0}$. This choice would ensure that $\langle \tilde{b}^\dagger_m + \tilde{b}_m \rangle = 0$ effectively reducing the number of phonons required for the calculation. Observables on the physical bosons are linked to the new one via the previous transformation. Although feasible, dealing with more than one phonon mode quickly becomes computationally very demanding, thus we will consider only one mode and we drop the $m$ index in the rest of this work.

The phonon frequency is passed to the EDIpack program via the variable `W0_PH` to be read from the input file. The displacement field acting on the phonon is passed through the variable `A_PH` from the input file. For every phonon mode, the $g$ coupling can be

written as a matrix $g_{\alpha\beta}$ which can be passed in two different ways. If the matrix is diagonal ($g_{\alpha\beta} = g_\alpha \delta_{\alpha\beta}$), from the input file one can simply write in order the values $g_\alpha$ in the variable `G_PH`. For any general shape one can pass to the input file a `GPHfile` variable containing the name of the file storing the $g_{\alpha\beta}$ array. If `GPHfile=NONE`, the matrix $g_{\alpha\beta}$ is read from the variable `G_PH`, otherwise it is read from the `GPHfile`.

### 3.1.4 System setup

We consider a quantum impurity system comprised by a single multi-orbital impurity, a bath discretized into a number $N_{\text{bath}}$ of *elements* possibly endowed with an internal structure, and a number of available phonons. The total size of the system is determined by three contributions: i) the number of impurity orbitals $N_\alpha$, ii) the total count of electronic bath levels $N_{\text{b}}$, and iii) by the maximum number of allowed phonons $N_{\text{ph}}$, introducing a cutoff to the unbounded dimension of the local phonon Hilbert space. The number of bath degrees of freedom $N_{\text{b}}$ is a function of the bath topology and of $N_{\text{bath}}$, which is determined internally by EDIpack. For instance, for a simple case of a purely electronic problem, each bath element corresponds to an independent electronic level coupled to the impurity, thus $N_{\text{b}} \equiv N_{\text{bath}}$. The total number of electronic levels in quantum impurity systems is indicated with $N_{\text{s}}$.

The setup of the quantum impurity problem is implemented in `ED_SETUP` through the input variables `Nspin=` $N_\sigma$, `Norb=` $N_\alpha$ and `Nbath=` $N_{\text{bath}}$ globally defined in `ED_INPUT_VARS`. The variable `Ns=` $N_{\text{s}}$, evaluated in `ed_setup_dimensions`, corresponds to the total number of electronic levels. The local non-interacting Hamiltonian $h^0_{\alpha\beta\sigma\sigma'}$ is specified using the function `ed_set_hloc` from `ED_AUX_FUNX`. The setup of the bath matrices $h^p_{\alpha\beta\sigma\sigma'}$ requires a more involved procedure, which will be illustrated in Sec. 3.6.

## 3.2 The Fock basis states

The Fock space of the quantum impurity problem is defined as $\mathcal{F} = \mathcal{F}_{\text{e}} \otimes \mathcal{F}_{\text{ph}}$, where:

$$\mathcal{F}_{\text{e}} = \bigoplus_{n=0}^{N_s} S_- \mathcal{H}_{\text{e}}^{\otimes n}$$

is the electronic Fock space built from the local electronic Hilbert space $\mathcal{H}_{\text{e}}$, and

$$\mathcal{F}_{\text{ph}} = \bigoplus_{n=0}^{N_q} S_+ \mathcal{H}_{\text{ph}}^{\otimes n}$$

is the phonon Fock space, with local phonon Hilbert space $\mathcal{H}_{\text{ph}} = \{|0\rangle, |1\rangle, \ldots, |N_{\text{ph}}\rangle\}$. Here $(S_-) \, S_+$ is the (anti-)symmetrization operator. The total dimension of the Fock space is

$$D = 4^{N_s}(N_{\text{ph}} + 1) = D_{\text{e}} D_{\text{ph}},$$

highlighting the exponential growth with the number of electronic levels.

The quantum states in the space $\mathcal{F}$ are naturally represented in the occupation number formalism of second quantization, i.e. the Fock basis. For a system of $N_{\text{s}}$ electrons, each Fock state is given as $|p\rangle|\vec{n}\rangle$, with

$$|\vec{n}\rangle = |n_{1\uparrow}, \ldots, n_{N_s\uparrow}, n_{1\downarrow}, \ldots, n_{N_s\downarrow}\rangle, \tag{7}$$

where $p = 1, \ldots, N_{\text{ph}}$ is the number of local phonons and $n_{a\sigma} = 0, 1$ indicates the absence or presence of an electron with spin $\sigma$ at level $a$.

The electronic part of the Fock state $|\vec{n}\rangle$ is represented as a binary string of length $2N_s$. Thus, any such state can be encoded in a computer using a sequence of $2N_s$ bits, or equivalently, as an integer $I = 0, \ldots 2^{2N_s} - 1$ such that $|\vec{n}\rangle = |I\rangle$. The electronic destruction and creation operators, $c_{a\sigma}$ and $c_{a\sigma}^\dagger$ respectively, act on the Fock space as:

$$c_{a\sigma}|\vec{n}\rangle = \begin{cases} (-1)^{\#_{a\sigma}} |\ldots, n_{a\sigma}-1, \ldots\rangle & \text{if } n_{a\sigma}=1 \\ 0 & \text{otherwise} \end{cases};$$

$$c_{a\sigma}^\dagger|\vec{n}\rangle = \begin{cases} (-1)^{\#_{a\sigma}} |\ldots, n_{a\sigma}+1, \ldots\rangle & \text{if } n_{a\sigma}=0 \\ 0 & \text{otherwise} \end{cases}$$

where $\#_{a\sigma} = \sum_{i\sigma' < a\sigma} n_{i\sigma'}$ accounts for the fermionic sign imposed by the Pauli principle. The bosonic operators $b$ and $b^\dagger$ act on the phonon part $|p\rangle$ of the Fock state as

$$b|p\rangle = \sqrt{p}|p-1\rangle,$$
$$b^\dagger|p\rangle = \sqrt{p+1}|p+1\rangle.$$

The implementation of Fock states and operators can be found in `ED_AUX_FUNX`. In particular, in that module we define the bitwise action of the fermionic creation and annihilation operators as functions `CDG` and `C`, used throughout the code. Additionally, we provide the function `bdecomp` for reconstructing the Fock state bit sequence from the integer representation.

### 3.3 Conserved quantum numbers

To mitigate the exponential scaling of the Fock space dimension, it is essential to exploit suitable symmetries. Specifically, if the Hamiltonian $H$ commutes with a set of operators $\vec{\mathcal{Q}} = [\mathcal{Q}_1, \mathcal{Q}_2, \ldots, \mathcal{Q}_M]$ such that $[H, \mathcal{Q}_i] = 0$, the Fock space can be decomposed into smaller, disjoint symmetry sectors labeled by quantum numbers $\vec{Q}$.

In the context of quantum impurity problems, two commonly used symmetries are: i) conservation of the total charge $N$ and ii) conservation of the total magnetization $S_z$. Although the total spin operator $S^2$ may also be conserved, its implementation is computationally challenging and typically provides only marginal gains, making it less practical in many cases. As the symmetries apply to the electronic terms, here we discuss the organization of the electronic Fock states only. The presence of phonons is accounted for by exploiting the tensor structure of the global Fock space.

In EDIpack, three different symmetry configurations controlled by the input variable `ed_mode` = **normal**, **superc**, **nonsu2** are available. A summary of the sector properties is provided in Table (2).

- The **normal** case conserves both the electrons' total occupation $N$ and total magnetization $S_z$, or equivalently, the total number of electrons with spin up $N_\uparrow$ and down $N_\downarrow$. Optionally, the symmetry can be extended to act independently on each orbital and spin component, i.e. $\vec{N}_\sigma = [N_\sigma^1, \ldots, N_\sigma^{N_\alpha}]$. This specific case is discussed extensively Ref. [82], so we will cover it no further here recalling that any consideration concerning the `normal` case extends directly to this case as well.

- The **superc** case maintains only the $S_z$ conservation, allowing the total charge to not be conserved. This setting captures systems with $s$-wave superconductivity, including intra- and inter-orbital pairing.

- The **nonsu2** case conserves the total charge $N$ while allowing the spin symmetry group to be broken. This scenario captures effects such as local spin-orbit coupling $\vec{L} \cdot \vec{S}$, in-plane spin ordering [111] or in-plane spin-triplet exciton condensation [112, 113].

| ed_mode | *Quantum Numbers* | *Sector Dimension* |
|---|---|---|
| **normal** | $[N, S_z] \Leftrightarrow [N_\uparrow, N_\downarrow]$ | $\binom{N_{\rm s}}{N_\uparrow}\binom{N_{\rm s}}{N_\downarrow}$ |
| **superc** | $S_z \equiv N_\uparrow - N_\downarrow$ | $\sum_m 2^{N_{\rm s} - S_z - 2m} \binom{N_{\rm s}}{N_{\rm s} - S_z - 2m}\binom{S_z + 2m}{m}$ |
| **nonsu2** | $N \equiv N_\uparrow + N_\downarrow$ | $\binom{2N_{\rm s}}{N}$ |

Table 2: **Quantum Numbers**. A table summarizing the possible values of the input variable `ed_mode` selecting the symmetries of the problem (first column). To each value correspond different quantum numbers listed in the second column. The third column reports the dimension of the electronic symmetry sector $\mathcal{S}_{\vec{Q}}$ for a given value of the quantum numbers $\vec{Q}$.

From a computational perspective, constructing a symmetry sector amounts to defining an injective map $\mathcal{M} : \mathcal{S}_{\vec{Q}} \to \mathcal{F}_e$ that relates the consecutive indexing of the electronic states $|i\rangle$ within a given symmetry sector $\mathcal{S}_{\vec{Q}}$ to the Fock states $|I_e\rangle$ in the electronic Fock space in occupation number representation [114]. This map is typically implemented as a rank-1 integer array, whose size corresponds to the dimension of the sector $D_{\mathcal{S}_{\vec{Q}}}$. The global Fock state $|I\rangle = |I_e\rangle|p\rangle$, including the phonon part, is univocally determined by the relation: $I \to I_e + p\, 2^{2N_s}$.

The **normal** case deserves a special note. Since $N_\uparrow$ and $N_\downarrow$ are conserved independently, the local Hilbert space and the electronic Fock space can be factorized as $\mathcal{H}_e = \mathcal{H}_{e\uparrow} \otimes \mathcal{H}_{e\downarrow}$ and $\mathcal{F}_{\rm e} = \mathcal{F}_{e\uparrow} \otimes \mathcal{F}_{e\downarrow}$, respectively. Consequently, each electronic Fock state can be written as a product $|\vec{n}_\uparrow\rangle \otimes |\vec{n}_\downarrow\rangle$. This factorization splits the symmetry sector as $\mathcal{S}_{\vec{Q}} = \mathcal{S}_{N_\uparrow} \otimes \mathcal{S}_{N_\downarrow}$, and finally the sector map can be expressed as the product $\mathcal{M} = \mathcal{M}_\uparrow \otimes \mathcal{M}_\downarrow$. The electronic part of each sector state $|i\rangle = |i_\uparrow\rangle \otimes |i_\downarrow\rangle$ in this factorized basis is labeled by two integers $[i_\uparrow, i_\downarrow]$, with $i_\sigma = 1, \ldots, D_{\mathcal{S}_\sigma}$ such that $i = i_\uparrow + i_\downarrow D_{\mathcal{S}_\downarrow}$. The maps $\mathcal{M}_\sigma$ then connect these basis states to Fock states $|I_e\rangle = |I_\uparrow\rangle|I_\downarrow\rangle$, labeled by two integers $[I_\uparrow, I_\downarrow]$ as $I_e = I_\uparrow + I_\downarrow 2^{N_s}$. For a more detailed discussion on the structure of the Fock basis in this case, see Ref. [82].

The presence of a symmetry reduces the electronic Hamiltonian matrix to a block-diagonal form, where each block labeled by $\vec{Q}$ has dimension $D_{\mathcal{S}_{\vec{Q}}}$. The sector Hamiltonian matrix $H_{\mathcal{S}}$ is represented in the basis $|i\rangle \in \mathcal{S}_{\vec{Q}}$ as a sparse matrix. In the **normal** case this block structure is particularly symmetric due to the factorized nature of the sectors as discussed above, see also Ref. [82]. The analysis of the spectrum is then reduced to inspecting the Hamiltonian within each symmetry sector, or to a subset thereof if additional constraints are present.

In EDIpack, the implementation of symmetry sectors is managed through the `sector` object, which is defined in `ED_VARS_GLOBAL`. This object contains all the relevant information for defining the symmetry, including the sector dimensions, quantum numbers and the map $\mathcal{M}$. The constructor (destructor) for this object is defined in the `ED_SECTORS` module via the function `build_sector` (`delete_sector`). These functions use different algorithms depending on the nature of the quantum numbers $\vec{Q}$, i.e. the value of `ed_mode`. The key idea is to loop over the Fock states and enforce the quantum number constraint. The following code snippets summarize the various available implementations and serve as a basis to understand other parts of the software, see Sec. 3.5.2.

**Normal**

```
i=0
do Iup=0,2**Nbit-1
  nup_ = popcnt(Iup)
  if(nup_ /= Nups(1))cycle
  i = i+1
  H(iud)%map(i) = lup
enddo
i=0
do Idw=0,2**Nbit-1
  ndw_= popcnt(Idw)
  if(ndw_ /= Ndws(1))cycle
  i = i+1
  H(iud+Ns)%map(i)=Idw
enddo
```

**Superc**

```
i=0
do Idw=0,2**Ns-1
  ndw_= popcnt(idw)
  do Iup=0,2**Ns-1
    nup_ = popcnt(iup)
    sz_  = nup_ - ndw_
    if(sz_ /=self%Sz)cycle
    i=i+1
    self%H(1)%map(i)= &
        Iup+Idw*2**N
  enddo
enddo
```

**Nonsu2**

```
i=0
do Idw=0,2**Ns-1
  ndw_= popcnt(Idw)
  do Iup=0,2**Ns-1
    nup_ = popcnt(Iup)
    nt_  = nup_ + ndw_
    if(nt_/=self%Ntot)cycle
    i=i+1
    self%H(1)%map(i)= &
        Iup+Idw*2**Ns
  enddo
enddo
```

In addition to the basic construction and destruction routines, the `ED_SECTORS` module also includes functions to retrieve sector indices and quantum number information. Furthermore, it provides a set of essential functions for applying creation (`apply_op_CDG`), destruction (`apply_op_C`) or arbitrary linear combinations (`apply_Cops`) of these to a given Fock state $|v\rangle \in \mathcal{S}$, i.e.:

$$\mathcal{O}|v\rangle = \sum_i \left( a_i C_{\alpha_i,\sigma_i} + b_i C^\dagger_{\beta_i,\rho_i} \right) |v\rangle$$

with $a_i, b_i \in \mathbb{C}$, $\alpha_i, \beta_i = 1, \ldots, N_\alpha$ and $\sigma_i, \rho_i = \uparrow, \downarrow$. These functions are widely used throughout the code.

## 3.4 Interaction setup

EDIpack offers two distinct methods to setup the interaction terms $\hat{H}^{\mathrm{int}}_{\mathrm{imp}}$ in Eq. (1), which are controlled by the input variables `ED_USE_KANAMORI` and `ED_READ_UMATRIX`.

The generalized Hubbard-Kanamori interaction, as defined in Eq. (4), is natively supported and controlled by the input parameters `ULOC`= $U$, `UST`= $U'$, `JH`= $J$, `JX`= $J_X$ and `JP`= $J_P$. These quantities can be specified either in the input file or directly via the command line, provided the logical input parameter `ED_USE_KANAMORI=T` and setting `ED_READ_UMATRIX=F`. In this mode, the maximum number of impurity orbitals `NORB`= $N_\alpha$ is limited to 5.

Alternatively, setting `ED_USE_KANAMORI=F` and `ED_READ_UMATRIX=T`, a list of two-body interaction operators can be supplied by the user in a properly formatted text file, whose name is specified by the input variable `UMATRIX_FILE`. The file must have the suffix `<UMATRIX_FILE>.restart` and should be formatted as follows:

```
<NORB> BANDS
i1 j1 k1 l1 U_i1j1k1l1
i2 j2 k2 l2 U_i2j2k2l2
...
```

where `NORB` is the number of orbitals $N_\alpha$ and $\{ijkl\}$ refer to the combined spin (`u/d`) and internal (e.g. orbital index) degrees of freedom. For example, if `i1` refers to orbital 1, spin up, it is written explicitly as `1 u`. Each line defines a two-body second-quantized operator of the form

$$\hat{H}^{\mathrm{int}} = \frac{1}{2} d_i^\dagger d_j^\dagger U_{ijkl} d_l d_k.$$

Note that the coefficient indices $l$ and $k$ are swapped in the input file with respect to the two-body operator they implement. The $\frac{1}{2}$ prefactor is applied internally by the parsing routine and need not be accounted for by the user. Empty lines and lines starting with `#, %, !` are ignored.

While `ED_READ_UMATRIX` and `ED_USE_KANAMORI` are mutually exclusive, it is allowed to set both to `F`. In this case, the user can specify additional interaction terms in the calling program or script by means of the procedure `add_twobody_operator`, before the impurity problem is solved. The input of this function has the same form of each line of the `UMATRIX` file. A list of user-provided interaction terms will be added to those read from file or from the Hubbard-Kanamori coefficients in case the relative flags are `T`. To clear the list, the procedure `reset_umatrix` has to be called.

For reference and future use, when the impurity problem is solved, a list of the interaction operators used in each call to the solver is saved in the output file `<UMATRIX_FILE>.used`. The reading, writing, parsing and initialization of the two-body interaction terms are implemented in the module `ED_PARSE_UMATRIX`.

## 3.5 Classes

The use of objects and classes greatly simplifies the implementation of critical mathematical concepts required for solving quantum impurity problems. This section provides an overview of the main classes used in EDIpack, focusing on their structure and functionality.

### 3.5.1 Sparse matrix

Sparse matrix storage is handled through a dedicated class in the `SPARSE_MATRIX` module. This module defines the `sparse_matrix_csr` object, which stores sparse matrices as hash tables. Each key corresponds to a row index, while the associated value contains a pair of dynamic arrays: one for the non-zero matrix elements and one for their respective column indices. In an MPI framework, each matrix instance on any given thread stores only a given set $Q = D_{\mathcal{S}}/N_{cpu}$ of matrix rows. In addition, the elements local in the memory are stored in a separate set of rows, i.e. `loc`, and used to speed up the execution of matrix-vector product. We keep track, for each thread, of the relative position of the matrix rows $q = 1, \ldots, Q$ with respect to the global indexing $i = 1, \ldots, D_{\mathcal{S}}$, through the definition of additional indices `istart, iend, ishift`. These correspond, respectively, to the initial global index $i_1$ for $q = 1$, the final global index $i_Q$ for $q = Q$ and the integer shift mapping the index $q$ into $i$. Each instance of `sparse_matrix_csr` can be stored serially (one copy per process) or distributed across multiple processes with rows dynamically assigned to each process.

```fortran
! sparse row: contains the NNZ elements on any row of the sparse matrix.
type sparse_row_csr
   integer                                 :: size  !size of the row = NNZ
   real(8),dimension(:),allocatable        :: dvals !rank-1 array for double precision
   complex(8),dimension(:),allocatable     :: cvals !rank-1 array for double complex
   integer,dimension(:),allocatable        :: cols  !rank-1 array for column indices
end type sparse_row_csr
! Sparse Matrix: allocatable array of sparse rows
type sparse_matrix_csr
   type(sparse_row_csr),dimension(:),pointer :: row  !array of {\tt sparse\_row\_csr}
   integer                                 :: Nrow !total number of rows
   integer                                 :: Ncol !total number of columns
   logical                                 :: status=.false. !Allocation status
#ifdef _MPI
   type(sparse_row_csr),dimension(:),pointer :: loc    !array for the diagonal blocks
   integer                                 :: istart=0 !start index MPI range
   integer                                 :: iend=0   !end index MPI range
   integer                                 :: ishift=0 !shift index MPI range
```

```
   logical                                    :: mpi=.false.
#endif
end type sparse_matrix_csr
```

Matrix elements are inserted using the `sp_insert_element` procedure, which leverages the Fortran intrinsic `move_alloc` for faster execution compared to implicit reallocation, i.e. `vec=[vec,new_element]`. This approach offers several key advantages, including efficient memory management for matrices with unknown numbers of non-zero elements per row, and $O(1)$ element access time, both of which are critical for the efficient implementation of Krylov subspace methods.

### 3.5.2  Sparse map

As will be discussed in Sec. 3.10, the construction of a symmetry sector often requires associating each sector state $|i\rangle$ with information about the corresponding electronic Fock state $|I_e\rangle$. In this context, the electronic Fock state can be decomposed into bit chunks, e.g. $|\vec{n}\rangle = |\vec{i}_\uparrow \vec{b}_\uparrow \vec{i}_\downarrow \vec{b}_\downarrow\rangle$, reflecting a natural grouping into impurity and bath components and where we highlight the overall ordering in spin-$\uparrow$ and spin-$\downarrow$ parts (see Eq. (7)).

The module `ED_SPARSE_MAP` provides an efficient hash table implementation, `sparse_map` which is part of the `sector` object and stores the relation between these bit chunks. Specifically, for each impurity configuration $\vec{i}_\sigma$ (*key*), it maintains a list of compatible bath configurations $\vec{b}_\sigma$ (*values*) consistent with the conserved quantum numbers of the given sector (see Sec. 3.3). The `sparse_map` objects $\mathcal{P}$ are constructed upon call of the `build_sector` (i.e. the `sector` constructor) using different algorithms for each value of `ed_mode`.

```
!Sparse row: contains bath state values for each impurity key
type sparse_row
   integer                                    :: size          !current size
   integer                                    :: bath_state_min!smallest bath state
   integer                                    :: bath_state_max!largest bath state
   integer,dimension(:),allocatable           :: bath_state    !Values:rank-1 bath states
   integer,dimension(:),allocatable           :: sector_indx   !bath state sector indices
end type sparse_row
!Sparse map: array of sparse rows
type sparse_map
   type(sparse_row),dimension(:),pointer :: imp_state      !key: impurity states
   integer                                    :: Nimp_state    !key upper limit
   logical                                    :: status=.false.!allocation status
end type sparse_map
```

In the **normal** mode, the `sector` object contains two separate `sparse_maps` objects $\mathcal{P}_{\sigma=\uparrow,\downarrow}$, reflecting the factorization of the Fock and sector states into independent spin components: $|J_e\rangle = |J_\uparrow\rangle \otimes |J_\downarrow\rangle \xleftarrow{\mathcal{M}} |j_\uparrow\rangle \otimes |j_\downarrow\rangle = |j\rangle$. The sparse maps are built as follows. For any spin Fock state appearing in the sector $|j\rangle \xrightarrow{\mathcal{M}} |J_\sigma\rangle = |\vec{I}_\sigma \vec{B}_\sigma\rangle$, the *key* is determined by the integer $I_\sigma$ corresponding to the impurity bitset $\vec{I}_\sigma$. The *values* are given by all integers $B_\sigma$ corresponding to any bath bit set $\vec{B}_\sigma$ associated with $\vec{I}_\sigma$ by the sector symmetry constraint: $\#\vec{I}_\sigma + \#\vec{B}_\sigma = \text{popcnt}(I_\sigma) + \text{popcnt}(B_\sigma) = N_\sigma$. Thus, any given *key-value* combination reconstructs an integer $J_\sigma$ representing a Fock state in the given spin sector according to the rule $J_s = I_\sigma + 2^{N_{\text{imp}}} B_\sigma$, where $N_{\text{imp}}$ is the number of impurity bits.

In both **superc** and **nonsu2** modes a single sparse map is used. Here, each electronic Fock state $|J_e\rangle$ is associated with four integers $I_\uparrow, B_\uparrow, I_\downarrow, B_\downarrow$ representing the bit decomposition of the state. These integers satisfy the relationship

$$J_e = I_\uparrow + B_\uparrow 2^{N_{\text{imp}}} + (I_\downarrow + B_\downarrow 2^{N_{\text{imp}}})2^{N_{\text{s}}} \tag{8}$$

where $N_{\text{imp}}$ is the number of impurity bits and $N_s$ is the total number of spin orbitals. We define the *key* for the sparse map as $I = I_\uparrow + I_\downarrow 2^{N_{\text{imp}}}$ and the corresponding *values* as the integers $B = B_\uparrow + B_\downarrow 2^{N_b}$. Note that, for the purpose of storing the key-value pairs, the impurity and bath indices are determined using an algorithm different from Eq. (8). These relations ensure a better scaling and can be easily inverted to obtain $I_{\sigma=\uparrow,\downarrow}$ from $I$ and $B_{\sigma=\uparrow,\downarrow}$ from $B$ in order to reconstruct the Fock state $|J\rangle$. The four integers satisfy the symmetry constraints: $\texttt{popcnt}(I_\uparrow + B_\uparrow 2^{N_{\text{imp}}}) + \texttt{popcnt}(I_\downarrow + B_\downarrow 2^{N_{\text{imp}}}) = N$ for the **nonsu2** case and $\texttt{popcnt}(I_\uparrow + B_\uparrow 2^{N_{\text{imp}}}) - \texttt{popcnt}(I_\downarrow + B_\downarrow 2^{N_{\text{imp}}}) = S_z$ for the **superc** case. This structure ensures an efficient and compact representation of the many-body Hilbert space, enabling a rapid state lookup during the diagonalization process.

An overview of the implementation leveraging the Fortran intrinsic $\texttt{ibits(I,pos,len)}$ is reported in the following listing:

**Normal**

```
i=0
do Iup=0,2**Nbit-1
  nup_ = popcnt(Iup)
  if(nup_ /= Nups(1))cycle
  i=i+1
  !...
  iImp  = ibits(iup,0,Norb)
  iBath = ibits(iup,Norb,Norb*Nbath)
  call sp_insert_state(self%H(1)%sp,
                       iImp,&
                       iBath,&
                       i)
enddo
!
i=0
do Idw=0,2**Nbit-1
  ndw_= popcnt(Idw)
  if(ndw_ /= Ndws(1))cycle
  i=i+1
  !...
  iIMP  = ibits(idw,0,Norb)
  iBATH = ibits(idw,Norb,Norb*Nbath)
  call sp_insert_state(self%H(2)%sp,&
                       iImp,&
                       iBath,&
                       i)
enddo
```

**Superc/Nonsu2**

```
i=0
do Idw=0,2**Ns-1
  ndw_= popcnt(idw)
  do Iup=0,2**Ns-1
    nup_ = popcnt(iup)
    select case (ed_mode)
      case('superc')
        sz_  = nup_ - ndw_
        if(sz_ /= self%Sz)cycle
      case('nonsu2')
        nt_  = nup_ + ndw_
        if(nt_ /= self%Ntot)cycle
    end select
    i=i+1
    !...
    iImpUp = ibits(iup,0,Norb)
    iImpDw = ibits(idw,0,Norb)
    iBathUp= ibits(iup,Norb,Norb*Nbath)
    iBathDw= ibits(idw,Norb,Norb*Nbath)
    iImp   = iImpUp  + iImpDw*(2**Norb)
    iBath  = iBathUp + iBathDw*(2**Ns)
    call sp_insert_state(self%H(1)%sp,&
                         iImp,&
                         iBath,&
                         dim)
  enddo
enddo
```

### 3.5.3   Eigenspace

The module $\texttt{ED\_EIGENSPACE}$ contains the implementation of the class $\texttt{sparse\_espace}$, which maintains a linked list of selected eigenvalues, eigenvectors, and quantum numbers for the low-lying spectrum of the quantum impurity Hamiltonian. In MPI mode, the memory footprint of each eigenvector is minimized by automatically distributing it among all processors in suitable shares according to the value of $\texttt{ed\_mode}$. While for zero-temperature calculations only the ground state (with its degeneracy) is stored, for finite temperature the list includes all the necessary excited states. If the input variable $\texttt{ED\_TWIN=T}$, the *twin sectors*, i.e. sectors with symmetric values of the quantum numbers ($S_z \to -S_z$ and $N \to N_s - N$), are excluded from the list and their contribution is reconstructed by symmetry where needed. A truncation mechanism is put in place to avoid the unbounded growth of this list. On the first call, a fixed number of states per sector $\texttt{lanc\_nstates\_sector}$ is collected, up to a maximum total $\texttt{lanc\_nstates\_total}$, both set via input. The list is then truncated to retain only the states satisfying the condition

$$e^{-\beta(E_i - E_0)} < \texttt{cutoff},$$

where $E_i$ is the energy of the $i^{\text{th}}$ state in the list, $E_0$ is the ground state energy, $\beta = 1/T$ is the inverse temperature ($k_B = 1$) and `cutoff` is an input parameter fixing an *a priori* energy threshold. In the following we indicate with $N_{\texttt{state\_list}}$ the number of eigen-solutions stored in the list. By successive call to the EDIpack solver, the list dynamically adjusts to optimize the distribution of states across sectors, balancing memory use and computation time. The variable `lanc_nstates_step` controls the increase or decrease in the number of states per sector. A histogram of the number of states per sector is produced after each diagonalization to monitor this distribution.

### 3.5.4  gfmatrix

One of the main goals of the EDIpack is to evaluate dynamical correlations functions (DCF), e.g. $C(t) = \langle \mathcal{T}[A(t)A^\dagger] \rangle$. Using Krylov methods, these DCFs can be expressed as truncated Källén-Lehmann spectral sums of the form

$$C(z) = \tfrac{1}{Z} \sum_{n=1}^{N_{\texttt{state\_list}}} e^{-\beta E_n} \sum_{m=1}^{N} \tfrac{|w_{mn}(\mathcal{A})|^2}{z - dE_{mn}}$$

where $w_{mn}(\mathcal{A})$ is the weight determined by the amplitude of operator $\mathcal{A}$ between the $m^{\text{th}}$ eigenstate and the $n^{\text{th}}$ Krylov basis vector, $dE_{mn} = E_m - E_n$ is the corresponding excitation energy.

The module `ED_GFMATRIX` implements the class `gfmatrix`, which provides an efficient and flexible representation of DCFs. It is structured as a hierarchical, or *multi-layered*, data container that organizes spectral weights and poles across three levels: (i) the initial eigenstate $|n\rangle$ of the Hamiltonian spectrum, (ii) the amplitude channel associated with this state, corresponding to the applied operator, and (iii) the set of excitations from $|n\rangle$, each characterized by a pole and its corresponding weight. This structure allows the `gfmatrix` class to simultaneously handle multiple initial states and operators, enabling on-the-fly evaluation of DCFs at arbitrary complex frequencies $z \in \mathbb{C}$. By avoiding precomputed grids and redundant storage this approach minimizes memory footprint and computational overhead. Through the use of `gfmatrix` EDIpack achieves efficient memory management, consistent storage of spectral data and high-performance post-processing.

### 3.6  Bath parametrization

The quantum impurity problem is characterized by the coupling between a local impurity and a surrounding bath. Following the structure of Eq. (5), the bath is parametrized by two components: the Hamiltonian matrices, $h^p_{\alpha\beta\sigma\sigma'}$, and the hybridization amplitudes, $V^p_{\alpha\beta\sigma\sigma'}$, for $p = 1, \ldots, N_{\text{bath}}$. Internally, the bath is represented by a dedicated object, `effective_bath`, defined in the `ED_VARS_GLOBAL` module. From the user's perspective, all parameters are consolidated into a rank-1 array of doubles, managed using a reverse communication strategy.

The bath topology, which defines the connectivity between the electronic levels assigned to the bath, is specified through the input variable `bath_type`. In EDIpack this variable can take one of four possible values: **normal**, **hybrid**, **replica**, and **general**.

`bath_type`=**normal.**  In this configuration, the bath consists of $N_{\text{bath}}$ electronic sites coupled to each impurity orbital, resulting in $N_{\text{b}} = N_\alpha N_{\text{bath}}$ bath levels and a total size of $N_{\text{s}} = (N_{\text{bath}} + 1)N_\alpha$. For `ed_mode`=**normal** the bath Hamiltonian is parametrized as diagonal matrices in both the orbital and spin spaces:

$$h^p_{\alpha\beta\sigma\sigma'} = \varepsilon_{\alpha\sigma}\delta_{\alpha\beta}\delta_{\sigma\sigma'},$$

where $\varepsilon_{\alpha\sigma}$ represents the on-site energies for each orbital and spin. If `ed_mode`=**superc**, anomalous amplitudes connecting bath levels with opposite spins must also be considered. In this case, the bath Hamiltonian gains an additional set of parameters $\Delta_\alpha^p \delta_{\alpha\beta}\delta_{\sigma\bar{\sigma}}$, where $\Delta_\alpha^p$ represents the orbital diagonal pairing amplitudes. This choice corresponds to a Nambu representation for each component, with the Hamiltonian matrices taking the form:

$$h^p_{\alpha\beta\sigma\sigma'} = \varepsilon_\alpha^p \delta_{\alpha\beta}\tau^z_{\sigma\sigma'} + \Delta_\alpha^p \delta_{\alpha\beta}\tau^x_{\sigma\sigma'}.$$

where $\tau^{i=0,x,y,z}$ are Pauli matrices. The hybridization amplitudes between the impurity and the bath levels are similarly structured. For all values of `ed_mode` these parameters are diagonal in both spin and orbital space: $V^p_{\alpha\beta\sigma\sigma'} = V^p_{\alpha\sigma}\delta_{\alpha\beta}\delta_{\sigma\sigma'}$. If `ed_mode`=**nonsu2** an additional set including terms describing spin-flip processes (as total magnetization is not conserved) should be included, $V^p_{\alpha\beta\sigma\sigma'} = W^p_\alpha \delta_{\alpha\beta}\delta_{\sigma\bar{\sigma}}$.

The resulting total number of parameters used to determine the bath is $N_{\text{param}} = 2N_\alpha N_{\text{bath}}$ for `ed_mode`=**normal** or $N_{\text{param}} = 3N_\alpha N_{\text{bath}}$ for the other two values of `ed_mode`.

`bath_type`=**hybrid.** The hybrid topology is the simplest generalization of the bath parametrization which captures the effects of locally coupled impurity orbitals. However, this setup typically requires a more challenging optimization process, especially with few available bath levels (see Sec. 3.12). The bath is formed by $N_{\text{bath}}$ sites coupled to all the impurity orbitals, so that there are $N_{\text{b}} \equiv N_{\text{bath}}$ bath levels and $N_{\text{s}} = N_{\text{bath}} + N_\alpha$. The parametrization of the bath Hamiltonian is diagonal in both orbital and spin spaces, i.e.

$$h^p_{\alpha\beta\sigma\sigma'} = \varepsilon_{\alpha\sigma}\delta_{\alpha\beta}\delta_{\sigma\sigma'}.$$

For `ed_mode`=**superc** additional anomalous components $\Delta_\alpha^p \delta_{\alpha\beta}\delta_{\sigma\bar{\sigma}}$ diagonal in the orbital basis are included.

The key difference compared to the previous case is that the hybridization matrix elements are generally off-diagonal in the orbital indices:

$$V^p_{\alpha\beta\sigma\sigma'} = V^p_{\alpha\beta\sigma}\delta_{\sigma\sigma'}.$$

If full spin symmetry is not preserved, i.e. for `ed_mode` = **nonsu2**, a further set of parameters must be specified to account for spin-flip processes: $V^p_{\alpha\beta\sigma\sigma'} = W^p_{\alpha\beta}\delta_{\sigma\bar{\sigma}}$.

The resulting total numbers of available bath parameters are $N_{\text{param}} = N_{\text{bath}} + N_\alpha N_{\text{bath}}$ for `ed_mode`=**normal**, $N_{\text{param}} = 2N_{\text{bath}} + N_\alpha N_{\text{bath}}$ if `ed_mode`=**superc** or $N_{\text{param}} = N_{\text{bath}} + 2N_\alpha N_{\text{bath}}$ for `ed_mode`=**nonsu2**.

`bath_type`=**replica, general.** This configuration provides a more flexible parametrization of the bath. The original core idea [115–117] is to assign each bath element a structure that *replicates* the internal structure of the impurity, while maintaining diagonal coupling between bath elements and the impurity. This offloads the complexity of representing structured quantum impurities to the bath Hamiltonian rather than the hybridizations.

From a broader perspective, this can be formalized by considering a user-defined matrix basis of dimension $N_{\text{sym}}$, $\vec{\Gamma} = \{\Gamma^\nu_{\alpha\beta\sigma\sigma'}\}_{\nu=1,\dots,N_{\text{sym}}}$ in the (Nambu-)spin-orbital space, and expressing the bath Hamiltonian as

$$h^p = \sum_{\nu=1}^{N_{\text{sym}}} \lambda_\nu^p \Gamma^\nu \equiv \vec{\lambda}^p \cdot \vec{\Gamma},$$

where $\vec{\lambda}^p \in \mathbb{R}^{N_{\text{sym}}}$ is a vector of variational parameters. The choice of the matrix basis can be inspired by symmetry group considerations [117, 118], by the internal structure of the

quantum impurity or determined case-by-case depending on the desired properties of the solution. In this scheme, the total number of bath levels is $N_{\rm b} = N_\alpha N_{\rm bath}$.

For the **replica** topology, the coupling between the impurity and each bath element is independent on spin, orbital, and internal bath structure: $V^p_{\alpha\beta\sigma\sigma'} = V^p$. In the **general** case, the coupling includes explicit diagonal dependence on the internal spin and orbital indices: $V^p_{\alpha\beta\sigma\sigma'} = V^p_{\alpha\sigma}\delta_{\alpha\beta}\delta_{\sigma\sigma'}$. The number of independent bath parameters to optimize is either $N_{\rm param} = (N_{\rm sym} + 1)N_{\rm bath}$ for `bath_type=`**replica** or $N_{\rm param} = (N_{\rm sym} + N_\alpha N_\sigma)N_{\rm bath}$ for `bath_type=`**general**.

We summarize the total number $N_s$ of electronic levels of the quantum impurity system according to the value of $N_\alpha$, $N_{\rm bath}$ and the bath topology in the following table:

| `bath_type` | normal | hybrid | replica | general |
|---|---|---|---|---|
| $N_s$ | $(N_{\rm bath} + 1)N_\alpha$ | $N_{\rm bath} + N_\alpha$ | $N_\alpha(N_{\rm bath} + 1)$ | $N_\alpha(N_{\rm bath} + 1)$ |

All the procedures concerning the bath are grouped into a set of modules wrapped by `ED_BATH`. We divided the set of modules into three categories, according to their scope.

**Bath Auxiliary Tools.** This group of modules provides user-oriented utilities, including standard symmetry operations on the bath array, e.g. orbital symmetry and particle-hole symmetry, available in `ED_BATH_USER`.

The critical functionality here is the determination of the total bath array dimension, implemented in `ED_BATH_DIM`. The function `ed_get_bath_dimension` calculates the required dimension $N_{\rm param}$ for user-allocated bath arrays, ensuring they contain exactly the necessary number of parameters.

The following table summarizes the total number of parameters $N_{\rm param}$, as returned by the function `ed_get_bath_dimension` and required to describe the bath according to the values of `ed_mode` and `bath_type`:

| $N_{\rm param}$ | normal | hybrid | replica | general |
|---|---|---|---|---|
| **normal** | $2N_\alpha N_{\rm bath}$ | $(N_\alpha + 1)N_{\rm bath}$ | $(N_{\rm sym} + 1)N_{\rm bath}$ | $(N_{\rm sym} + N_\alpha N_\sigma)N_{\rm bath}$ |
| **superc** | $3N_\alpha N_{\rm bath}$ | $(N_\alpha + 2)N_{\rm bath}$ | $(N_{\rm sym} + 1)N_{\rm bath}$ | $(N_{\rm sym} + N_\alpha N_\sigma)N_{\rm bath}$ |
| **nonsu2** | $3N_\alpha N_{\rm bath}$ | $(2N_\alpha + 1)N_{\rm bath}$ | $(N_{\rm sym} + 1)N_{\rm bath}$ | $(N_{\rm sym} + N_\alpha N_\sigma)N_{\rm bath}$ |

**Bath replica/general.** The module `ED_BATH_REPLICA` defines the class `Hreplica`/`Hgeneral` for the **replica**/**general** bath parametrization. In addition to standard object construction, destruction, reading and saving, this module provides the setup for either the matrix basis $\{\Gamma^\nu\}_{\nu=1,...,N_{\rm sym}}$ and the initial variational parameters $\vec{\lambda}$ through the function `set_Hreplica`/`set_Hgeneral`. The bath Hamiltonian is constructed with the function `build_Hreplica`/`build_Hgeneral` via the relation $h^p = \vec{\lambda}^p \cdot \vec{\Gamma}$.

**Effective Bath.** The module `ED_BATH_DMFT` implements the class `effective_bath` (defined in `ED_VARS_GLOBAL`), which manages all the actual bath parameters for any choice of `ed_mode` and `bath_type`. For the **replica**/**general** topologies, it directly references the `Hreplica` class. A global instance of this class, `dmft_bath`, is shared throughout the code, providing a centralized structure for bath parameter management.

The constructor for this class includes the function `init_dmft_bath`, which either initializes bath parameters from scratch or reads them from a file specified by the input variable `Hfile` as `<Hfile>.restart`. Additionally, the class offers methods to translate

bath parameters between the internal `dmft_bath` representation and the user-defined bath arrays via `get/set_dmft_bath`.

## 3.7 Lanczos based Diagonalization

In presence of symmetries (see Sec. 3.3) the matrix representing the Hamiltonian operator $\hat{H}$ can be reduced to a block-diagonal form. Each block corresponds to a symmetry sector with a fixed set of quantum numbers $\vec{Q}$ and is represented by a Hamiltonian matrix $H_{\mathcal{S}_{\vec{Q}}}$. Consequently, the analysis of the energy spectrum of the quantum impurity reduces to solving the secular equation within each individual symmetry sector:

$$H_{\mathcal{S}_{\vec{Q}}}|v\rangle = E_{\mathcal{S}_{\vec{Q}}}|v\rangle.$$

Even though the dimension of a given sector $\mathcal{S}_{\vec{Q}}$ is much smaller than the total Fock space dimension $D_{\mathcal{S}} \ll D_{\mathcal{F}}$, solving the corresponding eigenvalue problem completely remains a formidable challenge already for $N_s \simeq 8$. To outwit this, various algorithms exploiting the sparse nature of $H_{\mathcal{S}}$ have been developed, including Krylov subspace methods and related approaches [90–92, 119–121].

In EDIpack, the preferred algorithm is P-ARPACK, a state of the art Lanczos-based eigensolver with distributed memory support [91] ensuring the accurate convergence of the desired eigenpairs. However, it is also possible to select a simpler, though less efficient, parallel Lanczos algorithm using the input variable `lanc_method`.

The core computational task in any Krylov-based diagonalization algorithm is the Matrix-Vector Product (MVP), i.e. the application of the Hamiltonian to a given input vector:

$$H_{\mathcal{S}}|v\rangle \to |w\rangle \qquad |v\rangle, |w\rangle \in \mathcal{S}_{\vec{Q}}. \tag{9}$$

This operation amounts to more than 80% of the overall computational effort, making its optimization a critical aspect of high-performance diagonalization algorithms. In Ref. [82] we discussed in greater details the massively parallel strategies employed in EDIpack to optimize the MVP operations and presented various benchmarks of the parallel scaling. Here, we provide a brief outline of the main ideas following the notation introduced in Ref. [82], and focusing on the impact of the choice of `ed_mode`. For sake of clarity, here we focus on the purely electronic case, i.e. neglecting the electron-phonon part. The product structure of the electron and phonon Fock spaces ensure that the following approach applies with few algorithmic modifications in presence of electron-phonon coupling.

In the **normal** case, the tensor structure of the Fock space and the sector symmetries are fully leveraged. Within a given sector, the electronic part of the Hamiltonian can be expressed as:

$$H_{\mathcal{S}}^e = H_d + H_\uparrow \otimes \mathbb{1}_\downarrow + \mathbb{1}_\uparrow \otimes H_\downarrow + H_{nd}, \tag{10}$$

where $H_d$ represents the diagonal (local) part of the Hamiltonian, including density-density interactions, while $H_\sigma$ captures hopping processes for electrons with spin $\sigma = \uparrow, \downarrow$. The term $H_{nd}$ encompasses all remaining non-diagonal contributions, such as spin-exchange and pair-hopping terms. In this formulation, a sector state $|v\rangle$ can be represented in a matrix basis $\hat{v}$ with rows (columns) corresponding to $\uparrow$ ($\downarrow$) configurations. This structure allows for efficient MVP operations using `MPI_All2AllV`, which aims to optimize memory locality. However, terms in $H_{nd}$ that break this structure require `MPI_AllGatherV`, introducing a slight communication overhead which partially reduces the parallel efficiency.

For the **superc/nonsu2** modes, symmetries enforce specific constraints between $\uparrow$ and $\downarrow$ configurations, preventing the direct use of the factorized Fock space representation. The electronic part of the Hamiltonian is typically structured as:

$$H_{\mathcal{S}}^e = H_{imp} + H_{int} + H_{bath} + H_{imp-bath}, \tag{11}$$

where $H_{\text{imp}}$ encodes the local impurity Hamiltonian determined by $h^0_{\alpha\beta\sigma\sigma'}$, $H_{\text{int}}$ contains the interaction terms, $H_{\text{bath}}$ describes the effective bath contributions corresponding to $h^p_{\alpha\beta\sigma\sigma'}$, and $H_{\text{imp-bath}}$ accounts for the impurity-bath couplings $V^p_{\alpha\beta\sigma\sigma'}$. In an MPI setup, the first three terms contains also elements local to each node and stored separately within each `sparse_matrix` instance representing $H_\mathcal{S}$ (see Sec. 3.5.1). The MVP is carried out using the `MPI_AllGatherV` algorithm which, as noted Ref. [82], requires reconstruction of distributed arrays and incurs significant communication overhead.

The diagonalization process itself is divided into two primary phases:

**Global Setup.** This phase allocates the required memory, initializes the MPI environment, and configures the appropriate MVP procedure based on the symmetries of the problem. This is implemented in a set of independent modules, `ED_HAMILTONIAN_<ed_mode>`. Any module includes two functions: `build_Hv_sector_<ed_mode>` and `vecDim_Hv_sector_-<ed_mode>`.

The `build_Hv_sector_<ed_mode>` function builds the symmetry sector and allocates the MVP function by setting the shared abstract function pointer `spHtimesV_p` to the correct MVP function. This latter step is controlled by the value of `ed_sparse_H=T/F`. If `T`, the Hamiltonian $H_\mathcal{S}$ gets stored in a `sparse_matrix` instance and used to perform the MVPs. The corresponding algorithms are implemented in the modules `ED_HAMILTONIAN_<ed_mode>_STORED_HxV`. If `ed_sparse_H` is `F`, the MVP is operated on-the-fly, i.e. each element of $H_\mathcal{S}$ is directly applied to the input vector $|v\rangle$, either in serial or parallel mode as implemented in `ED_HAMILTONIAN_<ed_mode>_DIRECT_HxV`.

The function `vecDim_Hv_sector_<ed_mode>` returns the dimension of the vector used in the MVP. In an MPI-parallel execution the returned value is the dimension $d_i$ of the vector chunk per each node such that $\sum_{i=1}^{N_{\text{nodes}}} d_i = D_\mathcal{S}$. Yet, the specific value of $d_i$ depends on the MPI algorithm used for the MVP function.

**Diagonalization.** This step is managed by the `ED_DIAG_<ed_mode>` modules, which contain the main functions `diagonalize_impurity_<ed_mode>`. This phase includes: (i) selecting sectors for diagonalization, (ii) performing diagonalization within each sector, and (iii) analyzing the resulting `state_list` of conserved eigenstates.

## 3.8 Dynamical correlation functions

The determination of the low energy part of the Hamiltonian's spectrum enables evaluation of the Dynamical Correlation Functions (DCFs) using Krylov subspace algorithm. This capability is central to the library when using EDIpack as an impurity solver within the DMFT framework.

Before delving into the implementation specifics, we outline the generic approach. Consider the generic DCF:

$$C_\mathcal{A} = \langle \mathcal{T}_\pm [\mathcal{A}(t)\mathcal{A}^\dagger]\rangle, \tag{12}$$

where $\mathcal{A}(t) = e^{iHt}\mathcal{A}e^{-iHt}$, $\mathcal{T}_\pm$ is the time-ordering operator for fermions $(+)$ or bosons $(-)$, and $\langle\cdot\rangle = \frac{1}{Z}\text{Tr}\left[e^{-\beta H}\cdot\right]$, with $Z = \sum_n e^{-\beta E_n}$, the thermodynamic average. Using spectral decomposition, the expression Eq. (12) reduces to a Källén-Lehmann form:

$$
\begin{aligned}
C_\mathcal{A}(z) &= \langle \mathcal{A}\frac{1}{z-H}\mathcal{A}^\dagger\rangle \mp \langle\mathcal{A}^\dagger\frac{1}{z+H}\mathcal{A}\rangle \\
&\simeq \frac{1}{\tilde{Z}}\sum_{n=1}^{N_{\text{state\_list}}} e^{-\beta E_n}\sum_m \frac{|\langle\psi_m|\mathcal{A}^\dagger|\psi_n\rangle|^2}{z-(E_m-E_n)} \mp \frac{|\langle\psi_m|\mathcal{A}|\psi_n\rangle|^2}{z+(E_m-E_n)},
\end{aligned}
\tag{13}
$$

where $z \in \mathbb{C}$, $\tilde{Z} = \sum_{n=1}^{N_{\texttt{state\_list}}} e^{-\beta E_n}$ and $\{|\psi_n\rangle, E_n\}$ are the eigenpairs of the Hamiltonian $H$. This form is appealing but computationally prohibitive, as it requires the full spectrum of the Hamiltonian. However, the first line in Eq. (13) highlights that the $C_{\mathcal{A}}$ essentially corresponds to a specific matrix element of the resolvent operator $(z - H)^{-1}$, which can be efficiently approximated using the Krylov subspace method.

To illustrate this approach, consider the normalized initial state

$$|\phi_n\rangle = \mathcal{A}^\dagger |\psi_n\rangle / \mathcal{N}_n,$$

where $|\psi_n\rangle \in \mathcal{S}_{\vec{Q}}$ is an eigenstate of the sector Hamiltonian and $\mathcal{N}_n = \sqrt{\langle \psi_n | \mathcal{A} \mathcal{A}^\dagger | \psi_n \rangle}$. The Krylov basis is constructed by repeated applications of the sector Hamiltonian via MVP:

$$\mathcal{K}_N(|\phi_n\rangle) = \{|\phi_n\rangle, H|\phi_n\rangle, \dots, H^N |\phi_n\rangle\} \equiv \{|v_0^n\rangle, |v_1^n\rangle, \dots, |v_N^n\rangle\}$$

with $1 \ll N \ll \mathcal{D}_{\mathcal{S}}$. Given that any eigenstate $|\psi_n\rangle$ can be expressed within the Krylov basis as

$$|\psi_n\rangle = \sum_i \langle v_i^n | \psi_n \rangle | v_i^n \rangle = \sum_i a_i^n | v_i^n \rangle,$$

we can approximate the expression Eq. (13) as:

$$
\begin{aligned}
C_{\mathcal{A}}(z) &\simeq \frac{1}{\tilde{Z}} \sum_{n=1}^{N_{\texttt{state\_list}}} e^{-\beta E_n} \sum_{m=1}^{N} \frac{\langle \psi_n | \mathcal{A} \mathcal{A}^\dagger | \psi_n \rangle |a_m^n|^2}{z - (E_m - E_n)} \mp \frac{\langle \psi_n | \mathcal{A}^\dagger \mathcal{A} | \psi_n \rangle |a_m^n|^2}{z + (E_m - E_n)} \\
&= \frac{1}{\tilde{Z}} \sum_{n=1}^{N_{\texttt{state\_list}}} \sum_{m=1}^{N} \sum_{\nu=\pm} \frac{w_{mn}^\nu(\mathcal{A})}{z - dE_{mn}^\nu(\mathcal{A})} \equiv \frac{1}{\tilde{Z}} \sum_n \sum_{m=1}^{N} \sum_{\nu=\pm} g_{\mathcal{A}}(z; \nu, w_{mn}^\nu, dE_{mn}^\nu),
\end{aligned}
\tag{14}
$$

where the terms are grouped as a sum over spectral weights and poles. We introduced the notation $g(z; \nu, w_{mn}^\nu, dE_{mn}^\nu)$ for the `gfmatrix` object containing the weights $w_{mn}^\nu(\mathcal{A})$ and poles $dE_{mn}^\nu(\mathcal{A})$ for the operator $\mathcal{A}$, for every initial state $|\psi_n\rangle$ contributing to the low energy spectrum, for every order $m$ of the Krylov subspace algorithm and for channel $\nu$.

This method is limited to diagonal DCF like Eq. (12). In practice, many applications also require building off-diagonal functions of the form

$$C_{\mathcal{A}\mathcal{B}}(t) = \langle T_\pm [\mathcal{A}(t) \mathcal{B}^\dagger] \rangle .$$

This can be addressed by introducing auxiliary operators, e.g. $\mathcal{O} = \mathcal{A} + \mathcal{B}$ and $\mathcal{P} = \mathcal{A} - i\mathcal{B}$, enabling extraction of the desired function via simple algebraic combinations:

$$C_{\mathcal{A}\mathcal{B}} = \frac{1}{2} \left[ C_{\mathcal{O}} + C_{\mathcal{P}} - (1 - i)C_{\mathcal{A}} - (1 - i)C_{\mathcal{B}} \right]$$

In EDIpack, the computation of the impurity Green's functions

$$G_{\alpha\beta\sigma\sigma'}(t) = \langle \mathcal{T}_\pm [c_{\alpha\sigma}(t) c_{\beta\sigma'}^\dagger] \rangle$$

is handled by the module `ED_GREENS_FUNCTIONS`. It integrates more specialized methods based on the symmetry classification defined by the `ed_mode` parameter. Moreover, in the **normal** mode, EDIpack also includes spin, charge, pair, and excitonic susceptibility functions, providing a comprehensive framework for dynamical response calculations. From a computational perspective, the construction of the Krylov basis $\mathcal{K}_N(\mathcal{O}|\psi_n\rangle)$ for each eigenstate within the low-energy spectrum is typically the most resource-intensive step. As with the diagonalization process, a significant performance gain is achieved through the parallel execution of the MVP at the core of the Hamiltonian tri-diagonalization algorithm. The input variable `lanc_gfniter` regulates the maximum order of the Krylov basis, determining the upper limit on the number of excitations considered in Eq. (14). Operationally, each symmetry mode (`ed_mode`) requires a distinct strategy for Green's function construction, implemented in the corresponding `ED_GF_<ed_mode>` modules.

**normal.** In this mode, all orbital-dependent and spin-diagonal Green's functions $G_{\alpha\beta\sigma\sigma}$ must be evaluated. The diagonal case, where $\alpha = \beta$, is straightforward and involves applying the operator $\mathcal{A} = c_{\alpha\sigma}$ to any eigenstate $|\psi_n\rangle$ from the global `state_list` of the low-energy spectrum, which is represented by the `sparse_espace` object. This operation relies on the functions `apply_op_C/CDG` in `ED_AUX_FUNX`.

The sector Hamiltonian matrix in this mode is assumed to be real symmetric, which simplifies the evaluation of the off-diagonal terms by exploiting the following symmetry under orbital exchange $G_{\alpha\beta\sigma}(z) = G_{\beta\alpha\sigma}(z)$. In this case, the off-diagonal Green's function components can be obtained using just real-valued auxiliary operators such as $\mathcal{O} = c_{\alpha\sigma} + c_{\beta\sigma}$, along with the identity $G_{\alpha\beta\sigma\sigma} = \frac{1}{2}(C_\mathcal{O} - G_{\alpha\alpha\sigma\sigma} - G_{\beta\beta\sigma\sigma})$, where $C_\mathcal{O}$ denotes the DCF associated to $\mathcal{O}$ . This approach, avoiding the need for complex arithmetic, preserves the efficiency of real-valued computation.

The diagonal spin, charge and pair susceptibility terms $\chi_{\alpha\alpha}^{Sz}$, $\chi_{\alpha\alpha}^{N}$ and $\chi_{\alpha\alpha}^{\Delta}$ are constructed using the operators $S_\alpha^z = \sum_{\sigma\sigma'} c_{\alpha\sigma}^\dagger \tau_{\sigma\sigma'}^z c_{\alpha\sigma'}$, $N_\alpha = \sum_{\sigma\sigma'} c_{\alpha\sigma}^\dagger \tau_{\sigma\sigma'}^0 c_{\alpha\sigma'}$ and $\Delta_\alpha = c_{\alpha\downarrow} c_{\alpha\uparrow}$, where $\tau^{a=0,x,y,z}$ are the Pauli matrices. Off-diagonal terms are evaluated using the operators $(S_\alpha^z + S_\beta^z)$, $(N_\alpha + N_\beta)$ or $(\Delta_\alpha + \Delta_\beta)$. The excitonic susceptibilities, $\chi_{\alpha\beta}^{\vec{E}}$, are defined with respect to the vector operator $E_{\alpha\beta}^i = \sum_{\sigma\sigma'} c_{\alpha\sigma}^\dagger \tau_{\sigma\sigma'}^i c_{\beta\sigma'}$, where $i = 0$ represents the spin-singlet exciton [122], and $i = x, y, z$ correspond to the spin-triplet exciton [112].

**superc.** In the superconductive case, the orbital-dependent Nambu $s$-wave Green's function has the form:

$$\hat{G}_{\alpha\beta} = \begin{bmatrix} G_{\alpha\beta\uparrow\uparrow} & F_{\alpha\beta\uparrow\downarrow} \\ \bar{F}_{\alpha\beta\downarrow\uparrow} & \bar{G}_{\alpha\beta\downarrow\downarrow.} \end{bmatrix} \tag{15}$$

Exploiting symmetries between matrix components, it is sufficient to compute only the top row elements along with a few auxiliary functions [43,55–57,105,123]. The diagonal, normal, component $G_{\alpha\alpha\uparrow\uparrow}$ can be evaluated using the same approach as in the `ed_mode=normal` case. However, for the off-diagonal terms $G_{\alpha\beta\uparrow\uparrow}$, we forgo symmetry arguments and instead define two auxiliary operators, $\mathcal{O} = c_{\alpha\uparrow} + c_{\beta\uparrow}$ and $\mathcal{P} = c_{\alpha\uparrow} - i c_{\beta\uparrow}$. This allows us to express the Green's function as $G_{\alpha\beta\uparrow\uparrow} = \frac{1}{2}[C_\mathcal{O} + C_\mathcal{P} - (1 - i)(G_{\alpha\alpha\uparrow\uparrow} + G_{\beta\beta\uparrow\uparrow})]$.

Evaluating the diagonal and off-diagonal anomalous terms $F_{\alpha\beta\uparrow\downarrow}$ requires distinct combinations of creation and annihilation operators [43,55–57,105]. First, the component $\bar{G}_{\alpha\alpha\downarrow\downarrow}$ is evaluated as an auxiliary term using $\mathcal{A} = c_{\alpha\downarrow}^\dagger$. Then, we construct the two linear combinations

$$\mathcal{T} = c_{\alpha\uparrow} + c_{\beta\downarrow}^+,$$
$$\mathcal{R} = c_{\alpha\uparrow} - i c_{\beta\downarrow}^+,$$

which contribute to the auxiliary functions $C_\mathcal{T}$ and $C_\mathcal{R}$, respectively. The final expression for the anomalous function reads

$$F_{\alpha\beta\uparrow\downarrow} = \frac{1}{2}\left[ C_\mathcal{T} + C_\mathcal{R} - (1 - i)(G_{\alpha\alpha\uparrow\uparrow} + \bar{G}_{\beta\beta\downarrow\downarrow}) \right]. \tag{16}$$

**nonsu2.** In this case, all Green's function components must be explicitly evaluated, as the spin and orbital symmetries are in general broken. For the diagonal terms $G_{\alpha\alpha\sigma\sigma}$, the procedure follows the same approach outlined in the **normal** case.

The off-diagonal components $G_{\alpha\beta\sigma\sigma'}$, however, require a more general treatment. These are computed using auxiliary operators defined as:

$$\mathcal{O} = c_{\alpha\sigma} + c_{\beta\sigma'},$$
$$\mathcal{P} = c_{\alpha\sigma} - i c_{\beta\sigma'},$$

which allows the Green's function to be expressed as:

$$G_{\alpha\beta\sigma\sigma'} = \tfrac{1}{2}\left[C_{\mathcal{O}} + C_{\mathcal{P}} - (1-i)(G_{\alpha\alpha\sigma\sigma} + G_{\beta\beta\sigma'\sigma'})\right]. \tag{17}$$

This approach effectively reduces the complexity of evaluating the off-diagonal terms by leveraging auxiliary functions, despite the absence of full spin symmetry.

### 3.9 Observables

A wide range of predefined impurity observables and local static correlations, such as occupation numbers, total energy, pair amplitudes, and excitonic order parameters, are computed in the `OBSERVABLES` module. Similar to the previous cases, this module wraps different implementations depending on the operational mode specified by `ed_mode` and is distributed across the corresponding files `ED_OBSERVABLES_<ed_mode>`.

Local observables and correlations are generally defined through the thermal average $\langle\mathcal{O}\rangle = \frac{1}{Z}\text{Tr}\left[e^{-\beta H}\mathcal{O}\right]$, where $\mathcal{O}$ can be any local operator and $Z = \sum_n e^{-\beta E_n}$ is the partition function. At zero or low temperatures, this can be efficiently evaluated using the stored low-energy part of the spectrum, taking advantage of the exponential suppression provided by the Boltzmann factor:

$$\langle\mathcal{O}\rangle = \frac{1}{Z}\sum_{n=1}^{\infty} e^{-\beta E_n}\langle\psi_n|\mathcal{O}|\psi_n\rangle \simeq \frac{1}{\tilde{Z}}\sum_{n=1}^{N_{\text{state\_list}}} e^{-\beta E_n}\langle\psi_n|\mathcal{O}|\psi_n\rangle, \tag{18}$$

where $E_n$ and $|\psi_n\rangle$ are the $N_{\text{state\_list}}$ low-lying eigenstates of the system stored in the `state_list`, $Z = \sum_n e^{-\beta E_n}$ and $\tilde{Z} = \sum_{n=1}^{N_{\text{state\_list}}} e^{-\beta E_n}$.

### 3.10 Impurity reduced density matrix

The latest version of EDIpack introduces the calculation of the impurity Reduced Density Matrix (iRDM, $\rho^{\text{imp}}$), extending the algorithm initially proposed in Ref. [15, 124], to support the analysis of entanglement properties of quantum impurities for any value of `ed_mode`.

For sake of simplicity, this section focuses on the zero-temperature limit, assuming a non-degenerate ground state $|\psi\rangle$ present in the `state_list`. The generalization to the finite-temperature regime or degenerate ground states is straightforward, requiring only the introduction of an ensemble average $\langle\rho^{\text{imp}}\rangle$, as defined in Eq. (18).

A pure quantum state $|\psi\rangle$ belonging to a unique symmetry sector $\mathcal{S}_{\vec{Q}}$ can be represented in the Fock basis as: $|\psi\rangle = \sum_I a_I|I\rangle$, where the corresponding pure density matrix $\rho$ is given by:

$$\rho = |\psi\rangle\langle\psi| = \sum_{IJ=1}^{4^{N_s}} a_J^* a_I |I\rangle\langle J| = \sum_{IJ=1}^{4^{N_s}} \rho_{IJ}|I\rangle\langle J|. \tag{19}$$

The iRDM is obtained by tracing out the bath degrees of freedom:

$$\rho^{\text{imp}} = \text{Tr}_{\text{bath}}(\rho). \tag{20}$$

However, the memory footprint of $\rho$, as well as the CPU footprint of the summations involved in the trace, scale exponentially with the system size, i.e. as $4^{N_s}$, quickly becoming prohibitive. To overcome the issue, we implement a fast algorithm that exploits the block structure of the Fock space defined by the symmetry sectors and the subsequent map sparsity, to perform the trace *on-the-fly*. This approach significantly reduces the computational cost by limiting the summation to the size $D_{\mathcal{S}}$ of the symmetry sector $\mathcal{S}$, to which the ground state (or any eigenstate) belongs, and avoiding the storage of the

large matrix associated with $\rho$. We employ a `sparse_map` $\mathcal{P}$, as detailed in Sec. 3.5.2. This structure stores, for each impurity configuration (the *key*), the corresponding bath configurations (the *values*) that satisfy the symmetry constraints over the sector quantum numbers imposed by the `ed_mode` variable. We denote the number of (spin-dependent) keys and values respectively as $D_{i\sigma}$ and $D_{b\sigma}$. The fast summation algorithm for evaluating the iRDM differs significantly between the **normal** and **superc/nonsu2** modes, as outlined in the sections below.

**normal.** In this case, we take advantage of spin-resolved sector decomposition, as discussed in detail in Sec. 3.3. The global bitset can be split into spin-resolved impurity and bath components as

$$|\vec{n}\rangle = |\vec{n}_\uparrow\rangle \otimes |\vec{n}_\downarrow\rangle = |\vec{i}_\uparrow\rangle|\vec{b}_\uparrow\rangle \otimes |\vec{i}_\downarrow\rangle|\vec{b}_\downarrow\rangle. \tag{21}$$

Here, creation (destruction) operators $c_{p\sigma}^\dagger$ ($c_{p\sigma}^\dagger$) at position $p$ and for a given spin $\sigma$ act only on the corresponding spin subspace, i.e. they commute with the opposite spin subspace $|\vec{n}_{\bar{\sigma}}\rangle$ without introducing any fermionic sign. In this basis Eq. (19) takes the form:

$$\rho = \sum_{i_\uparrow=1}^{D_{i_\uparrow}} \sum_{p_\uparrow=1}^{D_{p_\uparrow}} \sum_{i_\downarrow=1}^{D_{i_\downarrow}} \sum_{p_\downarrow=1}^{D_{p_\downarrow}} \sum_{j_\uparrow=1}^{D_{j_\uparrow}} \sum_{q_\uparrow=1}^{D_{q_\uparrow}} \sum_{j_\downarrow=1}^{D_{j_\downarrow}} \sum_{q_\downarrow=1}^{D_{q_\downarrow}} a_{i_\uparrow p_\uparrow i_\downarrow p_\downarrow} a_{j_\uparrow q_\uparrow j_\downarrow q_\downarrow}^* |\vec{i}_\uparrow\rangle|\vec{p}_\uparrow\rangle \otimes |\vec{i}_\downarrow\rangle|\vec{p}_\downarrow\rangle \langle \vec{q}_\downarrow|\langle \vec{j}_\downarrow| \otimes \langle \vec{q}_\uparrow|\langle \vec{j}_\uparrow| \tag{22}$$

where the coefficients $a_I = a_{i_\uparrow b_\uparrow i_\downarrow b_\downarrow}$ are the expansion coefficients of the state on the Fock basis $|\psi\rangle = \sum_I a_I |i_\uparrow\rangle|b_\uparrow\rangle \otimes |i_\downarrow\rangle|b_\downarrow\rangle$. Before proceeding further, we show that in the **normal** mode the iRDM is restricted to spin-diagonal blocks. This follows from the observation that the conserved quantum numbers are *additive* quantities, i.e. $N_\sigma = N_\sigma^{\text{imp}} + N_\sigma^{\text{bath}}$, so that the corresponding generator of the symmetry group factorizes as $U_{N_\sigma} = U_{N_\sigma^{\text{imp}}} \otimes U_{N_\sigma^{\text{bath}}}$. Since the iRDM acts only on the impurity Fock space, this implies:

$$\begin{aligned}
\rho^{\text{imp}} &= U_{N_\sigma}^\dagger \rho^{\text{imp}} U_{N_\sigma} = U_{N_\sigma^{\text{bath}}}^\dagger \otimes U_{N_\sigma^{\text{imp}}}^\dagger \rho^{\text{imp}} U_{N_\sigma^{\text{imp}}} \otimes U_{N_\sigma^{\text{bath}}} \\
&= U_{N_\sigma^{\text{imp}}}^\dagger \rho^{\text{imp}} U_{N_\sigma^{\text{imp}}},
\end{aligned} \tag{23}$$

showing that $U_{N_\sigma^{\text{imp}}}$ defines a symmetry of $\rho^{\text{imp}}$, which is therefore block diagonal with respect to the impurity quantum numbers $N_\sigma^{\text{imp}}$.

Using Eq. (22) the iRDM takes the form:

$$\begin{aligned}
\rho^{\text{imp}} = \text{Tr}_{b_\uparrow b_\downarrow}\rho &= \sum_{b_\uparrow=1}^{D_{b_\uparrow}} \sum_{b_\downarrow=1}^{D_{b_\downarrow}} \langle b_\uparrow| \otimes \langle b_\downarrow|\rho|b_\downarrow\rangle \otimes |b_\uparrow\rangle \\
&= \sum_{b_\uparrow=1}^{D_{b_\uparrow}} \sum_{i_\uparrow=1}^{D_{i_\uparrow}} \sum_{p_\uparrow=1}^{D_{p_\uparrow}} \sum_{j_\uparrow=1}^{D_{j_\uparrow}} \sum_{q_\uparrow=1}^{D_{q_\uparrow}} \left( \sum_{b_\downarrow=1}^{D_{b_\downarrow}} \sum_{i_\downarrow=1}^{D_{i_\downarrow}} \sum_{p_\downarrow=1}^{D_{p_\downarrow}} \sum_{j_\downarrow=1}^{D_{j_\downarrow}} \sum_{q_\downarrow=1}^{D_{q_\downarrow}} C_{i,p}\, C_{j,q}\, a_{i_\uparrow p_\uparrow i_\downarrow p_\downarrow} a_{j_\uparrow q_\uparrow j_\downarrow q_\downarrow}^* \right. \\
&\qquad \left. \langle b_\uparrow|p_\uparrow\rangle|i_\uparrow\rangle\langle j_\uparrow|\langle q_\uparrow|b_\uparrow\rangle \otimes \langle b_\downarrow|p_\downarrow\rangle|i_\downarrow\rangle\langle j_\downarrow|\langle q_\downarrow|b_\downarrow\rangle \right) \\
&= \sum_{i_\uparrow=1}^{D_{i_\uparrow}} \sum_{j_\uparrow=1}^{D_{j_\uparrow}} \sum_{i_\downarrow=1}^{D_{i_\downarrow}} \sum_{j_\downarrow=1}^{D_{j_\downarrow}} \underbrace{\left( \sum_{b_\uparrow=1}^{D_{b_\uparrow}} \sum_{b_\downarrow=1}^{D_{b_\downarrow}} a_{i_\uparrow b_\uparrow i_\downarrow b_\downarrow} a_{j_\uparrow b_\uparrow j_\downarrow b_\downarrow}^* \right)}_{\rho_{i_\uparrow j_\uparrow}^{\text{imp}} \otimes \rho_{i_\downarrow j_\downarrow}^{\text{imp}}} |i_\uparrow\rangle\langle j_\uparrow| \otimes |i_\downarrow\rangle\langle j_\downarrow|,
\end{aligned} \tag{24}$$

Notably, the factors $C_{i,b}$, which account for the fermionic sign associated to swapping bath and impurity components, are trivially 1 due to the absence of cross-spin interference in this mode, as enforced by the Kronecker deltas for bath and impurity indices.

The numerical implementation relies on the use of the `sparse_map` ($\%sp$) for the ground state sector as reported in the listing below:

```fortran
do IimpUp=0,2**Norb-1
  do JimpUp=0,2**Norb-1
    !Finding the unique bath states connecting IimpUp and JimpUp
    call sp_return_intersection(sectorI%H(1)%sp,IimpUp,JimpUp,BATHup,lenBATHup)
    if(lenBATHup==0)cycle
    do IimpDw=0,2**Norb-1
      do JimpDw=0,2**Norb-1
        !Finding the unique bath states connecting IimpDw and JimpDw -> BATHdw(:)
        call sp_return_intersection(sectorI%H(2)%sp,IimpDw,JimpDw,BATHdw,lenBATHdw)
        if(lenBATHdw==0)cycle
        do ibUP=1,lenBATHup
          IbathUp = BATHup(ibUP)
          do ibDW=1,lenBATHdw
            IbathDw = BATHdw(ibDW)
            !Allowed spin Fock space Istates:
            !Iup = IimpUp +  2^Norb * IbathUp
            !Idw = IimpDw +  2^Norb * IbathDw
            iUP= binary_search(sectorI%H(1)%map,IimpUp + 2**Norb*IbathUp)
            iDW= binary_search(sectorI%H(2)%map,IimpDw + 2**Norb*IbathDw)
            i  = iUP + (iDW-1)*sectorI%DimUp
            !Allowed spin Fock space Jstates:
            !Jup = JimpUp +  2^Norb * IbathUp
            !Jdw = JimpDw +  2^Norb * IbathDw
            jUP= binary_search(sectorI%H(1)%map,JimpUp + 2**Norb*IbathUp)
            jDW= binary_search(sectorI%H(2)%map,JimpDw + 2**Norb*IbathDw)
            j  = jUP + (jDW-1)*sectorI%DimUp
            !
            io = (IimpUp + 2**Norb*IimpDw) + 1
            jo = (JimpUp + 2**Norb*JimpDw) + 1
            irdm(io,jo) = irdm (io,jo) + psi(i)*psi(j)*weight
          enddo
        enddo
      enddo
    enddo
  enddo
enddo
```

**superc/nonsu2.** For these lower-symmetry cases, the absence of a clean Fock space factorization introduces additional complexities. Specifically, the correct evaluation of the off-diagonal iRDM terms, which connect different $[N_\uparrow^{\mathrm{imp}}, N_\downarrow^{\mathrm{imp}}]$ blocks, requires an explicit computation of fermionic permutation signs.

Consider a generic Fock basis state in the form $|I\rangle = |i_\uparrow b_\uparrow i_\downarrow b_\downarrow\rangle$, and express the ground state as

$$|\psi\rangle = \sum_I a_I |I\rangle = \sum_{i_\uparrow=1}^{D_{i_\uparrow}} \sum_{b_\uparrow=1}^{D_{b_\uparrow}} \sum_{i_\downarrow=1}^{D_{i_\downarrow}} \sum_{b_\downarrow=1}^{D_{b_\downarrow}} a_{i_\uparrow b_\uparrow i_\downarrow b_\downarrow} |i_\uparrow b_\uparrow i_\downarrow b_\downarrow\rangle.$$

By tracing over the bath degrees of freedom of the corresponding pure density matrix $\rho$ we obtain:

$$
\begin{aligned}
\rho^{\mathrm{imp}} = \mathrm{Tr}_{b_\uparrow b_\downarrow} \rho &= \sum_{b_\sigma=1}^{D_{b_\sigma}} \langle b_\uparrow b_\downarrow | \rho | b_\downarrow b_\uparrow\rangle = \sum_{b_\sigma=1}^{D_{b_\sigma}} \langle b_\uparrow b_\downarrow|\psi\rangle\langle\psi|b_\downarrow b_\uparrow\rangle \\
&= \sum_{b_\sigma=1}^{D_{b_\sigma}} \sum_{i_\sigma=1}^{D_{i_\sigma}} \sum_{p_\sigma=1}^{D_{p_\sigma}} \sum_{j_\sigma=1}^{D_{j_\sigma}} \sum_{q_\sigma=1}^{D_{q_\sigma}} a_{i_\uparrow p_\uparrow i_\downarrow p_\downarrow} a^*_{j_\uparrow q_\uparrow j_\downarrow q_\downarrow} \langle b_\uparrow b_\downarrow|i_\uparrow p_\uparrow i_\downarrow p_\downarrow\rangle\langle q_\downarrow j_\downarrow q_\uparrow j_\uparrow|b_\downarrow b_\uparrow\rangle,
\end{aligned}
\tag{25}
$$

To further simplify this expression, it is essential to eliminate the sums over the internal bath indices $p_\sigma$ and $q_\sigma$, ideally contracting them with the outer bath indices $b_\sigma$. This requires reorganizing the bit representation of the basis states to bring the bath bitsets

before the impurity ones:

$$|i_\uparrow p_\uparrow i_\downarrow p_\downarrow\rangle \to C_{p_\uparrow,i_\downarrow}|i_\uparrow i_\downarrow p_\uparrow p_\downarrow\rangle,$$

where the fermionic sign factor $C_{p_\uparrow,i_\downarrow} = (-1)^{\#\vec{n}_{p\uparrow}\cdot\#\vec{n}_{i\downarrow}}$ accounts for the exchange of the bath $|p_\uparrow\rangle$ and impurity $|i_\downarrow\rangle$ bit configurations. Here, $\#\vec{n}_{\alpha\sigma}$ denotes the number of ones in the occupation vector $\vec{n}_{\alpha\sigma}$.

Substituting this relation back into the iRDM expression yields:

$$\rho^{\text{imp}} = \sum_{i_\uparrow=1}^{D_{i_\uparrow}}\sum_{j_\uparrow=1}^{D_{j_\uparrow}}\sum_{i_\downarrow=1}^{D_{i_\downarrow}}\sum_{j_\downarrow=1}^{D_{j_\downarrow}} \underbrace{\left(\sum_{b_\uparrow=1}^{D_{b_\uparrow}}\sum_{b_\downarrow=1}^{D_{b_\downarrow}} a_{i_\uparrow b_\uparrow i_\downarrow b_\downarrow}a^*_{j_\uparrow b_\uparrow j_\downarrow b_\downarrow}C_{b_\uparrow,i_\downarrow}C_{b_\uparrow,j_\downarrow}\right)}_{\rho^{\text{imp}}_{i_\uparrow i_\downarrow j_\downarrow j_\uparrow}} |i_\uparrow i_\downarrow\rangle\langle j_\downarrow j_\uparrow|, \qquad (26)$$

where the term $C_{b_\uparrow,i_\downarrow}C_{b_\uparrow,j_\downarrow}$ accounts for the fermionic sign associated with the bath-impurity bitset exchange.

In numerical implementations, this approach introduces a few key differences with respect to the **normal** mode. In particular, the ordering of the *key-value* pairs in the `sparse_map` must reflect the contiguous bitset structure, which is critical for correctly reconstructing the Fock state indices. See the following listing:

```fortran
do IimpUp=0,2**Norb-1
  do IimpDw=0,2**Norb-1
    do JimpUp=0,2**Norb-1
      do JimpDw=0,2**Norb-1
        !Build indices of the RDM in 1:4**Norb
        iImp = iImpUp + iImpDw*2**Norb
        jImp = jImpUp + jImpDw*2**Norb
        call sp_return_intersection(sectorI%H(1)%sp,iImp,jImp,Bath,lenBath)
        if(lenBATH==0)cycle
        do ib=1,lenBath
          iBath = Bath(ib)
          !Reconstruct the Fock state Ii map back to sector state i
          ii= iImpUp + iImpDw*2**Ns + iBath*2**Norb
          i = binary_search(sectorI%H(1)%map,ii)
          !Reconstruct the Fock state Jj map back to sector state j
          jj= jImpUp + jImpDw*2**Ns + iBath*2**Norb
          j = binary_search(sectorI%H(1)%map,jj)
          !Build the signs of each component of RDM(io,jo)
          nBup  = popcnt(Ibits(ii,Norb,Norb*Nbath))
          nIdw  = popcnt(Ibits(ii,Ns,Norb))
          nJdw  = popcnt(Ibits(jj,Ns,Norb))
          signI = (-1)**(nIdw*nBup)
          signJ = (-1)**(nJdw*nBup)
          sgn   = signI*signJ
          !
          io = (iImpUp+1) + 2**Norb*iImpDw
          jo = (jImpUp+1) + 2**Norb*jImpDw
          rdm(io,jo) = rdm(io,jo) + psi(i)*conjg(psi(j))*weight*sgn
        enddo
      enddo
    enddo
  enddo
enddo
```

## 3.11   Bath Functions

The module `ED_BATH_FUNCTIONS` implements the on-the-fly calculation of the hybridization function $\Delta_{\alpha\beta\sigma\sigma'}(z)$ and the non-interacting Green's functions $G^0_{\alpha\beta\sigma\sigma'}(z)$ for arbitrary complex frequencies $z \in \mathbb{C}$. These are defined as

$$\Delta_{\alpha\beta\sigma\sigma'}(z) = \sum_p \left[\hat{V}^p\left(z\mathbb{1} - \hat{h}^p\right)^{-1}\hat{V}^p\right]_{\alpha\beta\sigma\sigma'}, \qquad (27)$$

$$G^0_{\alpha\beta\sigma\sigma'}(z) = \left[(z+\mu)\mathbb{1} - \hat{h}^0 - \Delta(z)\right]^{-1}_{\alpha\beta\sigma\sigma'}, \tag{28}$$

where $V^p_{\alpha\beta\sigma\sigma'}$ and $h^p_{\alpha\beta\sigma\sigma'}$ are the bath coupling and bath Hamiltonian matrices, respectively, and $\mu$ is the chemical potential.

The module supports all cases defined by the `ed_mode` and `bath_type` variables, including both user-supplied baths (provided as rank-1 arrays of doubles) and the internally allocated `effective_bath` instance `dmft_bath`. This flexibility ensures compatibility with a wide range of other solvers.

Different routines are available for directly evaluating $\Delta$, $G^0$, and its inverse $[G^0]^{-1}$, which are critical for computing the self-energy functions on demand (see Sec. 3.13).

In the superconducting case, `ed_mode=`**superc**, special care is needed to properly handle the anomalous (off-diagonal) components within the Nambu basis. In this case, the bath functions include both the normal and anomalous components.

### 3.12 Bath Optimization

In the DMFT-ED framework, the bath parameters must be optimized to faithfully reproduce the Weiss field, $\mathcal{G}_0^{-1}{}_{\alpha\beta\sigma\sigma'}(z)$, or its corresponding hybridization function

$$\Theta(z) = (z+\mu)\mathbb{1} - H^{\text{loc}} - \mathcal{G}_0^{-1}(z), \tag{29}$$

where $\mu$ is the chemical potential and $H^{\text{loc}}$ is the local non-interacting Hamiltonian. The Weiss field is obtained from the DMFT self-consistency equation [7], while the bath discretization requires a careful fitting of this continuous function using a finite set of bath parameters (see Sec. 3.6).

Several algorithms have been proposed for this optimization step [125–127], each with strengths suited to different physical contexts. To maintain flexibility in EDIpack, we keep the optimization part independent from the core impurity solver. However, we also include a fully integrated optimization routine based on the conjugate gradient (CG) minimization of the cost function

$$\chi = \sum_{n=1}^{L_{\text{fit}}} \frac{1}{W_n} \left\| X(i\omega_n) - X^{\text{QIM}}(i\omega_n; \{V, h\}) \right\|_q, \tag{30}$$

where the $q$-norm is a suitably chosen distance metric in the matrix function space. Here, $X_{\alpha\beta} = \mathcal{G}_{0\alpha\beta\sigma\sigma'}$ or $\Theta_{\alpha\beta\sigma\sigma'}$ are the user-supplied local functions, while $X^{\text{QIM}}_{\alpha\beta} = G^0_{\alpha\beta\sigma\sigma'}$ or $\Delta_{\alpha\beta\sigma\sigma'}$ are the corresponding quantities for the quantum impurity model, see Sec. 3.11.

While different optimization methods have been developed for truncated algorithms, e.g. sCI [127] or DMRG [128], which handle systems with a larger number of bath levels, the CG minimization approach has proven both efficient and flexible for the small to moderate bath sizes typical in ED solvers.

The entire fit procedure is encapsulated in the function `ed_chi2_fitgf` provided by the module `ED_BATH_FIT`. To exploit the regularity of the bath functions, the fit is performed on the Matsubara frequency axis, where the functions are smooth and rapidly decaying. The form of $X_{\alpha\beta\sigma\sigma'}$ is controlled by the input parameter `cg_Scheme=Weiss,Delta`.

To provide maximum control over the fitting process, we include several tunable parameters:

- `cg_method=0,1` - Chooses the CG algorithm: `0` for a Fletcher-Reeves-Polak-Ribiere variant adapted from Numerical Recipes [129], and `1` for the original algorithm described in Ref. [7], commonly used in the DMFT community.

- `cg_grad=0,1` - Sets the gradient calculation method: `0` for analytical gradients (when available), `1` for numerical gradients (required if `cg_method=1`).

- `cg_Lfit` - Sets the number of Matsubara frequencies $L_{\text{fit}}$ used in the fit. This can be used to restrict the fit to the low-frequency regime, where the function behavior is most relevant.

- `cg_Weight=0,1,2` - Determines the frequency weighting scheme: `0` for uniform, `1` for inverse frequency weighting, and `2` for inverse Matsubara index weighting, which can emphasize low-energy contributions.

- `cg_pow` - Sets the power $q$ of the cost function, allowing the user to fine-tune the sensitivity of the optimization to outliers.

- `cg_norm=elemental,frobenius` - Sets the matrix norm to be used to define the cost function in Eq. (30). The default value (`elemental`) corresponds to a generic element-wise $\chi^q$ norm, defined as

$$||X - X^{\text{QIM}}||_q \equiv \sum_{ij} \left| X_{ij} - X_{ij}^{\text{QIM}} \right|^q.$$

  The `frobenius` option implements instead the matrix distance induced by the Frobenius inner product as

$$||X - X^{\text{QIM}}||_q \equiv \sqrt[q]{\text{Tr}\left[ (X - X^{\text{QIM}})^\dagger (X - X^{\text{QIM}}) \right]} = \left( \sum_{ij} \left| X_{ij} - X_{ij}^{\text{QIM}} \right|^2 \right)^{\frac{1}{q}}.$$

  For $q = 2$ (see `cg_pow`) the Frobenius norm is equivalent to the Euclidean distance in matrix space. We plan to expand the available options for the tt cg_norm parameter in future EDIpack updates, since the `nonsu2` and `superc` diagonalization modes involve subtle challenges in optimizing the off-diagonal components of X. These difficulties are related to the coexistence of matrix elements with different physical origin, so that they can live on different orders of magnitude, hence requiring carefully balanced optimization metrics to ensure an accurate bath representation.

- `cg_Ftol` - Controls the fit tolerance, setting the convergence threshold for the CG minimization.

- `cg_Niter` - Specifies the maximum number of allowed iterations for the CG minimization.

- `cg_stop=0,1,2` - Defines the exit condition of the minimization corresponding, respectively, to the options $C_1 \wedge C_2$, $C_1$ and $C_2$, with

$$C_1 : |\chi^{n-1} - \chi^n| < \texttt{cg\_Ftol}(1 + \chi^n),$$
$$C_2 : \|x_{n-1} - x_n\| < \texttt{cg\_Ftol}(1 + \|x_n\|),$$

  where $\chi^n$ is the cost function at the $n^{\text{th}}$ step, and $x_n$ the corresponding parameter vector. The stop condition is a logical OR between these criteria, with the parameter value selecting which condition to apply.

Together, these parameters provide precise control over the fitting process, allowing users to balance speed and accuracy according to their specific needs.

### 3.13  Input/Output

The `ED_IO` module provides comprehensive access to the results of the Lanczos diagonalization of the quantum impurity problem. Since each instance of the code persists in memory until a new calculation is initiated, access to the relevant data is managed through a set of dedicated functions. These include routines for extracting dynamical response functions, self-energy components, and impurity observables, as well as for performing on-the-fly recalculation of the impurity Green's functions and self-energies at arbitrary points in the complex frequency domain. The latter capability is enabled through the efficient use of `gfmatrix` objects for data storage.

A full list of available functions, including parameter descriptions and usage examples, can be found in the online documentation (see [edipack.github.io/EDIpack](edipack.github.io/EDIpack)). Here, we highlight two representative examples: The first is the function `ed_get_dens`, which populates the provided input array with the orbital impurity occupations $\langle n_{\alpha\sigma}\rangle$. This function is essential for extracting local density information, a key diagnostic in many DMFT studies.

The second example is `ed_get_sigma`, which retrieves the normal or anomalous components of the Matsubara or real-frequency self-energy function $\Sigma(z)$. This function represents a central output of any DMFT calculation, as the self-energy encapsulates the full set of local electronic correlations and their frequency dependence.

### 3.14  EDIpack2ineq: inequivalent impurities

In many contexts, it is necessary to solve systems with multiple, independent quantum impurities. This situation frequently arises in DMFT when modeling lattices with complex unit cells containing inequivalent atomic sites, or in supercell calculations where translational symmetry is broken, as in heterostructures, disordered systems, or multi-orbital setups.

Given that EDIpack allows for only a single instance of the solver at a time, a dedicated extension is required to handle these more complex cases. To address this, we developed the EDIpack2ineq sub-library, which extends the base functionality of EDIpack by managing memory and procedures for multiple impurities. This approach ensures that all functions remain accessible through standard Fortran interfaces, seamlessly integrating both EDIpack and EDIpack2ineq without compromising overall software design.

Below, we provide a brief overview of the main features introduced by EDIpack2ineq.

#### 3.14.1  Structure

The core of the EDIpack2ineq sub-library consists of several Fortran modules, all wrapped under the main interface module `EDIPACK2INEQ`. This module provides access to the full range of procedures and variables required to solve inequivalent quantum impurity problems. To use this extension, the user must include both the main EDIpack and the EDIpack2ineq modules, as shown below:

```fortran
program test
  !Load EDIpack library
  USE EDIPACK
  !Load the Inequivalent impurities extension
  USE EDIPACK2INEQ
  ...
```

A significant part of the EDIpack2ineq extension is the definition of global variables that extend the memory pool of EDIpack. This includes higher-rank arrays for storing impurity-specific data, ensuring that each impurity's state is maintained separately during the computation. While dedicated MPI communication could be used to manage specific objects in parallel, e.g. `effective_bath` or `gfmatrix`, we opted for a simpler and safer

file-based approach to maintain compatibility with the standard I/O procedures described in Sec. 3.13.

### 3.14.2 Core routines

The module `E2I_MAIN` wraps the key extensions to the main algorithms of EDIpack, including initialization, diagonalization, and finalization, while preserving the original function names for consistency.

`ed_init.` **Initialization** of the solver. This function extends the corresponding one in EDIpack to accept a rank-2 bath array, where the leading dimension specifies the number of inequivalent impurity problems. This ensures each bath is properly initialized for subsequent diagonalization.

`ed_solve.` **Diagonalization** of the inequivalent quantum impurity problems. This function is extended to accept a rank-2 bath array and to manage the diagonalization of each impurity problem. Parallel execution is controlled by the input flag `mpi_lanc=T/F`, which determines whether the Lanczos diagonalization is performed in parallel or sequentially across different impurities.

`ed_finalize.` **Finalization** of the solver. This handles the global memory release extending the corresponding function in EDIpack, which clears the memory pool for all inequivalent impurity instances.

### 3.14.3 Inequivalent Baths

The bath setup for multiple impurities is managed by the `E2I_BATH` module, which provides functions for defining site-specific bath matrices and their corresponding variational parameters. This includes support for conventional symmetry operations and the handling of replica bath structures.

In the module `E2I_BATH_REPLICA`, we extend the matrix basis definition for the variational parameters $\vec{\lambda}$, allowing for flexible bath optimization across multiple sites. Although the matrix basis is currently shared among all impurities, future versions may allow for fully independent bath parameterizations.

Additionally, the module `E2I_BATH_FIT` extends the generic function `ed_chi2_fitgf` to support simultaneous, independent bath optimization for all impurities using MPI, thereby improving the efficiency of large-scale calculations.

### 3.14.4 Input/Output

A key component of the EDIpack2ineq extension is the enhanced I/O capability for handling impurity-specific observables. The `E2I_IO` module includes a variety of functions for retrieving site-resolved quantities, such as local Green's functions, self-energies, and density matrices. These functions use the same naming conventions as the core EDIpack library, ensuring a consistent user experience across single and multi-impurity calculations.

## 4 Interoperability

The recent growing availability of state-of-the-art software dedicated to the solution of quantum impurity problems using different methods [63, 66, 67, 69, 127, 128, 130] poses a

challenge to test accuracy and reliability of the results. As such, software packages are expected to develop a higher level of interoperability, i.e. the capability to operate with other software, possibly written in different programming languages. Modern Fortran, which is the language of choice for EDIpack, since many years supports the standardized generation of procedures and variables that are interoperable with C. Here we describe the implementation of an interoperability layer aiming at developing APIs for other languages as well as integrating EDIpack in complex scientific frameworks, e.g. TRIQS [63] or w2dynamics [67].

## 4.1 C bindings

The interoperability with C language is provided by the `ISO_C_BINDING` module, which is part of the Fortran standard since 2003 [131, 132]. The module contains definitions of named constants, types and procedures for C interoperability. Alongside, a second key feature essential to expose any Fortran entity to C is the `BIND(C)` intrinsic function. In EDIpack we exploit these features of the language to provide a complete interface from the Fortran code to C/C++ named `EDIPACK_C`.

### 4.1.1 Installation and inclusion

The C-binding module is included in the build process of EDIpack and compiled into a dynamical library `libedipack_cbindings.so` (`.dylib`). As discussed in the Sec. 2.3, support for inequivalent impurities is configured at the build level and propagates to the C-bindings library as well. An exported variable `has_ineq` is defined and exported to C/C++ as a way to query the presence of support for the inequivalent impurities. The generated library and header files get installed in the include directory at the prefix location, specified during configuration step. The corresponding path is added to the environment variable `LD_LIBRARY_PATH`, valid of any Unix/Linux system, via any of the loading methods outlined in Sec. 2.4.

The C/C++ compatible functions and variables are declared in the header file `edipack_cbindings.h`. Since `ISO_C_BINDING` only provides, to date, C compatibility, the functions and variables are declared with C linkage, which prevents name mangling. Function overloading is also not supported, hence all interfaced Fortran functions, for example supporting multiple input variable combinations with different types and ranks, are here represented by multiple alternative functions.

As an example, we consider the `ed_chi2_fitgf` Fortran function, which handles the fitting of bath parameters. This is callable in C++ in the following variants:

- `chi2_fitgf_single_normal_n3`: for `ed_mode=normal/nonsu2` rank-3 Weiss field-/hybridization function arrays

- `chi2_fitgf_single_normal_n5`: for `ed_mode=normal/nonsu2` rank-5 Weiss field-/hybridization function arrays

- `chi2_fitgf_single_superc_n3`: for `ed_mode=superc` rank-3 Weiss field/hybridization function arrays

- `chi2_fitgf_single_superc_n5`: for `ed_mode=superc` rank-5 Weiss field/hybridization function arrays

Analogous functions for the inequivalent impurities case, i.e. EDIpack2Ineq, are available. All functions are listed and thoroughly documented in the online manual associated to this release. When using EDIpack functions in a C/C++ program, care must be taken

in the way arrays are passed. Consistently with `ISO_C_BINDING`, non-scalar parameters have to be passed as raw pointers. An array of integers containing the dimensions of the former need to be passed as well to allow for proper Fortran input parsing. A working C++ example is provided in the `examples` directory of the EDIpack repository and is discussed in Section 5.

### 4.1.2 Implementation

The interface layer is contained in the Fortran module `EDIPACK_C`. It contains a common part and two set of functions, one to interface the procedures from EDIpack and a second one to extend the interface to the inequivalent impurities case. Specifically, the implemented interface functions expose to C through `bind(C)` statement a number of procedures composing the Fortran API of EDIpack, i.e. contained in `ED_MAIN`. The procedures and shared variables can be divided in four main groups (note that contrary to the Fortran API the functions here lose the prefix `ed_`):

**Variables.**    A number of relevant input and shared variables, which are normally required to setup or to control the calculation, are interfaced to C directly in the EDIpack modules `ED_INPUT_VARIABLES` using the `bind(C)` constructs. These are implicitly loaded into the C-binding module `EDIPACK_C` through the Fortran `USE EDIPACK` statement and then further interfaced in the C++ header file.

**Main.**    This group contains interface to the exact diagonalization method, interfacing the solver `solve_site/ineq` as well as its initialization `init_solver_site/ineq` and finalization `finalize_solver` procedures. It also includes the functions used to set the non-interacting part of the impurity Hamiltonian $h^0_{\alpha\beta\sigma\sigma'}$ through the functions `set_Hloc`, or the interaction Hamiltonian through `add_twobody_operator`.

**Bath.**    In this group we implement a number of procedures dealing with bath initialization, symmetry operations and optimization. In particular it contains the function `get_bath_dimension` returning the dimension of the bath array on the user side as well as the setup of the matrix basis $\vec{\Gamma}$ for the replica (general) bath via different instances of `set_Hreplica(general)_{site,lattice}_d{3,5}`. A crucial part of the DMFT self-consistency loop is the optimization of the bath through CG algorithm (see Sec. 3.12). To this end we interfaced a number of functions which cover all the cases supported in EDIpack, conveniently called `chi2_fitgf_{single,lattice}_{normal,superc}_n{3,4,5,6}`. Note that, because the actual optimization is still performed through the Fortran code, no changes apply to the outcome of this step.

**Input/Output.**    The input and output parts of the software are interfaced in this group of functions. In particular, `read_input` exposes to C the input reading procedure of EDIpack, which sets all the internal variables of EDIpack. Next, in `edipack{2ineq}_c_binding_io` we interface all the functions implementing the communication from the EDIpack instance to the user, namely those to retrieve static observables (e.g. `get_dens`), the impurity Green's functions and self-energies (e.g. `get_Sigma_{site,lattice}_n{3,5}`), the impurity susceptibilities (e.g. `get_spinChi`), the impurity reduced density matrix as well as the non-interacting Green's and hybridization functions starting from the user bath array.

## 4.2 EDIpack2py, the Python API

As a first application of the EDIpack C-bindings we implemented a complete Python interface called EDIpack2py. This is a Python module which enables access to all the library features and unlocks implementation of further interfaces of EDIpack as a plug-in solver for external Python-based software. Detailed documentation can be found online at edipack.github.io/EDIpack2py.

### 4.2.1 Installation

**From source.** EDIpack2py is available as a platform-agnostic Python module depending on EDIpack. The code can be obtained and and installed from source as:

```
git clone https://github.com/edipack/EDIpack2py EDIpack2py
cd EDIpack2py
pip install .
```

**PyPi.** The EDIpack2py package is also available in PyPi at pypi.org/EDIpack2py. The package can be easily installed in any system supporting `pip` as:

```
pip install edipack2py
```

**Anaconda.** Similarly to EDIpack, the Python API EDIpack2py is available as an Anaconda package for GNU/Linux and macOS systems. Packages are available for Python$\geq$ 3.10. The EDIpack package contains the `EDIpack2py` Python module as well as EDIpack and SciFortran libraries. Using Conda or Mamba the installation proceeds as following:

```
conda create -n myenv    #create a virtual environment called "myenv"
conda activate myenv     #activate it
conda install -c conda-forge -c edipack edipack #install edipack
```

When the `EDIpack2py` module is imported, it attempts to load the dynamic library `libedipack_cbindings.so` (`.dylib`) containing the Fortran-C bindings for EDIpack. By default the library search proceeds as follow:

1. The user can override the location of the library (determined during the EDIpack build configuration) by exporting an environment variable called `EDIPACK_PATH`.

2. By default, the Python module detects the location of the Fortran libraries via `pkg-config`. Any of the loading methods outlined in Sec. 2.4 automatically pushes the correct configuration to `PKG_CONFIG_PATH`.

3. As a last resort, the environment variables `LD_LIBRARY_PATH` and `DYLD_LIBRARY_PATH` are analyzed to retrieve the correct location.

If none of the previous attempts succeeds, the module will not load correctly and an error message will be printed.

### 4.2.2 Implementation

The Python API provided in the EDIpack2py module consists essentially of a class called for convenience `global_env`. This class contains all the global variables inherited from the EDIpack C-bindings library, and implements a number of interface functions leveraging the Python duck typing to EDIpack. The variables and the functions of EDIpack are exposed to the user and are accessed as properties and methods of the `global_env` class.

The `global_env` class needs to be imported at the beginning of the Python script, along with other useful modules. Numpy is necessary, while mpi4py is strongly recommended.

```python
import numpy as np
import mpi4py
from mpi4py import MPI
from edipack2py import global_env as ed
import os,sys
```

The EDIpack2py module supports the solution of problems with independent impurities, interfacing to the EDIpack2ineq extension of the library, if present. Should the inequivalent impurities package not be built, the Python module silently disables the support to it, so that invoking any related procedure will result in a `RuntimeError`. The user can check the availability of the inequivalent impurities interface by querying the value of `edipack2py.global_env.has_ineq`.

The implementation of the Python API is divided into two main parts. The first is a set of global variables, the second includes 4 groups of functions: solver, bath, input/output, auxiliary.

**Global variables.**   This includes a subset of the input variables available in EDIpack which are used to control the calculation. The variables are loaded globally in `EDIpack2py` and can be accessed or set locally as properties of the class `global_env`. The global variables are initialized, alongside the remaining default input variables, through a call to procedure `edipack2py.global_env.read_input` which interface the `ed_read_input` function in EDIpack. A given example is reported in the following code extract:

```python
import numpy as np
from edipack2py import global_env as ed
ed.Nspin = 1              # set a global variable
mylocalvar = ed.Nspin    # assign to a local variable
print(ed.Nspin)          # all functions can have global variables as arguments
np.arange(ed.Nspin)      # array of integers from 0 to Nspin-1
```

**Solver functions.**   This group includes a number of functions enabling initialization, execution and finalization of the EDIpack solver.

- `init_solver` and `set_Hloc`. The first initializes the EDIpack environment for the quantum impurity problem solution, sets the effective bath either reading it from a file or initializing it from a flat band model. Once this function is called, it is not possible to allocate a second instance of the solver. `set_Hloc` sets the non-interacting part of the impurity Hamiltonian $h^0_{\alpha\beta\sigma\sigma'}$. Either function takes different argument combinations there including support for inequivalent impurities.

- `solve` This function solves the quantum impurity problem, calculates the observables and any dynamical correlation function. All results remain stored in the memory and can be accessed through input/output functions.

- `finalize_solver` This function cleans up the EDIpack environment, frees the memory deallocating all relevant arrays and data structures. A call to this functions enables a new initialization of the solver, i.e. a new call to `init_solver`.

**Bath functions.**   This set covers the implementation of utility functions handling the effective bath on the user side as well as interfaces to specific EDIpack procedures, either setting bath properties or applying conventional symmetry transformation. Here we discuss a pair of crucial functions in this group.

- `bath_inspect` This function translates between the user-accessible continuous bath array and the bath components (energy levels, hybridization and so on). It functions in both ways, given the array returns the components and vice-versa. It autonomously determines the type of bath and ED mode.

- `chi2_fitgf` This function fits the Weiss field or hybridization function ($\Delta$) with a discrete set of levels. The fit parameters are the bath parameters contained in the user-accessible array. Depending on the type of system we are considering (normal, superconductive, non-SU(2) symmetric) a different set of inputs has to be passed. The specifics of the numerical fitting routines are controlled in the input file.

Additionally, the group includes the function `get_bath_dimension`, returning the correct dimension for the user bath array to be allocated and `set_Hreplica/general`, which sets the matrix basis $\vec{\Gamma}$ and initializes the bath variational parameters $\vec{\lambda}$ for `bath_type=replica,general`.

**Input/Output functions.** This group includes functions that return to the userspace observables or dynamical correlation functions evaluated in EDIpack and stored in the corresponding memory instance. Each function provides a general interface, which encompasses all dimension of the input array there including inequivalent impurities support. For example, the function `get_sigma` returns the self-energy function array (evaluated on-the-fly) for a specified supported shape, normal or anomalous type and on a specific axis or frequency domain.

**Auxiliary functions.** This group includes some auxiliary functions, either interfacing EDIpack procedures or defined locally in Python to provide specific new functionalities. Among the latter we include `get_ed_mode`, which returns an integer index depending on the value of the variable `ed_mode=normal,superc,nosu2`, and `get_bath_type`, which works similarly for `bath_type`.

## 4.3 EDIpack2TRIQS: the TRIQS interface

A thin compatibility layer between EDIpack and TRIQS [63], i.e. Toolbox for Research on Interacting Quantum Systems, called EDIpack2TRIQS is available as a stand-alone project. This is a pure Python package built upon EDIpack2py (see Sec. 4.2) that provides a limited object-oriented interface to the most important features of EDIpack.

EDIpack2TRIQS strives to offer seamless interoperability with program tools based on TRIQS by adopting data types, conventions and usage patterns common to other TRIQS-based impurity solvers, such as TRIQS/CTHYB [66]. It also enables execution of EDIpack calculations in the MPI parallel mode with no extra effort from the user. Detailed documentation of the package can be found online at krivenko.github.io/edipack2triqs.

### 4.3.1 Installation

The package depends on EDIpack, and its dependencies therein, EDIpack2py and TRIQS version 3.1 or newer. Assuming the three prerequisites are correctly installed in the system, the current development version of EDIpack2TRIQS can be installed with `pip` from its GitHub repository as follows:

```
git clone https://github.com/krivenko/edipack2triqs
cd edipack2triqs
pip install .
```

**Anaconda.** Another option for installing the package is by using the Anaconda package manager on Unix/Linux systems. The following commands will create a new `conda` environment named 'edipack' and install the most recently released version of EDIpack2TRIQS along with its dependencies (EDIpack, EDIpack2py and the TRIQS libraries).

```
conda create -n edipack
conda activate edipack
conda install -c conda-forge -c edipack edipack2triqs
```

### 4.3.2 Implementation

The programming interface of EDIpack2TRIQS is built around the singleton Python class `EDIpackSolver`, defined in the module `edipack2triqs.solver`, whose instance represents the internal state of the EDIpack library and exposes its functionality through a number of attributes and methods.

**Constructor.** The constructor of `EDIpackSolver` accepts the Hamiltonian to be diagonalized in the form of a second-quantized fermionic operator, the TRIQS `Operator` object described in Sec. 8.7 of Ref. [63] (impurity problems involving bosons are not supported in the current version). In addition to the Hamiltonian, four *fundamental operator sets* `fops_imp_up`, `fops_imp_dn`, `fops_bath_up` and `fops_bath_dn` must be provided. Each of the sets contains pairs of labels `(b,j)` carried by the operators $c_{b,j}^\dagger/c_{b,j}$ corresponding to either impurity or bath electronic degrees of freedom with a certain spin projection. Having these two crucial pieces of information, the constructor initializes the underlying Fortran library and automatically selects the appropriate `ed_mode` (see Sec. 3.3) and `bath_type` (see Sec. 3.6). In addition, the constructor accepts a vast array of keyword arguments that allow for fine-tuning of the diagonalization process. Among others, these include the ED algorithm selection, the quantum numbers to be used, parameters of the Krylov space, the spectrum cutoff and various tolerance levels.

**Method `solve()`.** The method `solve()` calls `edipack2py.global_env.solve()` and therefore performs the bulk of calculations. It accepts the inverse temperature $\beta$ required to calculate the expectation value of physical observables, and parameters of energy grids for Green's function calculations.

**Input parameters.** It is possible to read off and change parameters of the Hamiltonian between successive calls to `solve()` via respective attributes of `EDIpackSolver`. Note, however, that the changes that would necessitate a change of `ed_mode` or `bath_type` are disallowed. The relevant attributes are the following.

- `nspin` — the number of non-equivalent spin projections (read-only).

- `norb` — the number of impurity orbitals (read-only).

- `hloc` — matrix $h^0_{\alpha\beta\sigma\sigma'}$ of the quadratic impurity Hamiltonian Eq. (1).

- `U` — 8-dimensional array $U_{\alpha\sigma_1,\beta\sigma_2,\gamma\sigma_3,\delta\sigma_4}$ of two-particle interactions as defined in Eq. (3).

- `bath` — an object representing the bath. Depending on the bath type selected upon solver object construction, this object is an instance of `BathNormal`, `BathHybrid` or `BathGeneral` (all three classes are members of the module `EDIpackSolver.bath`). The bath objects support basic arithmetic operations so that a mixing scheme within

a DMFT calculation can be easily implemented. The way to access individual bath parameters is specific to each of the three classes, and is described in detail in the online API reference.

**Calculation results.** After a successful invocation of `solve()`, one can extract results of the calculation from the following attributes.

- `e_pot`, `e_kin` — thermal average of the potential (interaction) and kinetic energy respectively.

- `densities`, `double_occ` — lists of average densities and double occupancies of impurity orbitals.

- `magnetization` — Cartesian components of the average impurity magnetization vectors, one row per orbital.

- `superconductive_phi` — Matrix of impurity superconductive order parameters $\phi_{\alpha\beta} = \langle c_{\alpha\uparrow} c_{\beta\downarrow} \rangle$ in orbital space.

- `g_iw`, `g_an_iw` — Normal and anomalous components of the Matsubara impurity Green's function.

- `Sigma_iw`, `Sigma_an_iw` — Normal and anomalous components of the Matsubara impurity self-energy.

- `g_w`, `g_an_w` — Normal and anomalous components of the real-frequency impurity Green's function.

- `Sigma_w`, `Sigma_an_w` — Normal and anomalous components of the real-frequency impurity self-energy.

The Green's functions are returned as TRIQS `BlockGf` containers with names of the individual blocks determined from the block labels `'b'` found in `fops_imp_up`, `fops_imp_dn` and from solver's `ed_mode`. The Matsubara and real-frequency meshes of the TRIQS GF containers (see Sec. 8.2 of Ref. [63]) are constructed according to the parameters passed to `solve()`. The anomalous components of the Green's functions and self-energies are only available when anomalous bath terms are present in the Hamiltonian. If this is not the case, an attempt to access these attributes results in a `RuntimeError`.

**Bath parameter fitting.** The method `EDIpackSolver.chi2_fit_bath()` is essentially a wrapper around `edipack2py.global_env.chi2_fitgf()`. It accepts the function to fit (either the hybridization function or the Weiss field) in the `BlockGf` format, and returns the parameter fit result as a `Bath*` object along with a `BlockGf` representation of the fitted function. In the superconducting case (`ed_mode=superc`), this method accepts and returns pairs of the `BlockGf` containers corresponding to the normal and anomalous components of the quantities in question.

## 4.4 w2dynamics interface

EDIpack is supported as an alternative impurity solver in the w2dynamics DMFT package [67], available on github.com/w2dynamics/w2dynamics. This integration enables users to seamlessly switch from the default hybridization expansion continuous-time QMC (CT-HYB), included natively in w2dynamics, to the EDIpack ED solver. This requires no changes to the input file and at most minor adjustments to configuration parameters.

### 4.4.1  Installation

Use of the interface requires a current version of w2dynamics ($\geq 1.1.6$) and a working installation of EDIpack2py (see Sec. 4.2). Users may build and optionally install w2dynamics using conventional CMake based source installation:

```
git clone https://github.com/w2dynamics/w2dynamics.git
cd w2dynamics
mkdir build
cd build
cmake ..
make install
```

This enables the default CT-HYB solver. For DMFT calculations using the EDIpack interface, it is sufficient that the `EDIpack2py` module is available for import at runtime, and no compilation is required on the w2dynamics side.

### 4.4.2  Implementation

The EDIpack interface is implemented as the class `EDIpackSolver` in the Python module `w2dyn.dmft.edipack_solver`. It is a subclass of `ImpuritySolver` and can be used as a drop-in replacement for `CtHybSolver`. The w2dynamics DMFT solver provides the hybridization function and local Hamiltonian for the auxiliary impurity problem via an `ImpurityProblem` instance passed to `set_problem`. It then it invokes the `solve` method to obtain the results, including, e.g., the Green's function, as an `ImpurityResult`. The `solve` method sets up the calculation using EDIpack by calling `EDIpack2py`, writing input files to a subdirectory, running EDIpack, and processing both the return values of `EDIpack2py` methods and the output files. It then formats the results following the w2dynamics conventions.

This implementation allows w2dynamics to abstract over the specific choice of impurity solver as much as possible. EDIpack is only called to solve individual impurity problems, while the standard w2dynamics code handles higher-level tasks such as the DMFT loop, support for multiple inequivalent impurities, and the user-facing interface via standard w2dynamics input files and output in its usual HDF5 format.

As a result, features that require explicit support from w2dynamics but are not yet implemented cannot be used through the EDIpack interface. In particular, solving impurity problems in the superconducting phase (`ed_mode=superc`) is not yet supported.

### 4.4.3  Configuration and Usage

w2dynamics reads its configuration parameters from a file named `Parameters.in` by default. To use EDIpack methods, the `solver` parameter in the section `General` needs to be set as follows:

```
[General]
solver = EDIPACK
```

Additional parameters specific to EDIpack are defined in the accessory section `[EDIPACK]`. These set many of the EDIpack's corresponding input variables (in uppercase letters). For example, the number of bath sites `NBATH` can be configured as:

```
[EDIPACK]
NBATH = 7
```

This section includes also the options to control the bath optimization and diagonalization algorithms. Other input variables that define the model (e.g., `NORB`, `ULOC`, and `BETA`) or relate to functionalities handled by w2dynamics itself (e.g., `NLOOP`) must be

provided through standard parameters in other sections, such as `[General]` or `[Atoms]`. An example of a complete properly formatted input file is provided in Sec. 5.1.1. The full set of configuration parameters can be found in the `configspec` file, located in the `w2dyn/auxiliaries` of the w2dynamics repository, see [67] for further details.

A calculation can be launched by running the w2dynamics DMFT program `DMFT.py`, which supports parallel execution via MPI to take advantage of EDIpack's parallelization.

### 4.4.4   Output and Results

Calculation results are stored in an HDF5 file, grouped by DMFT iteration (with group names formatted as `dmft-001`) and inequivalent impurity (for impurity-specific data, using group names like `ineq-001`). The content of this file can be accessed using the `hgrep` script provided with w2dynamics or other HDF5 tools. Stored quantities include, for example, the impurity self-energy (in datasets `siw-full`) and Green's function (in `giw-full`) on the Matsubara frequency axis, as well as single and double occupations (in `occ`). Additionally, the interface provides access to results not available when using the CT-HYB solver, such as the impurity self-energy (`somega`) and Green's function (`gomega`) on the real frequency axis. Standard EDIpack output files for each individual ED impurity solution are also available in subdirectories named like the the main HDF5 output file with iteration and impurity numbers appended.

### 4.5   EDIpack2jl, the Julia API

The C-binding approach is extremely handy, in that it opens the way for interoperability with a large number of languages and frameworks. As a further significant example, an **experimental** Julia API is provided for EDIpack. Although this is at the present a proof of concept, it is capable of replicating the results obtained with Fortran, C++ and Python implementations for the Bethe lattice example driver (see Sec. 5). Its structure and operation mimic that of the EDIpack2py layer, with minimal language-specific differences.

The EDIpack Julia API consists of a module called `EDIpack2jl`, which provides access to the global variables and functions contained in `libedipack_cbindings.so (.dylib)`. The library is searched upon loading the module, with the following order of priority:

- `EDIPACK_PATH`: if this environment variable is set, `EDIpack2jl` will look there for the library first

- `LD_LIBRARY_PATH`: for Linux systems

- `DYLD_LIBRARY_PATH`: for macOS systems

### 4.5.1   Structure

In partial analogy to the Python API, the global variables, but not the functions, are contained in a `struct` called `global_env`.

Similarly to the C++ case, global variables and functions are accessed as raw pointers. As a consequence, it is important that the dimensions of the array-like variables (such as `ULOC`) and, in general, the amount of memory occupied by each variable are correctly accounted for. This is achieved within the Julia module by appropriately casting the variables to compatible types, such as `Cint, Cbool, Cdouble`. Functions are called from the dynamic library, making use of the `ccall` procedure. As previously stated, as a consequence of the C linking conventions multiple alternative version of the interfaced Fortran procedures are present, to account for the different input variable combinations;

the EDIpack2jl wrapper functions take care of selecting the appropriate Fortran procedure depending on the set of input parameters provided by the user.

### 4.5.2 Installation and usage

At present, the EDIpack Julia interface is not offered as a package. The git repository has to be cloned via

```
git clone https://github.com/EDIpack/EDIpack2jl.git
```

and the location of the source files needs to be included in the user program via

```
push!(LOAD_PATH, joinpath(@__DIR__, "PATH/TO/REPO/src"))
```

The EDIpack2jl module can then be loaded via

```
using EDIpack2jl
```

The correct way to access global variables and functions is as in the following example:

```
EDIpack2jl.read_input("inputED.conf")
ed = EDIpack2jl.global_env
println("Nspin = ", ed.Nspin)
ed.Nspin = 2
```

The names and inputs of the Julia-wrapped functions are entirely analogous to those of the Python API.

We provide an example script for the simple case of the Bethe lattice in the normal phase in the `examples` folder of the EDIpack2jl repository. This script is intended to be run serially. Documentation of the API and further examples are being developed.

## 5 Examples

In this section we illustrate and benchmark the functionalities and of EDIpack library and its interfaces as a solver for DMFT through a variety of examples. We also discuss in detail the relevant code parts from different programming languages. The codes, the data and the scripts to rework some of the examples presented here, as well as additional upcoming tutorials, can be found in the repository github.com/EDIpack/EDIpack2examples.

Some tasks—particularly those associated with the implementation of the DMFT self-consistency—are inherently independent of EDIpack itself, and the impurity solver is agnostic about them. Thus, in the implementations using Fortran, Python or other EDIpack interfaces to scientific toolboxes, we adopt a reverse communication strategy, which requires the user to independently carry out these parts or, equivalently, to take advantage of existing utilities.

In our examples based on the Fortran APIs, we rely on external open-source libraries, such as DMFTtools (see github.com/aamaricci/DMFTtools), to perform specific tasks, including evaluating the local Green's function, implementing the self-consistency, constructing the tight-binding model, or calculating the kinetic energy. Where appropriate, we include comments that reference these external procedures.

On the other hand, the results obtained with w2dynamics rely on the internal framework provided by the software itself – see the script `DMFT.py` [67], which seamlessly handles all these tasks. Similarly, the TRIQS library offers a comprehensive framework for manipulating Green's functions and related quantities [63], making these tasks easily accessible while maintaining very high standards of optimization and accuracy.

## 5.1 Hubbard model on the Bethe lattice (Fortran/C++ API, `ed_mode=normal`)

The description of the Mott transition using the Hubbard model on the Bethe lattice is conventionally regarded as the standard test bed for any DMFT implementation. [7, 39–41, 44]. Here, we present a guided implementation of the DMFT solution for the Bethe lattice at $T = 0$ using EDIpack as impurity solver. The model under consideration is the Fermi-Hubbard Hamiltonian:

$$H = -t \sum_{\langle ij \rangle, \sigma} c_{i\sigma}^\dagger c_{j\sigma} + U \sum_i n_{i\uparrow} n_{i\downarrow},$$

where $c_{i\sigma}^\dagger$ ($c_{i\sigma}$) are the creation (annihilation) operators for an electron at site $i$ with spin $\sigma$, and $n_{i\sigma} = c_{i\sigma}^\dagger c_{i\sigma}$ is the corresponding occupation operator. The first sum runs over nearest-neighbor pairs $\langle ij \rangle$. We consider the system defined on a Bethe lattice with density of states $\rho(\varepsilon) = \frac{2}{\pi D^2}\sqrt{D^2 - \varepsilon^2}$, where $D = 2t$ is the half-bandwidth. Within DMFT framework [7], this lattice model is mapped onto a quantum impurity problem with an effective electronic bath that must be determined self-consistently.

Below, we discuss the key components of a basic DMFT implementation using EDIpack for the Bethe lattice, employing either the Fortran or C++ APIs and which can be found in the `examples` directory of the EDIpack source code. Both samples share a similar initial structure, including memory allocation, creation of the Bethe DOS and solver initialization:

**Fortran**

```fortran
program ed_hm_bethe
   USE EDIPACK
   USE SCIFOR
   implicit none
   integer              :: Le=1000
   real(8)              :: wmixing=0.5d0
   real(8)              :: D=1d0
   integer              :: Nb
   real(8),allocatable  :: Bath(:)
   complex(8),allocatable:: Hloc(:,:,:,:)
   real(8),allocatable  :: Ebands(:),Dbands(:)
   complex(8),allocatable:: Smats(:,:,:,:,:)
   complex(8),allocatable:: Delta(:,:,:,:,:)
   !...
   !EDIpack: Read variables
   call ed_read_input('inputED.conf')

   !Solver-specific arrays. Using Rank-5
   allocate(Smats(Nspin,Nspin,Norb,Norb,Lmats))
   allocate(Delta(Nspin,Nspin,Norb,Norb,Lmats))
   allocate(Hloc(Nspin,Nspin,Norb,Norb))
   Hloc=0d0
   !...
   !Construct Bethe DOS using SciFortran
   !de = 2*D/(Le-1)
   allocate(Ebands(Le),Dbands(Le))
   Ebands = linspace(-D,D,Le,mesh=de)
   Dbands = dens_bethe(Ebands,D)*de
   !
   !EDIpack: Set impurity Hamiltonian
   call ed_set_hloc(Hloc)
```

**C++**

```cpp
#include <edipack_cbindings.h>
using namespace std;
//...
//Main variables
int Le = 1000;
int iloop = 0;
double wmixing = 0.5;
double D = 1.0;

//EDIpack: Read  variables
char input[] = "inputED.conf";
read_input(input);
//Dimensions
int64_t d[4] = {Nspin,Nspin,Norb,Norb};
int total_size = d[0] * d[1] * d[2] * d[3];
int total_size_n5 = total_size * Lmats;
//Solver-specific arrays rank5
vector<complex<double>> Hloc(total_size);
vector<complex<double>> Smats(total_size_n5);
vector<complex<double>> Delta(total_size_n5);
//...
//Construct Bethe DOS
vector<double> Ebands, Dbands;

//Locally defined functions: de = 2*D/(Le-1)
Ebands = linspace(-D,D,Le);
Dbands = dens_bethe(Ebands,D,de);

//EDIpack: Set impurity Hamiltonian
ed_set_Hloc_single_N4(Hloc.data(), d);
```

The bath is described by the (unknown) function $\mathcal{G}_0^{-1}$, i.e. the Weiss field (`Weiss`). In the ED method implemented in EDIpack, the bath is approximated using a finite number of discrete energy levels. The Weiss field $\mathcal{G}_0^{-1}$ is then used to determine the bath parameters $\vec{x} = \{\hat{V}, \hat{h}\}$ through the optimization method outlined in Sec. 3.12.

The starting point for any calculation is a reasonable initial guess for the Weiss field, or equivalently, the bath parameters. In EDIpack, this is accomplished using the function `ed_init_solver` (Fortran API) or `init_solver_site` (C++ API).

**Fortran**

```fortran
!EDIpack: Initialize solver
Nb=ed_get_bath_dimension()
allocate(bath(Nb))
call ed_init_solver(bath)
```

**C++**

```cpp
//EDIpack: Initialize solver
int Nb;
vector<double> Bath(Nb);
Nb = get_bath_dimension_direct();
int64_t bath_dim[1] = {Nb};
init_solver_site(Bath.data(), bath_dim);
```

The iterative algorithm to solve the DMFT problem proceeds as follows:

**EDIpack** Call the EDIpack**impurity solver** whose only input is the set of parameters $\vec{x}$ contained in a rank-1 array. All EDIpack options are controlled through input file specifications.

**EDIpack** Retrieve the self-energy functions $\Sigma_{\alpha\beta\sigma\sigma'}(i\omega)$ on the Matsubara axis using dedicated function `ed_get_sigma` available in the EDIpack API.

**User** Evaluate the local interacting Green's function

$$G_{\mathrm{loc}}(i\omega) = \int_{-D}^{D} \frac{\rho(\varepsilon)}{\zeta - \varepsilon} d\varepsilon$$

with $\zeta = i\omega + \mu - h^0 - \Sigma(i\omega)$. Note that this step can be performed analytically for the Bethe lattice [7] or completely substituted by retrieving the impurity Green's function $G_{\mathrm{imp}}$ via the `ed_get_gimp` function.

**User** Update the Weiss field via the **self-consistency** relation: $\mathcal{G}_0^{-1}(i\omega) = G_{\mathrm{loc}}^{-1}(i\omega) + \Sigma(i\omega)$. For the Bethe lattice, this simplifies to $\Delta = \frac{D^2}{4}G_{\mathrm{loc}}$ or, equivalently, $\Delta = \frac{D^2}{4}G_{\mathrm{imp}}$.

**User > EDIpack** Optimize the bath parameters $\vec{x}$ against the updated Weiss field using the conjugate gradient procedures supplied by EDIpack. Then, restart at step 1.

The first two steps are handled directly by EDIpack routines, while during the subsequent steps the control returns to the user, who must implement the algebraic updates required to close the self-consistency loop and optimize the bath. Given the critical importance of this step, EDIpack provides access to a well-tested implementation of the conjugate gradient method for performing the bath optimization, ensuring stability and reproducibility of the results. This task is conceptually distinct from the diagonalization of the impurity problem, which remains the core focus of the package. Alternative optimization methods can also be employed as needed [127, 133, 134]. An example of implementation is provided in the following listings.

**Fortran**

```fortran
!DMFT loop
do while(.not.converged.AND.iloop<nloop)
    iloop=iloop+1

    !EDIpack: Call ED solver
    call ed_solve(bath)
    !EDIpack: Retrieve Σ(iωₙ)
    call ed_get_sigma(Smats,'m')

    !Build local Green's function
    wfreq = pi/beta*(2*arange(1,Lmats)-1)
    do i=1,Lmats
        zeta= xi*wfreq(i)+xmu - Smats(1,1,1,1,i)
        Gmats(1,1,1,1,i)=sum(DOS/(zeta-Ene))*de
    enddo

    !Self-consistency
    Delta = 0.25d0*D*Gmats

    !Fitting -> new bath
    call ed_chi2_fitgf(Weiss,bath,ispin=1)

    !Check convergence: from DMFTtools
    converged=check_convergence(Delta,&
        dmft_error,Nsuccess,Nloop)
enddo
```

**C++**

```cpp
//DMFT loop
while (iloop < Nloop && !converged) {
    //EDIpack: Call ED solver
    solve_site(Bath.data(),bath_dim,1,1);
    //EDIpack: Retrieve Σ(iωₙ)
    get_sigma_site_n5(Smats.data(),//
        0,0,wm.data(),Lmats,0);
    //Build local Green's function
    for (int i=0;i<Lmats;++i) {
        zeta= wm[i] + xmu - Smats[i];
        Gmats[i] = complex<double>(0.0,0.0);
        for (int j=0; j< Le; j++) {
            Gmats[i]+=Dbands[j]/(zeta-Ebands[j]);
        }
    }
    //Self-consistency
    for (int i = 0; i < Lmats; ++i) {
        Delta[i] = 0.25 * D * Gmats[i];
    }
    //Fit -> new bath
    chi2_fitgf_single_normal_n5(Delta.data(),//
        delta_dim,Bath.data(),bath_dim,1,0,1);
    //Check convergence: local functions
    converged = check_convergence(Delta,//
        dmft_error, Nsuccess, Nloop, comm);
}
```

**Results.** In the following, we present EDIpack results for the interaction-driven MIT obtained with previous implementations. The MIT captures the gradual transformation of a partially-filled metallic state into a correlated insulator.

To illustrate this, we report in panel (A) the evolution of the spectral function $-\mathrm{Im}G_{\mathrm{loc}}(\omega)/\pi$ as a function of the interaction strength $U$. Despite the inherently *spiky* nature of the spectrum, resulting from the finite number of poles in the discretized effective bath, the characteristic features of the Mott transition are clearly visible. The results in panel (A) have been obtained using a broadening `eps=0.01`. At low energies, a renormalized quasi-particle peak develops at the Fermi level ($\omega = 0$). Simultaneously, the system exhibits the formation of incoherent high-energy features, which eventually evolve into well-defined Hubbard bands in the Mott insulating phase for $U > U_{\mathrm{c}}$, with $U_{\mathrm{c}} \simeq 2.8D$.

The formation of a spectral gap separating the Hubbard bands in the Mott insulating phase is associated with the divergence of the imaginary part of the self-energy at the Fermi level. This divergence reflects the complete localization of the electrons, effectively suppressing coherent quasi-particle excitations. Causality dictates that the real part of the self-energy must also grow significantly near the singularity, making it impossible to satisfy the quasi-particle pole equation

$$\omega + \mu - h^0 - \varepsilon - \mathrm{Re}\Sigma(\omega) = 0,$$

which governs the formation of coherent excitations near the Fermi level. Panels (B) and (C) illustrate this phenomenon by showing the evolution of the self-energy $\Sigma$. In panel (B), we present the Matsubara self-energy $\mathrm{Im}\Sigma(i\omega)$ in the low-energy regime. As the interaction strength $U$ increases, this function progressively grows, eventually diverging as the critical interaction threshold $U > U_{\mathrm{c}}$ is crossed. This divergence along the Matsubara axis is directly linked to the particle-hole symmetry of the Bethe lattice, which pins the $\mathrm{Im}\Sigma$ singularity at $\omega = 0$. Panel (C) complements this picture by displaying the real part $\mathrm{Re}\Sigma(\omega)$ on the real-axis near the Fermi level. Here, increasing $U$ leads to a rapid rise of this component, culminating in a discontinuous behavior as the critical point is approached. This discontinuity directly reflects the divergence in the imaginary part on the real-axis, confirming the transition to the Mott insulating state.

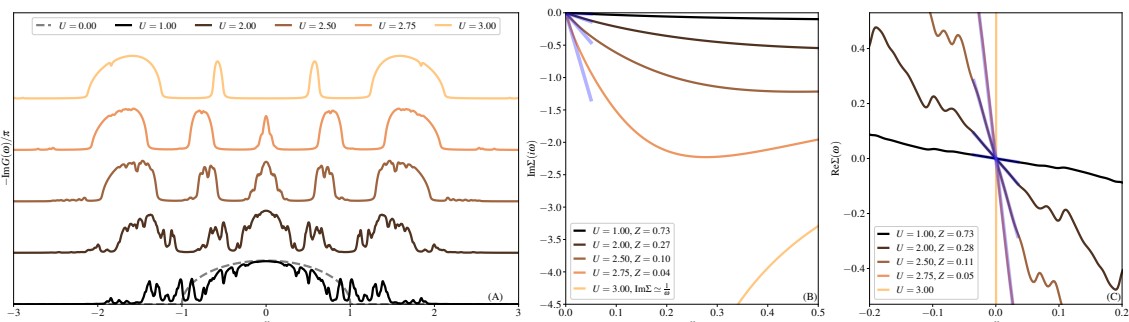

Figure 1: **The metal-insulator Mott transition.** (a) Evolution of the spectral function $-\text{Im}G(\omega)/\pi$ as a function of increasing interaction $U$. The critical interaction $U_{\mathsf{c}} \simeq 2.8D$ separates the correlated metal $U < U_{\mathsf{c}}$ from the Mott insulator $U > U_{\mathsf{c}}$. (b)-(c) The corresponding evolution of the Matsubara self-energy $\text{Im}\Sigma(i\omega)$ (b) and real-axis one $\text{Re}\Sigma(\omega)$ across the Mott transition. For a particle-hole symmetric case, both allow to estimate the renormalization constant $Z$ (see main text), using linear order expansion in frequency (blue solid lines). The values of $Z$ are reported in the legend. The Mott insulating solution is associated with a singularity at $\omega = 0$ of $\text{Im}\Sigma$.

A quantitative measure of this transition is provided by the quasi-particle renormalization factor $Z$, which can be used to capture the degree of electron delocalization. This parameter ranges from 1 for a non-interacting metal to 0 for a fully localized Mott insulator. It is defined through the low-energy expansion of the self-energy as

$$Z = \left(1 - \left.\frac{\partial \text{Re}\Sigma}{\partial \omega}\right|_{\omega \to 0}\right)^{-1},$$

which can also be estimated from the linear behavior of $\text{Im}\Sigma(i\omega)$ for $\omega \to 0$ in the metallic regime using the relation:

$$\left.\frac{\text{Im}\Sigma(i\omega_n)}{\omega_n}\right|_{\omega_n \to 0} = \frac{1}{\pi} \int_{\mathbb{R}} d\epsilon \frac{\text{Re}\Sigma(\epsilon)}{\epsilon^2} = \left.\frac{\partial \text{Re}\Sigma}{\partial \omega}\right|_{\omega \to 0}.$$

The linear fits highlighted in panels (B) and (C), along with the corresponding $Z$ values provided in the legends, clearly indicate that the slope of the self-energy at low energy increases with $U$ on both the Matsubara and real axes. At the transition point, this slope diverges, reflecting the onset of complete electron localization as $Z \to 0$, consistent with the singular behavior $-\text{Im}\Sigma(\omega \to 0) \to \infty$.

### 5.1.1 Finite temperature (w2dynamics interface, CT-HYB benchmark)

To demonstrate how the w2dynamics interface integrates with EDIpack, we briefly discuss how to solve the same problem, i.e. the Hubbard model on the Bethe lattice within DMFT using w2dynamics. In order to showcase the capabilities of EDIpack to address low-temperature problems, we compare the continuous-time Quantum Monte Carlo (CTQMC) solver using the hybridization expansion method (CT-HYB) included in w2dynamics against the EDIpack solver at finite temperature.

Unlike EDIpack, which provides only the impurity solver and bath optimization procedures and requires the user to implement the DMFT algorithm themselves (potentially using various methods), w2dynamics adopts a fundamentally different approach: a single

Python script, `DMFT.py`, handles the entire DMFT calculation, leveraging dedicated classes that implement the generic self-consistency. The w2dynamics calculation is then entirely controlled by a model-dependent parameters file `Parameters.in`, which contains a number of variable specifications including options to control the ED solver inherited from EDIpack. Further information about the functioning of w2dynamics can be found in Ref. [67]

Another important difference concerns the initial point of the iterative DMFT solution algorithm: while EDIpack starts from a given discrete bath, w2dynamics is initialized with a zero self-energy function or alternatively reads this quantity from a converged solution file. Thus, when using the EDIpack interface in w2dynamics, the initial Weiss field is determined using self-consistency and an initial discrete bath is obtained through the EDIpack bath optimization procedure.

The following is the w2dynamics configuration file used to solve the Bethe lattice problem at finite temperature:

```
[General]
DOS             = Bethe          # support for the Bethe lattice is built-in
half-bandwidth  = 1              # list of half-bandwidths per orbital
NAt             = 1              # number of impurities
beta            = 100            # inverse temperature
mu              = 1.0            # chemical potential set to achieve half-filling
EPSN            = 0.0            # turns off filling-based chemical potential search
DMFTsteps       = 100            # given no convergence checking, we might want fewer
magnetism       = para           # symmetrize self-energies per spin
FileNamePrefix  = bethe_dmft_U2  # prefix for the output file name
fileold         = bethe_dmft*hdf5 # file to read an initial self-energy from
readold         = 0              # iteration number to read initial self-energy from, 0 turns off
mixing          = 0.5            # mixing, but mixes self-energies and not Weiss fields
mixing_strategy = linear         # linear mixing as in the Fortran example
FTType          = none           # (for CT-HYB): use the NFFT-measured G
solver          = EDIPACK        # use EDIpack as impurity solver, not default CTHYB

[Atoms]
[[1]]                            # one subsection per impurity
Nd              = 1              # number of orbitals
Hamiltonian     = Kanamori       # create a Hubbard-Kanamori interaction Hamiltonian
Udd             = 2.0            # equivalent to ULOC
Vdd             = 1.0            # equivalent to UST, meaningless for 1 orbital
Jdd             = 0.5            # equivalent to JH, JX and JP, also meaningless here

[EDIPACK]                        # further ED parameters, as in the Fortran example
NBATH           = 7              # number of bath sites
ED_TWIN         = True           # use twin symmetry
LFIT            = 2048           # number of Matsubara frequencies used for the bath fit
LANC_NGFITER    = 500            # number of Lanczos iterations for Green's function
CG_FTOL         = 1e-10          # conjugate-gradient tolerance
CG_NITER        = 2048           # maximum number of conjugate-gradient iterations
ED_FINITE_TEMP  = True           # at finite temperature T = 1/beta

[QMC]                            # Parameters for some grid sizes and the CT-HYB calculation
Ntau            = 1024           # imaginary time grid size
Niw             = 4096           # number of positive Matsubara frequencies
# parameters only relevant for CT-HYB follow
MeasGiw         = 1              # enable NFFT measurement of G
NCorr           = 175            # estimate of the autocorrelation length
Nmeas           = 200000         # number of measurements / sample size
Nwarmups        = 1000000        # number of initial warmup steps for Markov chain thermalization
```

Assuming the parameters are listed in a plain text file called `bethe_dmft.in`, the DMFT simulation can then by run on `NC` cores via:

```
mpiexec -n NC /path/to/w2dynamics/DMFT.py bethe_dmft.in
```

The results are collected into an HDF5 [135] archive in the usual w2dynamics format, including output quantities inherited from EDIpack. Results can be viewed using the script `hgrep` provided with w2dynamics or with other HDF5 tools. In this example we extract the Matsubara self-energy function $\Sigma(i\omega_n)$ for the last DMFT iteration:

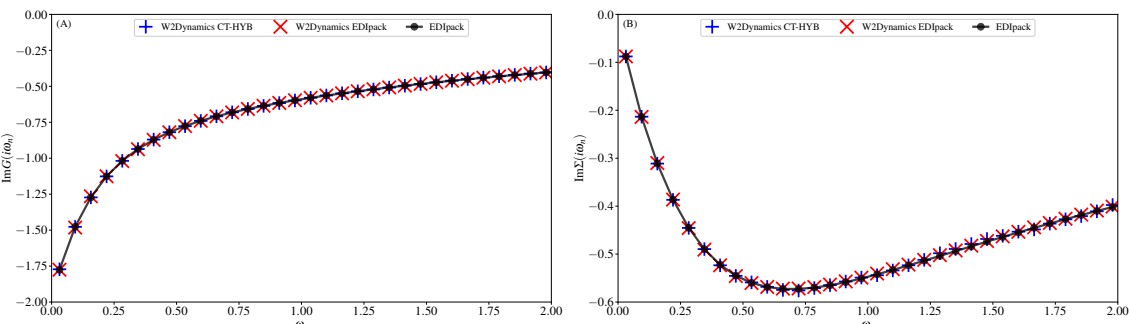

Figure 2: **Finite temperature DMFT solution.** Comparison of the imaginary parts of the Green's function $\mathrm{Im}G(i\omega_n)$ and self-energy $\mathrm{Im}\Sigma(i\omega_n)$ from different solutions of the Hubbard model on the Bethe lattice using DMFT. Data are for $T/D = 0.01$ and $U/D = 2.0$. CTQMC results from the solver included with w2dynamics are compared against EDIpack ED results, used both through the w2dynamics interface and with a standalone Fortran program.

```
/path/to/w2dynamics/hgrep latest siw-full -1
```

In Fig. 2 we report a comparison of the results obtained using the hybridization expansion CTQMC and the EDIpack ED method as solvers in w2dynamics. In addition we compare with results obtained using the Fortran API directly as shown in the previous subsection. Using DMFT, we solve the Hubbard model on a Bethe lattice for $T/D = 0.01$ and $U/D = 2.00$, corresponding to a correlated metallic state.

In panel (A) we show the behavior of the imaginary part of the impurity Green's function $\mathrm{Im}G(i\omega_n)$ which is the direct output for both methods (note that w2dynamics does not directly have access to the real-axis Green's function when using the CTQMC solver). In panel (B) we show the same comparison for impurity self-energy $\mathrm{Im}\Sigma(i\omega_n)$. The results obtained from the three calculations are in excellent agreement with each other already for a small number of bath levels (`NBATH=7`) and relatively low QMC statistics (`Nmeas=200000` with `NC=10` parallel processes).

## 5.2 Attractive Hubbard model (Python API, `ed_mode=superc`)

The second example we present concerns the DMFT description of the attractive Hubbard model [54, 56, 57] on a two-dimensional square lattice. This example has two main goals: (i) to demonstrate EDIpack's support for *s*-wave superconductivity, and (ii) to showcase the Python API through a concrete example.

The model Hamiltonian is given by:

$$H = \sum_{\mathbf{k},\sigma} \epsilon(\mathbf{k}) c^\dagger_{\mathbf{k}\sigma} c_{\mathbf{k}\sigma} - U \sum_i n_{i\uparrow} n_{i\downarrow},$$

where $U > 0$, $c^\dagger_{i\sigma}$ is the creation operator for an electron at site $i$ with spin $\sigma$ and $c^\dagger_{\mathbf{k}\sigma} = \frac{1}{\sqrt{N}} \sum_i e^{-i\mathbf{k}\cdot R_i} c^\dagger_{i\sigma}$. The occupation operator is $n_{i\sigma} = c^\dagger_{i\sigma} c_{i\sigma}$, and the energy dispersion relation is $\varepsilon(\mathbf{k}) = -2t[\cos(k_x a) + \cos(k_y a)]$, where we set the lattice spacing $a = 1$ and the choose the energy unit such that $4t = D = 1$ for convenience.

The DMFT workflow for this case is largely similar to the previous example, but it now operates in the Nambu basis defined by the spinor $\psi_i = [\hat{c}_{i\uparrow} \quad \hat{c}^\dagger_{i\downarrow}]^T$ where the symbol $\hat{o}$ indicates the potential multi-orbital nature of the system, which reduces to a scalar in the

present single-orbital case. In this basis, the Green's function takes the matrix form:

$$\mathbf{G} = \begin{pmatrix} \hat{G}_{\uparrow\uparrow} & \hat{F}_{\uparrow\downarrow} \\ \hat{\bar{F}}_{\downarrow\uparrow} & \hat{\bar{G}}_{\downarrow\downarrow} \end{pmatrix} \tag{31}$$

The components in the second row, denoted by $\hat{\bar{A}}$, are connected to the first row by particle-hole and time-reversal symmetries. The specific relations depend on the symmetry of the order parameter (here, $s$-wave) and whether the functions are defined on the Matsubara or real-frequency axis:

$$\hat{\bar{G}}(i\omega) = -\hat{G}^*(i\omega) \; ; \quad \hat{\bar{F}}(i\omega) = \hat{F}(i\omega)$$
$$\hat{\bar{G}}(\omega) = -\hat{G}^*(-\omega) \; ; \quad \hat{\bar{F}}(\omega) = \hat{F}^*(i\omega) \tag{32}$$

The code implementation closely follows the structure of the previous example, with some notable adjustments related to the Nambu basis. These symmetries allow computing only the independent components in the first row, reducing the computational effort. Note that part of the operations required to implement the DMFT cycle are implemented in the Python module `aux_funx.py`, adapting from `DMFTtools` functions. The initial part of the code handles the lattice structure and solver initialization, as described below.

```python
import numpy as np
#Import EDIpack2py:
from edipack2py import global_env as ed
#Import MPI support
import mpi4py
from mpi4py import MPI
#Import functions to build G_loc and perform DMFT self-consistency in Nambu space
from aux_funx import *
import os,sys

#Start MPI framework:
comm = MPI.COMM_WORLD
rank = comm.Get_rank()
master = (rank==0)

#Functions: build 2D grid and dispersion ε(k)
def generate_kgrid(Nk):
    b1=2*np.pi*np.array([1.0,0.0])
    b2=2*np.pi*np.array([0.0,1.0])
    n1, n2 = np.meshgrid(np.arange(Nk), np.arange(Nk))
    n1=n1/Nk;n2=n2/Nk
    gridout = np.stack([n1.ravel(), n2.ravel()], axis=-1)
    return np.dot(gridout,[b1,b2])

def h_square2d(k,t):
  return -2*t*( np.cos(k[...,0,np.newaxis,np.newaxis])+
            np.cos(k[...,1,np.newaxis,np.newaxis]))*np.eye(ed.Norb)

#Read input
ed.read_input("inputAHM.conf")

#Generate H_k = ε(k) and set Hloc
kgrid   = generate_kgrid(Nk)
Hk      = h_square2d(kgrid,t_hop)
Hloc    = np.sum(Hk,axis=0)/Nk**2
ed.set_hloc(Hloc.astype(complex))

#Build dispersion in Nambu space ε(k)τ^z
HkNambu = np.array([h_square2d(kgrid,t_hop),-np.conj(h_square2d(-kgrid,t_hop))])

#Setup ED Solver
Nb=ed.get_bath_dimension()
bath = ed.init_solver()
```

The iterative scheme for the solution of DMFT closely follows the sequence already discussed in Sec. 5.1:

EDIpack Call the exact diagonalization **impurity solver ed.solve** providing the set of bath parameters $\vec{x} = \{V, h\}$ as input.

- Use the dedicated input/output EDIpack procedures to retrieve the self-energy functions $\hat{\Sigma}(i\omega)$ and $\hat{S}(i\omega)$ on the Matsubara axis.

EDIpack Evaluate the interacting local Green's functions $\hat{G}_{\rm loc}$ and $\hat{F}_{\rm loc}$:

$$
\mathbf{G}_{\rm loc}(i\omega) = \int_{\mathbb{R}} d\varepsilon \rho(\varepsilon) \begin{pmatrix} (i\omega + \mu)\hat{\mathbb{1}} - \hat{h}^0 - \hat{\Sigma}(i\omega) - \varepsilon & -\hat{S}(i\omega) \\ -\hat{S}(i\omega) & (i\omega + \mu)\hat{\mathbb{1}} + \hat{h}^0 + \hat{\Sigma}^*(i\omega) + \varepsilon \end{pmatrix}^{-1}
$$
(33)

User Update the Weiss field's components, respectively $\mathcal{G}_0^{-1}$ and $\mathcal{F}_0^{-1}$, through the **self-consistency** relation: $\boldsymbol{\mathcal{G}}_0^{-1}(i\omega) = \mathbf{G}_{\rm loc}^{-1}(i\omega) + \boldsymbol{\Sigma}(i\omega)$ in Nambu space.

User> EDIpack Optimize the bath parameters $\vec{x}$ to best describe the updated Weiss fields, potentially using the EDIpack provided conjugate gradient fit procedures.

The corresponding implementation in Python reads:

```python
#DMFT CYCLE
converged=False;iloop=0
while (not converged and iloop<ed.Nloop):
    iloop=iloop+1
    #Solve quantum impurity problem for the current bath
    ed.solve(bath)

    #Retrieve the Matsubara self-energy components Σ(iωn) and S(iωn)
    Smats = np.array([ed.get_sigma(axis="m",typ="n"),ed.get_sigma(axis="m",typ="a")])

    #Perform self-consistency levaraging {\tt aux\_funx.py} procedures:
    Gmats = get_gloc(wm*1j,ed.xmu,HkNambu,Smats,axis="m")
    Weiss = dmft_weiss_field(Gmats,Smats)

    #Fit Weiss field and update the bath
    bath = ed.chi2_fitgf(Weiss[0],Weiss[1],bath)

    #Error check
    err,converged=ed.check_convergence(Weiss,ed.dmft_error)
ed.finalize_solver()
```

**Results.** Here we showcase some results for the DMFT solution of the attractive Hubbard model across the BCS-to-BEC crossover regime [17, 56, 57] illustrating the capability of EDIpack to handle $s$-wave superconductivity at both zero and finite temperatures.

To begin, panel (A) of Fig. 3 shows the evolution of the spectral density, obtained from the local normal Green's function as $-\frac{1}{\pi}\mathrm{Im}G_{\rm loc}(\omega)$, as a function of the attraction $U$. For any finite $U$, the Van Hove peak near the Fermi level characteristic of the 2D square lattice (visible at $U = 0$) is split by the formation of a superconducting gap. The latter reflects the emergence of a finite order parameter $\phi = \langle c_\uparrow c_\downarrow \rangle$ and onset of superconducting coherence.

The evolution of the order parameter with attraction $U$ is presented in panel (B). The figure highlights the crossover from the weak-coupling BCS regime to the strong-coupling BEC regime. In the BCS limit, $\phi$ displays the characteristic exponential growth with $U$, known to be computationally challenging. In the opposite, strong-coupling, limit $\phi$ saturates at its theoretical maximum of $\phi \to 1/2$. The comparison with the BCS mean-field result (gray line) reveals the effect of local dynamical fluctuations, which slightly suppress the order parameter, particularly in the intermediate regime. To further quantify these dynamical effects, we plot in the inset of the same panel the *correlation strength* [136, 137] $\Theta = |S(i\omega \to 0) - S(i\omega \to \infty)|/S(i\omega \to \infty)$. Large values of $\Theta$ indicate an anomalous

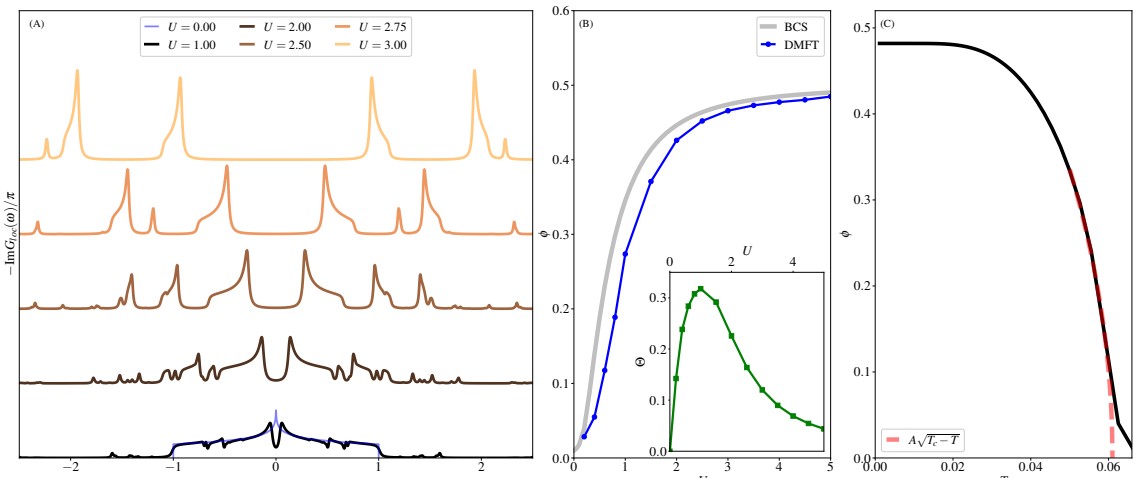

Figure 3: **The BCS to BEC crossover.** (A) Evolution of the spectral functions $-\mathrm{Im}G_{\mathrm{loc}}(\omega)/\pi$ as a function of increasing attraction $U$. (B) The order parameter $\phi = \langle c_\uparrow c_\downarrow \rangle$ as a function of the attraction $U$. Data for BCS (gray) is compared to DMFT results (blue line and symbols). Inset: the correlation strength $\Theta$ (see main text) as a function of the attraction $U$ across the BCS-BEC crossover. (C) Superconducting order parameter $\phi$ as a function of temperature across the superconductor-to-normal phase transition. Data for $U = 4$. The fit highlights the critical behavior with a mean-field exponent $\beta = 1/2$ (red dashed line) and parameters $A \simeq 3.7$, $T_{\mathrm{c}} = 0.61$.

self-energy with significant dynamical effects, hence a more correlated superconducting state. Our results show that the DMFT solution significantly departs from the BCS picture in the strongly correlated intermediate-coupling, a regime known to host the highest critical temperature [56, 57].

Finally, we demonstrate the capability of EDIpack to describe finite temperature effects. Panel (E) shows the temperature dependence of the order parameter $\phi(T)$ across the superconducting-to-normal transition. The mean-field nature of this transition is evident from the scaling near the critical temperature $\phi \sim (T_{\mathrm{c}} - T)^\beta$ with $\beta = 1/2$, consistent with Ginzburg-Landau theory.

These results collectively illustrate the versatility of EDIpack in handling both ground state and finite temperature superconducting phases, for instance capturing the complex physics of the BCS-BEC crossover with high accuracy.

## 5.3 Holstein model on the Bethe lattice (electron-phonon coupling)

One of the new features introduced in EDIpack is the support for local phonons in combination with superconductivity, i.e. `ed_mode=superc`. In order to illustrate this property using a simple application, in this example we discuss the normal and superconductive solution of the pure Holstein model on the Bethe lattice within DMFT. Note that the code implementation for this case is essentially identical to the listings in the previous sections 5.1 (`ed_mode=normal`) and 5.2 (`ed_mode=superc`), provided electron-electron interaction is set to zero and phonon parameters are properly configured.

We consider the model introduced in Sec. 5.1 with $U = 0$ and the additional phononic and electron-phonon terms:

$$H_{\mathrm{int}} = \sum_i \left[ \omega_0 b_i^\dagger b_i + g(b_i^\dagger + b_i) \sum_\sigma \left( c_{i\sigma}^\dagger c_{i\sigma} - \frac{1}{2} \right) \right]. \tag{34}$$

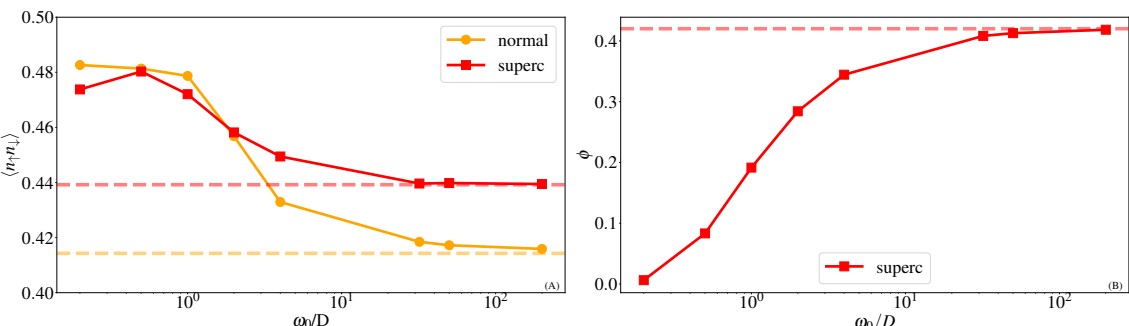

Figure 4: Evolution of the double occupancy (A) and superconductive order parameter (B) as a function of $\omega_0/D$ for $\lambda = 1$ in the normal (orange) and superconductive (red) phase. The horizontal broken lines are the values for the corresponding Hubbard model in the anti-adiabatic limit.

We focus on the half-filling regime of the particle-hole symmetric Bethe lattice DOS. We set the half-bandwidth as our energy unit $D = 1$ and introduced the electron-phonon coupling $\lambda = \frac{2g^2}{\omega_0}$. The iterative DMFT solution algorithm follows the same principles illustrated in the previous sections.

**Results.** In the following we discuss the adiabatic ($\omega_0 \to 0$) to anti-adiabatic ($\omega_0 \to \infty$) crossover for the uniform solution of the Holstein model at constant coupling $\lambda = 1.0$. In the anti-adiabatic limit, the Holstein interaction takes a particularly simple form:

$$H_{\text{int}} \xrightarrow{\omega_0 \to \infty} -\frac{\lambda}{2} \sum_i \left[ \sum_\sigma \left( c_{i\sigma}^\dagger c_{i\sigma} - \frac{1}{2} \right) \right]^2, \tag{35}$$

which describes a local Hubbard-like attraction among electrons (mediated by local phonons). In the adiabatic limit, $\omega_0 \to 0$ the system enters a Bipolaronic Insulating phase for our choice of the coupling [103].

We characterize the model solution by showing the evolution of the double occupation $\langle n_\uparrow n_\downarrow \rangle$ as a function of the phonon frequency $\omega_0$, see panel (A) of Fig. 4. In this panel we compare the behavior for the normal phase (`ed_mode=normal`) and the superconductive phase (`ed_mode=superc`). Our results capture the whole crossover from adiabatic to anti-adiabatic regime. In the former regime, the double occupation takes a similar value for the two phases. However, approaching the anti-adiabatic regime, the two solutions reach the limiting values corresponding to the residual attraction Eq. (35). To better characterize the nature of the superconducting phase in the Holstein model, we report the evolution of the anomalous order parameter $\phi$ in the adiabatic to anti-adiabatic crossover, see panel (B) of Fig. 4. The DMFT results obtained with EDIpack show a rapid increase in the superconducting order parameter as phonon frequency grows large. Finally, in the anti-adiabatic regime, $\phi$ saturates to a finite value corresponding to the attractive Hubbard-like interaction of strength $\lambda/2$.

## 5.4 Multi-orbital impurity with Kanamori interaction (TRIQS interface)

We proceed to demonstrate how to use the EDIpack2TRIQS compatibility layer (Sec. 4.3) to solve a quantum system comprised by a 3-orbital correlated impurity coupled to a few non-interacting bath sites. The interaction term of the impurity Hamiltonian is in the Hubbard-Kanamori form of Eq. (4).

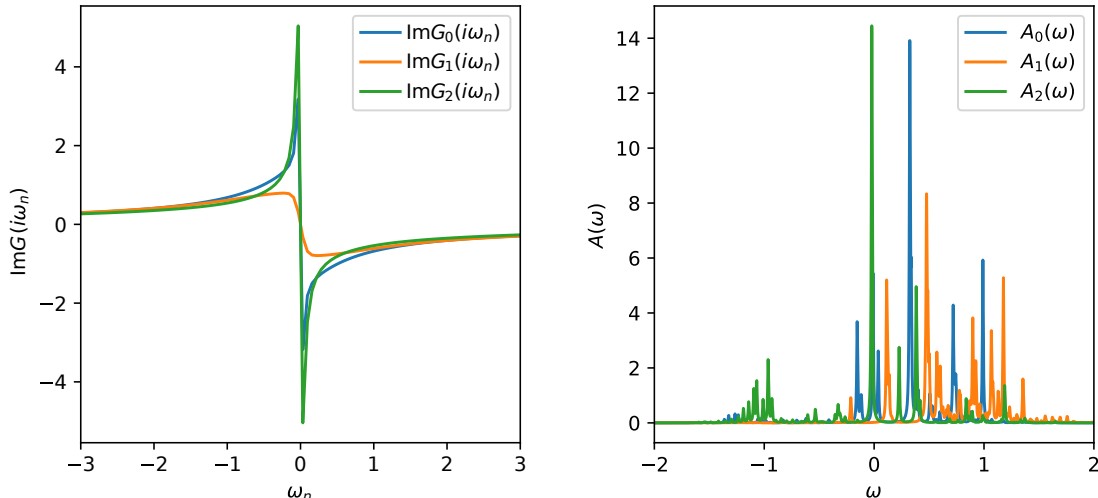

Figure 5: Imaginary part of the Matsubara Green's function $G_\alpha(i\omega_n)$ (left) and the corresponding orbital-resolved spectral function $A_\alpha(\omega) = -\mathrm{Im}G_\alpha(\omega)/\pi$ (right) computed for a three-orbital impurity model with an interaction of the Hubbard-Kanamori type (4). This illustration is produced by the EDIpack2TRIQS example script presented in Sec. 5.4.

A Python script implementing a calculation that makes use of EDIpack2TRIQS generally begins with a few module imports.

```python
# Import common Python modules
from itertools import product
import numpy as np

# Initialize the MPI library before creating the solver object
from mpi4py import MPI

# Import TRIQS many-body operator objects: annihilation, creation and
# occupation number operators
from triqs.operators import c, c_dag, n

# Import the solver object
from edipack2triqs.solver import EDIpackSolver
```

One then proceeds to defining the system under consideration. In this case we consider an impurity atom with three correlated orbitals and two bath states per each impurity orbital, which corresponds to `bath_type=normal` in EDIpack. This information must be encoded in the fundamental set objects that are later used to construct the solver.

```python
spins = ['up', 'dn']      # Names of spin projections
orbs = [0, 1, 2]          # List of impurity orbitals
bath_sites = [0, 1]       # List of bath sites, each carrying 3 orbitals

# Fundamental sets for impurity spin up/down degrees of freedom
fops_imp_up = [('up', orb) for orb in orbs]
fops_imp_dn = [('dn', orb) for orb in orbs]

# Fundamental sets for bath spin up/down degrees of freedom
# There are a total of 6 bath states for each spin projection
fops_bath_up = [('B_up', i) for i in range(3 * 2)]
fops_bath_dn = [('B_dn', i) for i in range(3 * 2)]
```

The next step is to define a TRIQS many-body operator expression that represents the Hamiltonian to be diagonalized.

```python
# Non-interacting part of the impurity Hamiltonian
h_loc = np.diag([-0.7, -0.5, -0.7])
H = sum(h_loc[o1, o2] * c_dag(spin, o1) * c(spin, o2)
        for spin, o1, o2 in product(spins, orbs, orbs))

# Interaction part
U = 3.0      # Local intra-orbital interactions U
Ust = 1.2    # Local inter-orbital interaction U'
Jh = 0.2     # Hund's coupling
Jx = 0.15    # Spin-exchange coupling constant
Jp = 0.1     # Pair-hopping coupling constant

H += U * sum(n('up', o) * n('dn', o) for o in orbs)
H += Ust * sum(int(o1 != o2) * n('up', o1) * n('dn', o2)
               for o1, o2 in product(orbs, orbs))

H += (Ust - Jh) * sum(int(o1 < o2) * n(s, o1) * n(s, o2)
                      for s, o1, o2 in product(spins, orbs, orbs))
H -= Jx * sum(int(o1 != o2) * c_dag('up', o1) * c('dn', o1) * c_dag('dn', o2) * c('up', o2)
              for o1, o2 in product(orbs, orbs))
H += Jp * sum(int(o1 != o2) * c_dag('up', o1) * c_dag('dn', o1) * c('dn', o2) * c('up', o2)
              for o1, o2 in product(orbs, orbs))

# Bath part

# Matrix dimensions of eps and V: 3 orbitals x 2 bath sites
eps = np.array([[-0.1, 0.1],
                [-0.2, 0.2],
                [-0.3, 0.3]])
V = np.array([[0.1, 0.2],
              [0.1, 0.2],
              [0.1, 0.2]])

# Dispersion of the bath states
H += sum(eps[o, nu] * c_dag("B_" + s, nu * 3 + o) * c("B_" + s, nu * 3 + o)
         for s, o, nu in product(spins, orbs, bath_sites))
# Coupling between the impurity and the bath
H += sum(V[o, nu] * (c_dag(s, o) * c("B_" + s, nu * 3 + o)
                     + c_dag("B_" + s, nu * 3 + o) * c(s, o))
         for s, o, nu in product(spins, orbs, bath_sites))
```

Finally, one creates a solver object and performs the actual calculation by calling its method `solve()`. This is normally the most time- and memory-consuming step.

```python
# Create a solver object
solver = EDIpackSolver(H, fops_imp_up, fops_imp_dn, fops_bath_up, fops_bath_dn)

# Solve the impurity model
beta = 100.0                    # Inverse temperature
n_iw = 1024                     # Number of Matsubara frequencies for GF calculations
energy_window = (-2.0, 2.0)     # Energy window for real-frequency GF calculations
n_w = 4000                      # Number of real-frequency points for GF calculations
broadening = 0.005              # Broadening on the real axis for GF calculations

solver.solve(beta=beta,
             n_iw=n_iw,
             energy_window=energy_window,
             n_w=n_w,
             broadening=broadening)
```

The computation results are readily available as attributes of the solver object. The following code snippet shows how to access measured expectation values of the static observables, and how to employ the plotting framework of TRIQS to visualize the obtained Matsubara and real-frequency impurity Green's functions (Fig. 5).

```python
# On master MPI node, output some calculation results
if MPI.COMM_WORLD.Get_rank() == 0:
    # Print values of the static observables
    print("Potential energy:", solver.e_pot)
    print("Kinetic energy:", solver.e_kin)
    print("Densities (per orbital):", solver.densities)
```

```
print("Double occupancy (per orbital):", solver.double_occ)

# Use TRIQS' extensions to matplotlib to plot computed Green's functions
from triqs.plot.mpl_interface import plt, oplot

# Plot the Matsubara Green's functions (imaginary part)
plt.figure(figsize=(4, 4))
for orb in orbs:
    oplot(solver.g_iw['up'][orb, orb], mode='I',
          label=r"${\rm Im} G_" + str(orb) + r"(i\omega_n)$")
plt.xlim((-3, 3))
plt.ylabel(r"${\rm Im} G(i\omega_n)$")
plt.legend()
plt.savefig("G_iw.pdf")

# Plot the orbital-resolved spectral functions
plt.figure(figsize=(4, 4))
for orb in orbs:
    oplot(solver.g_w['up'][orb, orb], mode='S', label=f"$A_%i(\\omega)$" % orb)
plt.xlim(energy_window)
plt.ylabel(r"$A(\omega)$")
plt.legend()
plt.savefig("A_w.pdf")
```

## 5.5   Interacting Bernevig-Hughes-Zhang model (Fortran API, `ed_mode=nonsu2`)

In this section, we focus on the DMFT solution of the interacting Bernevig-Hughes-Zhang (BHZ) model. As shown in Ref. [112], this model exhibits an excitonic phase at moderate interactions in which opposite orbital electrons and holes across the band gap bind and form a coherent phase [112, 113, 122, 138–140]

We consider a system of two-orbital electrons on a square lattice, interacting via a Hubbard-Kanamori term. This system realizes a quantum spin Hall insulator [112, 136, 141–146]. We consider a suitable matrix basis in terms of the Dirac matrices $\Gamma_{a\alpha} = \sigma_a \otimes \tau_\alpha$, where $\sigma_a$ and $\tau_\alpha$ are Pauli matrices, respectively, in the spin and orbital pseudo-spin space. The model Hamiltonian reads

$$H = \sum_k \psi_k^\dagger H(k)\psi_k + H_{\rm int},$$

where $\psi_k = [c_{1\uparrow k}, c_{2\uparrow k}, c_{1\downarrow k}, c_{2\downarrow k}]^T$ is the spinor collecting annihilation operator $c_{a\sigma k}$ destroying an electron at orbital $a = 1, 2$ with spin $\sigma = \uparrow, \downarrow$ and lattice momentum $k$. The non-interacting part of the Hamiltonian is:

$$H(k) = [M - 2t(\cos k_x + \cos k_y)]\,\Gamma_{03} + \lambda \sin k_x \Gamma_{31} - \lambda \sin k_y \Gamma_{02},$$

where $M$ is the mass term, which plays the role of a crystal field splitting among the orbitals. The presence of this term breaks the symmetry in the orbital pseudo-spin channel. The interaction reads:

$$H_{\rm int} = (U - J)\frac{\hat{N}(\hat{N} - 1)}{2} - J\left(\frac{1}{4}\hat{N}^2 + \hat{S}_z^{\,2} - 2\hat{T}_z^{\,2}\right),$$

where $\hat{N} = \frac{1}{2}\psi_i^\dagger \Gamma_{00}\psi_i$ is the total density operator, $\hat{S}_z = \frac{1}{2}\psi_i^\dagger \Gamma_{30}\psi_i$ is the total spin polarization operator and $\hat{T}_z = \frac{1}{2}\psi_i^\dagger \Gamma_{03}\psi_i$ is the orbital pseudo-spin polarization operator. This form corresponds to the density-density part of the Kanamori interaction. We neglect the pair-hopping and spin-flip purely for numerical reasons [82]. In the non-interacting regime this model describes a quantum spin Hall insulator (QSHI) for $M < 4t$ and a trivial Band Insulator (BI) for $M > 4t$. The transition point at $M = 4t$ describes the formation of a gapless Dirac state at $k = [0, 0]$.

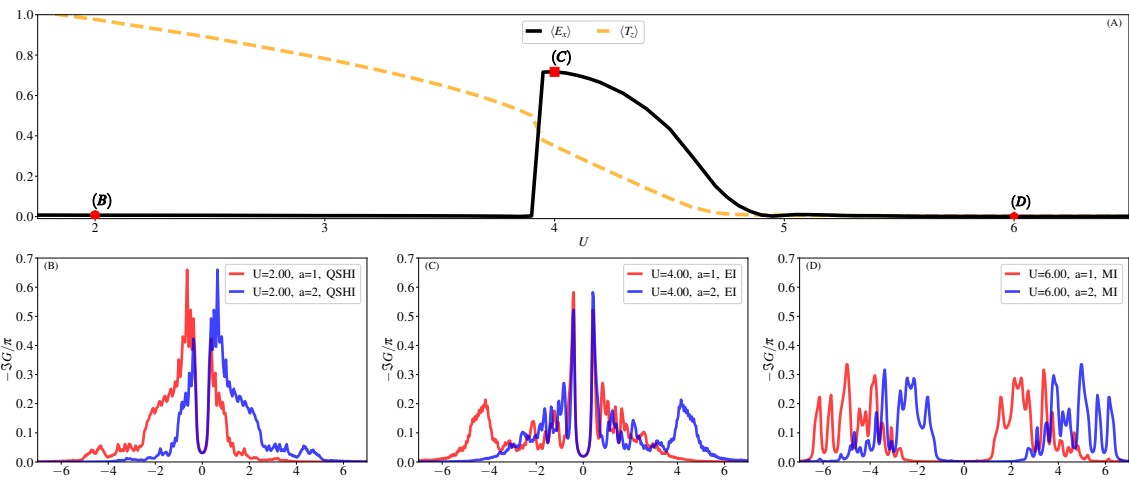

Figure 6: **Topological and Exciton Transition.** (A) Evolution of the spin-triplet, in-plane excitonic order parameter $\langle E_x \rangle$ (black solid line) and orbital polarization $\langle T_z \rangle$ (orange dashed line) as a function of the interaction $U$. (B-D) Spectral functions of the two orbital electrons for three values of the interaction $U = 2.00$ (red circle), 4.00 (red square) and 6.00 (red diamond) capturing, respectively, the QSHI **(B)**, the Excitonic Insulator **(C)** and the Mott Insulator **(D)**.

The code implementation follows the same guidelines discussed above for the other examples. The user is required to generate the Hamiltonian $H(k)$ on a discretized Brillouin zone. This is employed to evaluate the local interacting Green's function:

$$G_{\text{loc}}(z) = \frac{1}{N_k} \sum_k \left[ z + \mu - H(k) - \Sigma(z) \right]^{-1},$$

entering in self-consistency conditions $\mathcal{G}^{-1} = G_{\text{loc}}^{-1} + \Sigma$. Here $\zeta \in \mathbb{C}$, $N_k$ is the number of $k$-points and $\Sigma(z)$ is the self-energy matrix obtained from the DMFT solution of the problem. Moreover, the $H(k)$ is used to construct the *renormalized* topological Hamiltonian [147–149]

$$H_{\text{top}} = \sqrt{Z}[H(k) + \text{Re}\Sigma(\omega \to 0)]\sqrt{Z},$$

where $Z$ is the renormalization constant matrix. The matrix $H_{\text{top}}$ describes the low-energy properties of the many-body solution and can be used to characterize the topological nature of the interacting system [148–152].

**Results**  To capture the possible symmetry breaking in any excitonic channel driven by the local interaction, we consider the vector order parameter $\vec{E} = [E_0, E_x, E_y, E_z]$, where $E_a = \langle \psi^\dagger \Gamma_{a1} \psi \rangle$ [138,153–160].

The first component $E_0$ describes the singlet excitonic state, whereas the remaining ones correspond to the triplet states with different spin orientation [112,113]. The analysis of the strong-coupling regime as well as impurity excitonic susceptibility available in EDIpack suggest the possible instability to an in-plane spin-triplet exciton phase, i.e. with $E_x$ and/or $E_y$ different from zero, see [112]. Interestingly, this state breaks several symmetries including time-reversal and spin SU(2), which protect the topological state.

Here we showcase the DMFT description of the excitonic phase in the interacting BHZ model as a way to illustrate the EDIpack ability to capture matter phases with lowered spin-symmetry (`ed_mode=nonsu2`) and the use of `bath_type=replica`.

The initial part of the code implementation is a simple generalization of the previously discussed cases. The next non-trivial steps are: i) construct the lattice Hamiltonian $H(k)$

and the corresponding local part $H_{\text{loc}}$, which determines the impurity properties, and ii) construct a matrix basis representation for the replica bath. These two steps can be implemented as follows:

```
!> Set H_loc
allocate(Hloc(Nso,Nso))
Hloc = sum(Hk,dim=3)/Lk
where(abs(dreal(Hloc))<1d-6)Hloc=zero

!EDIpack: set the impurity Hamiltonian: H_loc → h^0
call ed_set_hloc(Hloc)

!EDIpack: set the replica bath matrix basis Γ and λ
!Here Nsym=4.
!Build the basis and init the variational parameters:
allocate(lambdasym_vector(Nbath,4))
allocate(Hsym_basis(Nso,Nso,4))
Hsym_basis(:,:,1)=Gamma03 ;lambdasym_vector(:,1)= Mh
Hsym_basis(:,:,2)=Gamma01 ;lambdasym_vector(:,2)= sb_field
Hsym_basis(:,:,3)=Gamma31 ;lambdasym_vector(:,3)= sb_field
Hsym_basis(:,:,4)=Gamma11 ;lambdasym_vector(:,4)=-sb_field

!EDIpack: set the basis and initial values of λ
call ed_set_Hreplica(Hsym_basis,lambdasym_vector)

!EDIpack: get bath dimension and allocate user bath to this size
Nb=ed_get_bath_dimension(4)
allocate(Bath(Nb))

!EDIpack: Initialize the ED solver
call ed_init_solver(bath)
```

Here we use the local non-interacting Hamiltonian $H_{\text{loc}} = \frac{1}{N_k} \sum_k H(k)$ to set $h^0_{\alpha\beta\sigma\sigma'}$, i.e. the impurity Hamiltonian. To anticipate the possibility of forming an excitonic ordered phase, the replica bath is constructed out of a matrix basis with 4 distinct elements $\Gamma_{03}$, $\Gamma_{01}$, $\Gamma_{31}$, $\Gamma_{11}$ which are proportional, respectively, to the mass term, the exciton singlet $E_0$, the exciton triplet along easy-axis $E_z$ and in-plane $E_x$ (we assume $E_y = 0$ leveraging on residual $U(1)$ in-plane symmetry). In the following we consider the case of $M = 1$, which corresponds to a QSHI in the non-interacting limit and $J/U = 0.25$. The implementation is nearly identical to the cases discussed above and we report it here for completeness:

```
iloop=0;converged=.false.
do while(.not.converged.AND.iloop<nloop)
   iloop=iloop+1

   !EDIpack: Solve the impurity problem
   call ed_solve(bath)

   !EDIpack: Retrieve Σ(iω)
   call ed_get_sigma(Smats,axis="mats")

   !Get G_loc using DMFTtools
   call get_gloc(Hk,Gmats,Smats,axis="m")

   !Update the Weiss field (self-consistency) using DMFTtools
   call dmft_self_consistency(Gmats,Smats,Weiss)

   !Linear mixing the Weiss fields
   if(iloop>1)Weiss = wmixing*Weiss + (1.d0-wmixing)*Weiss_;Weiss_=Weiss

   !EDIpack: Fit to update the bath
   call ed_chi2_fitgf(Weiss,bath,ispin=1)

   !Check convergence: using DMFTtools
   converged = check_convergence(Weiss(1,1,:),dmft_error,nsuccess,nloop)
enddo
```

The main effect of the interaction on the topological properties is contained in the mass term renormalization. The real part of the self-energy being proportional to $\Gamma_{03}$ corrects the mass term with respect to its bare value: $M_{\text{eff}} = M + \frac{1}{4}\text{TrRe}\Sigma(i\omega \to 0)$. To leading order (mean-field) this correction is proportional to $\frac{U-5J}{4}\langle \hat{T}_z\rangle$. So, for our choice of parameters, the effect of interaction would be to effectively reduce the mass term. Thus, in the strong-coupling limit the two orbitals get populated by one electron per site, reaching the conditions for the formation of a Mott insulating state. This effect is highlighted in panel (A) of Fig. 6, where we report the progressive reduction of the orbital polarization $\langle T_z\rangle$ and, thus, of the effective mass.

In the same panel (A), we show that at intermediate coupling the DMFT solution features the formation of a region of exciton condensation with $\langle E_x\rangle > 0$, $\langle E_0\rangle = \langle E_z\rangle = 0$.

Unlike the static mean-field description [113], the transition from the QSHI topological state to the Excitonic Insulator (EI) becomes of first-order upon including the local dynamical fluctuations [111, 146] contained in DMFT. The EI continuously evolves into a Mott Insulator (MI) for larger interaction (neglecting for simplicity any antiferromagnetic ordering that would naturally occur in this regime). The EDIpack solution of this model nicely captures the two transitions which involve breaking of the spin symmetry group SU(2).

To further illustrate the nature of the three distinct phases of this system, we rely on the direct access to the real-axis spectral function provided by EDIpack solver. In panels (B)-(D) of Fig. 6 we report the spectral functions $-\text{Im}G_{a\uparrow,\text{loc}}(\omega)/\pi$ for $a = 1, 2$, and three distinct values of the interaction $U$ placing the solution, respectively, in the QSHI, the EI and the MI.

The correlated QSHI is characterized by the presence of an inverted band gap featuring a substantial orbital spectral mixing. The EI spectrum is instead characterized by a narrow gap, related to the finite order parameter, flanked by two sharp resonances and featuring larger high-energy weight. Finally, in the MI the two-orbital spectral function describes the two characteristic Hubbard bands separated by a large Mott gap.

## 5.6 Interacting Kane-Mele model (Fortran API, EDIpack2ineq, iRDM)

To complete the overview of EDIpack's features, we consider an additional interacting quantum spin-Hall insulator model, defined on the honeycomb lattice, with two inequivalent sites in the unit cell. Neglecting the Rashba spin-orbit interaction and discarding any ionic character of the unit cell, the noninteracting Kane-Mele Hamiltonian [141, 161] can be written in momentum space as

$$H_{\text{KM}} = \sum_k \psi_k^\dagger H(k)\psi_k,$$

with $\psi_k = [c_{k,\text{A},\uparrow},\, c_{k,\text{B},\uparrow},\, c_{k,\text{A},\downarrow},\, c_{k,\text{B},\downarrow}]$ and

$$\begin{aligned}
H(k) &= x_k\Gamma_{01} + y_k\Gamma_{02} + \delta_k\Gamma_{33}, &(36)\\
x_k &= t\big[\cos(k \cdot u) + \cos(k \cdot v) + 1\big],\\
y_k &= t\big[\sin(k \cdot u) + \sin(k \cdot v)\big],\\
\delta_k &= 2\lambda_{\text{so}}\big[\sin(k \cdot u) - \sin(k \cdot v) - \sin(k \cdot u - k \cdot v)\big],
\end{aligned}$$

where $u = (3a/2, \sqrt{3}a/2)$, and $v = (3a/2, -\sqrt{3}a/2)$ are suitable basis vectors for the honeycomb lattice, $t$ is the nearest-neighbor hopping amplitude and $\lambda_{\text{so}}$ is the amplitude of a complex next-nearest-neighbor hopping, with a spin-dependent $\pm\frac{\pi}{2}$ chiral phase arising from spin-orbit coupling, which is well-known to open a topological gap at the

Fermi level [141, 161]. The spinorial $4 \times 4$ matrices $\Gamma_{a\ell} = \sigma_a \otimes \tau_\ell$ are defined in terms of Pauli matrices $\sigma_a$, $\tau_\ell$ referred, respectively to spin and sublattice degrees of freedom. To investigate the effects of electron-electron repulsion we consider a local Hubbard term [162, 163]:

$$H_{\text{KMH}} = H_{\text{KM}} + U \sum_i \left[ \left( n_{i,\uparrow} - \frac{1}{2} \right) \left( n_{i,\downarrow} - \frac{1}{2} \right) \right], \tag{37}$$

where $n_{i,\sigma} = c_{i,\sigma}^\dagger c_{i,\sigma}$ are the local spin-density operators. The interaction is written in an explicitly particle-hole symmetric form so that the model is at half-filling for zero chemical potential.

The essential features of the ground state phase diagram have been established by intensive multi-method analyses [163]. For $\lambda_{\text{so}} = 0$ the model describes a Dirac semimetal, up to large repulsion, and magnetizes to an isotropic Néel state above a critical $U/t$ ratio [164, 165]. For a finite spin-orbit coupling $\lambda_{\text{so}} \neq 0$, the quantum spin-Hall state results increasingly stable against magnetic ordering, due to its symmetry-protected nontrivial topology. Furthermore, the long-range ordering eventually stabilized at large $U/t$, is characterized by a lowered spin symmetry, due to the inherent coupling of spin and lattice degrees of freedom that favors in-plane easy-axis magnetization [166], as predicted by Hartree-Fock and $t$-$J$ perturbative expansions [167] and later confirmed by Auxiliary-Field Quantum Monte Carlo (AFQMC) simulations [162].

To treat the in-plane Néel ordering with EDIpack, we can once again exploit the `ed_mode=nonsu2` option, allowing off-diagonal spin amplitudes in all dynamical matrices and so eventually in the resulting impurity self-energy. Referring to the two inequivalent sublattices as A and B, whereas a standard off-plane ($S_z$) calculation would enforce Néel symmetry (AFM$_\perp$) as $\Sigma_{\uparrow\uparrow}^A = \Sigma_{\downarrow\downarrow}^B$, $\Sigma_{\downarrow\downarrow}^A = \Sigma_{\uparrow\uparrow}^B$, with $\Sigma_{\uparrow\downarrow}^A = \Sigma_{\downarrow\uparrow}^A = \Sigma_{\uparrow\downarrow}^B = \Sigma_{\downarrow\uparrow}^B = 0$, an in-plane magnetization (AFM$_\parallel$) corresponds to $\Sigma_{\uparrow\uparrow}^A = \Sigma_{\uparrow\uparrow}^B$, $\Sigma_{\downarrow\downarrow}^A = \Sigma_{\downarrow\downarrow}^B$, $\Sigma_{\uparrow\downarrow}^A = -\Sigma_{\downarrow\uparrow}^B$, $\Sigma_{\downarrow\uparrow}^A = -\Sigma_{\uparrow\downarrow}^B$.

The same off-diagonal spin symmetry must be realized also in the bath Hamiltonian, which can be conveniently achieved with either the `replica` and `general` choices for the `bath_type` option. The most robust strategy to verify that the ground state of the model is magnetized in the plane also in DMFT is to allow either an $S_z$ or an $S_x$ (equivalently $S_y$) magnetization in the bath, and then compare the energies of the respective solutions at $T = 0$. The corresponding Hamiltonian forms are:

$$H_\perp^{\text{bath}} = \sum_{p=1}^{N_b} (\epsilon_p \sigma_0 + m_p \sigma_z), \tag{38}$$

$$H_\parallel^{\text{bath}} = \sum_{p=1}^{N_b} (\epsilon_p \sigma_0 + m_p \sigma_x). \tag{39}$$

which corresponds to $N_{\text{sym}} = 2$, with $\lambda_{1,p} \equiv \epsilon_p$ as the degenerate bath levels that are Zeeman split by $\lambda_{2,p} \equiv m_p$, respectively in the $S_z$ and $S_x$ bases (see Sec. 3.6).

```fortran
program dmft_kane_mele_hubbard
  !Load EDIpack library
  USE EDIPACK
  !Load the inequivalent impurities extension
  USE EDIPACK2INEQ
  ...
  !Setup bath and solver for anisotropic AFM phases
  !> define the symmetry-basis:
  Nineq = 2
  Nsym = 2
  allocate(Hsym_basis(Nspin*Norb,Nspin*Norb,Nsym))
  Hsym_basis(:,:,1) = pauli_0
  select case(ansatz)
```

```fortran
  case('off-plane');Hsym_basis(:,:,2) = pauli_z
  case('in-plane') ;Hsym_basis(:,:,2) = pauli_x
end select
!> initialize the symmetry parameters
allocate(lambdasym_vectors(Nineq,Nbath,Nsym))
call build_replica_band(onsite_band,ed_hw_bath,Nbath)
! (user-built guess of the starting bath levels)
lambdasym_vectors(:,:,1) = onsite_band
lambdasym_vectors(:,:,2) = 0d0 ! paramagnetic
!> define a symmetry-breaking AFM kick, to help numerics
if(afmkick)then
   lambdasym_vectors(1,:,2) = +sb_field
   lambdasym_vectors(2,:,2) = -sb_field
endif
!> setup H_bath inside the solver
select case(bath_type)
  case('replica')
      call ed_set_Hreplica(Hsym_basis,lambdasym_vectors)
  case('general')
      call ed_set_Hgeneral(Hsym_basis,lambdasym_vectors)
end select
!> finally initialize the two solver instances, one per sublattice
!  (the EDIpack2ineq layer automatically dispatches it for the two
!   impurity models, if we feed a Bath with the [Nineq,Nb] shape!)
Nb = ed_get_bath_dimension(Hsym_basis)
allocate(Bath(Nineq,Nb))
call ed_init_solver(Bath)
...
```

The DMFT loop can then be implemented, with the explicit solution of two independent impurity problems (prone to numerical noise) or by solving only one impurity problem and building the full solution by exploiting the appropriate Néel symmetry:

```fortran
 iloop=0;converged=.false.
 do while(.not.converged.AND.iloop<nloop)
    iloop=iloop+1
    !
    !> Solve the two inequivalent impurity problems
    if(neelsym)then
      !> solve just one sublattice and get the other by Neel symmetry
      call ed_solve(Bath(1,:))
      call ed_get_sigma(Smats(1,:,:,:,:,:),axis='m')
      select case(ansatz)
          case('off-plane')
            Smats(2,2,2,:,:,:) = Smats(1,1,1,:,:,:)
            Smats(2,1,1,:,:,:) = Smats(1,2,2,:,:,:)
            if(master)write(*,*) ">>> Enforcing AFMz symmetry"
          case('in-plane')
            Smats(2,1,1,:,:,:) = Smats(1,1,1,:,:,:)
            Smats(2,2,2,:,:,:) = Smats(1,2,2,:,:,:)
            Smats(2,1,2,:,:,:) = -Smats(1,1,2,:,:,:)
            Smats(2,2,1,:,:,:) = -Smats(1,2,1,:,:,:)
            if(master)write(*,*) ">>> Enforcing AFMx symmetry"
      end select
    else
      !> solve both sublattices independently using EDIpack2ineq:
      !  mpi_lanc=T => MPI lanczos, mpi_lanc=F => MPI for ineq sites
      call ed_solve(Bath,mpi_lanc=.true.)
      !> retrieve all self-energies:
      call ed_get_sigma(Smats,Nineq,axis='m')
      !
    endif
    !
    !> Get G_loc(iω_n) : using DMFTtools
    call dmft_gloc_matsubara(Hk,Gmats,Smats)
    !
    !> Update local Weiss fields 𝒢_0^{-1} = G_{loc}^{-1} + Σ: using DMFTtools
    call dmft_self_consistency(Gmats,Smats,Weiss)
    !
    !> Fit the new bath:
    !  - normal mode: normal/AFMz and we fit spin-components independently
    !  - nonsu2 mode: broken Sz-conservation and we fit
```

(a)

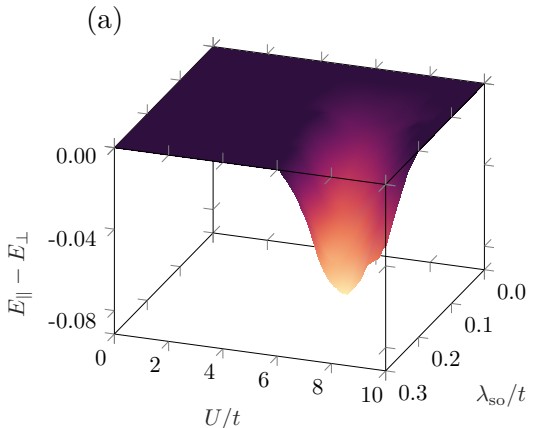

(b)

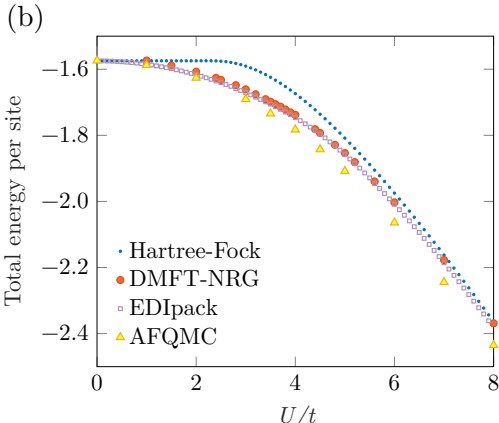

Figure 7: In panel (a) we report the total energy difference between the in-plane and the off-plane AFM solutions found with EDIpack at $T = 0$. Negative values signal a ground state with in-plane magnetization. The plateau at zero energy difference marks the topological phase of the model (a paramagnetic quantum spin-Hall insulator), at all points except on the $\lambda_{so} = 0$ line, where the ground state is an interacting Dirac liquid, at weak coupling, or an isotropic antiferromagnet at strong coupling. In panel (b) we compare the total energies, across the $\lambda_{so} = 0$ line, to published data for Hartree-Fock, DMFT-NRG and AFQMC solutions [168].

```
    !                both spin components together
   select case(ed_mode)
    case("normal")
      call ed_chi2_fitgf(Bath,Weiss,ispin=1)
      call ed_chi2_fitgf(Bath,Weiss,ispin=2)
    case("nonsu2")
      call ed_chi2_fitgf(Bath,Weiss,Hloc)
   end select
   !
   !> linear mixing and convergence check: using DMFTtools
   if(iloop>1) Bath = wmixing*Bath + (1.d0-wmixing)*Bath_prev
   Bath_prev = Bath
   converged = check_convergence(Weiss(:,1,1,1,1,:),dmft_error,nsuccess,nloop)
   !
enddo
!
!> Compute Kinetic Energy: using DMFTtools
call dmft_kinetic_energy(Hk,Smats)
```

After convergence is reached, we evaluate some quantities of interest, e.g. computing the lattice kinetic energy (as implemented in the DMFTtools library). The potential energy, as well as the AFM order parameters and impurity observables are automatically computed by EDIpack, and saved to files with the appropriate inequivalent-impurity label.

We evaluate the total energy of both the AFM$_\perp$ and AFM$_\parallel$ calculations and compare them to establish which is favored as the ground state of the system. In Fig. 7(a) we show results for the energy difference $E_\parallel - E_\perp$ across a wide range of $U/t$ and $\lambda_{so}$ values. To assess the degree of accuracy of our total energy calculation, we report in Fig. 7(b) a $\lambda_{so} = 0$ benchmark against Hartree-Fock (simple static mean-field, but inherently variational), DMFT with NRG solver and lattice AFQMC data, as taken from Ref. [168].

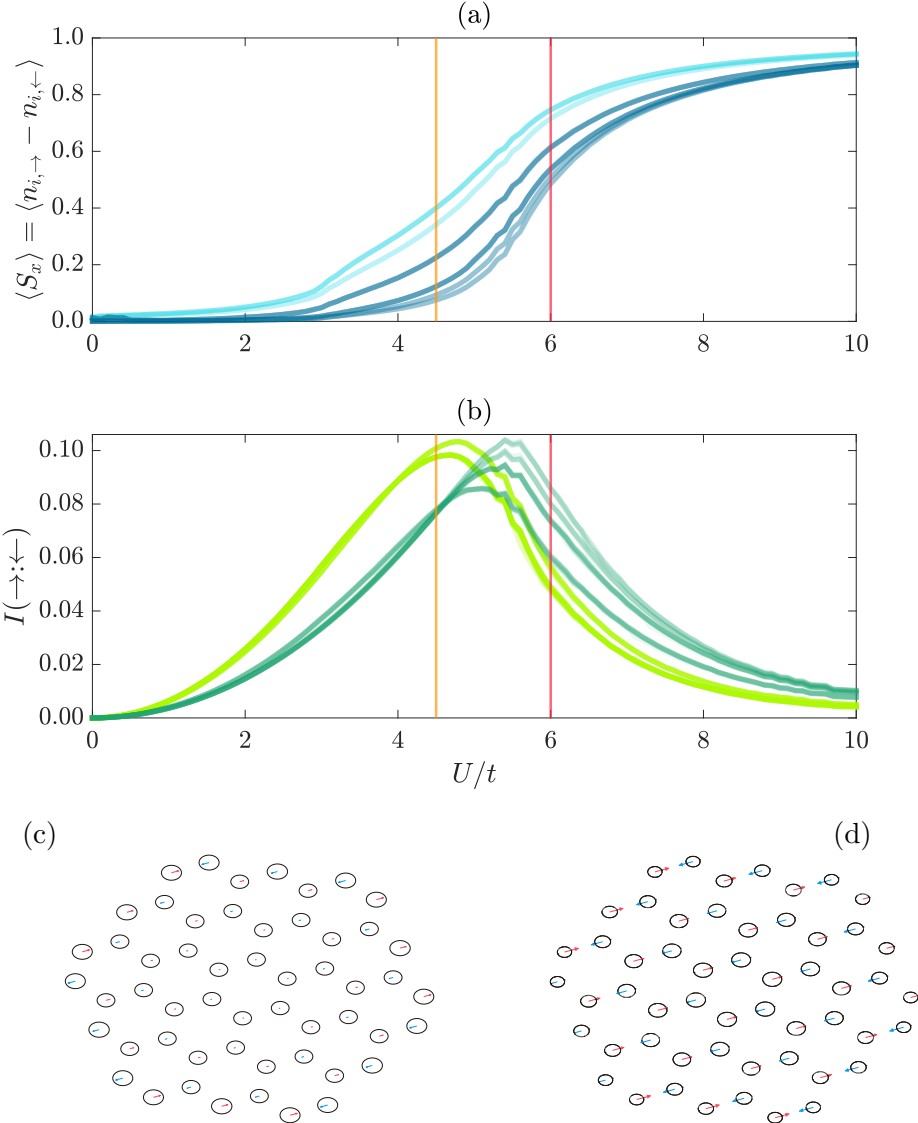

Figure 8: Kane-Mele-Hubbard model at the nanoscale. In (a) the local in-plane magnetization $\langle S_x \rangle = \langle n_{i,\rightarrow} - n_{i,\leftarrow} \rangle$. In (b) the local correlation as measured by the mutual information between local in-plane spin eigenstates $I(\rightarrow : \leftarrow)$ [111, 124, 169]. Both the (a) and (b) panels mark the edge and "bulk" sites in lighter and darker colors, respectively. In (c) and (d) a visual representation of the given nano-system at different values of $U/t$, as marked respectively by the orange and red vertical lines in panels (a) and (b). The area of the circles is proportional to $I(\rightarrow : \leftarrow)$, the length of the arrows to $\langle S_x \rangle$. All data are taken at $\lambda_{so} = 0.3t$ and $T = 0$.

### 5.6.1   Kane-Mele-Hubbard model at the nanoscale

The capabilities of the EDIpack2ineq layer go well beyond the simple case of a two-site unit cell. Building on the previous example, we can readily adapt the calculation to deal with a nanoscopic [17] system with $N_{\text{ineq}}$ inequivalent atoms, such as hexagonal flakes [170, 171]. We consider a $N_{\text{ineq}} \times N_{\text{ineq}}$ local Hamiltonian $H_{\text{loc}}$ with open boundary conditions, and implementing the self-consistent equations with a trivial single momentum $k = 0$. In this configuration the system develops an inhomogeneous magnetization, so that in general no particular lattice symmetry group, e.g. Néel, can be exploited.

In Fig. 8 we report results for a nanosystem of $N_{\text{ineq}} = 54$ sites, solved with the bath parametrization described in Eq. (39), for the in-plane antiferromagnetic state. Panel (a) shows the interaction dependence of the local magnetization, computed from the impurity ground state as $\langle S_x \rangle = \langle c_{i,\uparrow}^{\dagger} c_{i,\downarrow} + c_{i,\downarrow}^{\dagger} c_{i,\uparrow} \rangle \equiv \langle n_{i,\rightarrow} \rangle - \langle n_{i,\leftarrow} \rangle$, so that we can define the $S_x$ spin occupation numbers as

$$\langle n_{i,\rightarrow} \rangle \equiv \frac{\langle n_{i,\uparrow} + n_{i,\downarrow} \rangle + \langle c_{i,\uparrow}^{\dagger} c_{i,\downarrow} + c_{i,\downarrow}^{\dagger} c_{i,\uparrow} \rangle}{2}, \qquad \langle n_{i,\leftarrow} \rangle \equiv \frac{\langle n_{i,\uparrow} + n_{i,\downarrow} \rangle - \langle c_{i,\uparrow}^{\dagger} c_{i,\downarrow} + c_{i,\downarrow}^{\dagger} c_{i,\uparrow} \rangle}{2},$$

which trivially satisfy the conservation of the local average charge $\langle n_{i,\rightarrow} + n_{i,\leftarrow} \rangle = \langle n_{i,\uparrow} + n_{i,\downarrow} \rangle$. Having a direct expression for $\langle n_{i,\rightarrow} \rangle$ and $\langle n_{i,\leftarrow} \rangle$ we can verify that the impurity reduced density matrix, computed by tracing the bath states in the `ed_mode=nonsu2` representation (see 3.10), once diagonalized, takes the familiar form [10, 11, 172]

$$\rho^{\text{imp}} = \begin{pmatrix} \langle (1 - n_{i,\rightarrow})(1 - n_{i,\leftarrow}) \rangle & 0 & 0 & 0 \\ 0 & \langle n_{i,\rightarrow}(1 - n_{i,\leftarrow}) \rangle & 0 & 0 \\ 0 & 0 & \langle (1 - n_{i,\rightarrow}) n_{i,\leftarrow} \rangle & 0 \\ 0 & 0 & 0 & \langle n_{i,\rightarrow} n_{i,\leftarrow} \rangle \end{pmatrix} \tag{40}$$

in the basis of the *natural* spin-orbitals [111, 124, 169], whose occupation numbers are given by $n_{i,\rightarrow}$ and $n_{i,\leftarrow}$. As discussed in [111, 124, 169], by defining the spin traces of $\rho^{\text{imp}}$, as

$$\rho_{\sigma}^{\text{imp}} = \text{Tr}_{\bar{\sigma}}\left[ \rho^{\text{imp}} \right] = \begin{pmatrix} \langle n_{i,\sigma} \rangle & 0 \\ 0 & \langle n_{i,\bar{\sigma}} \rangle \end{pmatrix} \tag{41}$$

we can directly measure local correlations by defining the mutual information between the impurity spin-orbitals $I(\rightarrow:\leftarrow) = S(\rho_{\rightarrow}^{\text{imp}}) + S(\rho_{\leftarrow}^{\text{imp}}) - S(\rho^{\text{imp}})$, where $S(\cdot)$ denotes the von Neumann entropy of its argument.

In panel (b) of Fig. 8 we report the spatially resolved interaction dependency of $I(\rightarrow:\leftarrow)$, as a direct observation of the enhanced local correlations at the boundary of the finite system. While it is usually assumed that edge sites must be more correlated than internal ("bulk") sites, based on the reduced lattice coordination, the inter-spin mutual information quantifies directly the phenomenon: the edge correlations, marked in lighter green, are decidedly higher as long as the system remains in a weakly magnetized regime. When the local magnetization exceeds half of its saturation value, all local correlations start to decrease, as spin fluctuations are damped by ordering. Significantly, the edge correlations get lower than the bulk correlations as the maximum is crossed, consistently with the observation that their magnetization is always higher and so their fluctuations are frozen faster. Finally, we observe that the nanosystem of Fig. 8 has finite local magnetization for arbitrary low interaction strengths, in striking contrast with the infinite lattice. This is a well-known effect of quantum confinement, as proposed in early works on graphene flakes [170, 171].

# 6 Conclusions

We have presented EDIpack, a versatile and high-performance ED solver for generic quantum impurity problems. Building on the massively parallel algorithms introduced by its predecessor [82], this version of the library features new capabilities allowing for different broken-symmetry solutions within a unified framework that provides, for instance, reliable evaluations of a variety of local dynamical correlation functions of an arbitrary complex frequency ranging from the Green's functions to superconducting, excitonic and magnetic response functions.

This new version allows for an efficient evaluation of reduced density matrices of the impurity directly from Fock space quantities and thus enables quantum information analysis of correlated systems, which is becoming an essential tool to address emergent phases of quantum materials.

A central feature of EDIpack is its strong focus on interoperability, achieved through modern Fortran constructs, C/C++-bindings, and comprehensive APIs for Python (EDIpack2py). These interfaces enable seamless integration with broader computational frameworks, like TRIQS [63] and w2dynamics [67], expanding the functionality of these platforms and providing a robust foundation for reproducible research.

We thoroughly discussed the implementation of the EDIpack library and its most important algorithms and classes. We presented in detail the third-party interfaces which extend ED capabilities beyond the domain of the library itself. Finally, we showcased the use of the EDIpack software in different contexts through elaborated examples, which can serve as a reference for potential users.

The modular and extensible design of EDIpack provides a natural foundation for future extensions. For instance it provides a suitable basis to implement cluster-DMFT [9, 115, 173] methods, where the impurity problem is generalized to include clusters of interacting sites. This approach includes spatial correlations beyond the single-site approximation, capturing effects such as $d$-wave pairing, charge ordering, and complex spin textures in strongly correlated systems. A different outlook is the development of an integration layer with other quantum embedding methods, such as the ghost Gutzwiller scheme [25], which heavily rely on solving effective impurity problems. We anticipate that EDIpack will become a valuable tool for the computational condensed matter community, supporting a wide range of studies on strongly correlated materials and providing a reliable reference platform for quantum impurity solvers.

## Acknowledgements

We acknowledge helpful discussions occurring at various stages of development of EDIpack with: C. Mejuto-Zaera, P. Villar Arribi, M. Chatzieleftheriou, S. Adler, F. Paoletti, M. Collura, C. Weber, A. Sartori, H. Choi, A. Valli, M. Rozenberg.

**Author contributions**  A.A. designed the project, developed the main structure of the software and conceived the computational algorithms in the core library. A.A., L.C. and M.C. conceived the diagonalization, correlation function calculation and bath optimization strategies implemented in the library. A.A., G.M., A.S., S.G. and M.C. devised the inclusion of the electron-phonon coupling and of the superconducting state. L.C., G.M., F.P. A.S., S.G. and G.B. equally contributed to the development and implementation of many different aspects of the software, including code writing, configuration, installation, testing and continuous integration procedures. L.C. devised and developed the C-binding

layer, the Python, Julia and Monicelli interface together with Python specific deployment and installation tools. I.K. developed the TRIQS interface and contributed to the testing part. A.K., M.W. and G.S. developed the integration with w2dynamics. A.A., M.C., L.d.M. and G.S designed additional algorithms for extended functionalities of the core software. A.A., L.C., I.K., A.K., S.G. and G.B. performed the calculations, analyzed the results and prepared the data presentation in the examples. A.A. and L.C. wrote the backbone of the manuscript. All authors participated in writing this manuscript.

**Funding information** L.C. acknowledges support from Deutsche Forschungsgemeinschaft (DFG, German Research Foundation) through the cluster of excellence "CUI: Advanced Imaging of Matter" of the Deutsche Forschungsgemeinschaft (DFG EXC 2056, Project ID 390715994) and gratefully acknowledges the scientific support and HPC resources provided by the Erlangen National High Performance Computing Center (NHR@FAU) of the Friedrich-Alexander-Universität Erlangen-Nürnberg (FAU) under the NHR project b158cb. NHR funding is provided by federal and Bavarian state authorities. NHR@FAU hardware is partially funded by the German Research Foundation (DFG)– 440719683. I.K. acknowledges support from the European Research Council (ERC) under the European Union's Horizon 2020 research and innovation programme (Grant agreement No. 854843-FASTCORR). G.S. and A.K. acknowledge financial support by the Deutsche Forschungsgemeinschaft (DFG, German Research Foundation) through FOR 5249 (project-id 449872909, Project P05), and the Würzburg-Dresden Cluster of Excellence on Complexity and Topology in Quantum Matter-ct.qmat (EXC 2147, project-id 390858490). F.P. acknowledges support from the Swiss National Science Foundation via NCCR Marvel and SNSF Grant No. 200021-196966. M.W. acknowledges support from the Austrian Science Fund (FWF) through grant DOI 10.55776/P36332. G.M. acknowledges support from the MUR - Italian Ministry of University and Research - through the Rita-Levi Montalcini program. L.dM. acknowledges support by the European Commission through the ERC-CoG2016, StrongCoPhy4Energy, GA No. 724177. M.C., S.G. and G.B. acknowledge financial support from the National Recovery and Resilience Plan PNRR MUR Project No. CN00000013-ICSC and by MUR via PRIN 2020 (Prot. 2020JLZ52N-002) and PRIN 2022 (Prot. 20228YCYY7) programmes. A.A. and M.C. acknowledge financial support from the National Recovery and Resilience Plan PNRR MUR Project No. PE0000023-NQSTI.

# 7 Appendix A: Monicelli interface

In order to further showcase the interoperability capabilities of EDIpack provided by the C-bindings module, we describe a simple interface to `Monicelli`.

`Monicelli` is an esoteric programming language based on the LLVM toolchain. It is written in C++ and offers wrappers for the basic types and operations thereof. The syntax replicates a cultural Italian phenomenon known as "supercazzola", a rambling nonsensical discourse which gives the false impression of carrying an actual meaning, introduced in the movies trilogy "Amici Miei", directed by Mario Monicelli (see it.wikipedia.org/wiki/Amici-_miei).

The `Monicelli` language can be found at github.com/esseks/monicelli.git and can be installed using CMake. The software depends on C++ compiler with `stdlib`, LLVM and in some cases `libz` library. The installation gives access to the `Monicelli` compiler `mcc`, which statically links LLVM.

We believe the discourse flow of this language is best captured by illustrating its features and EDIpack interface via a paraphrase of the original supercazzola, an epic non-sensical

dialogue between a traffic policeman (V), count Mascetti (M) and G. Perozzi (P) (see it.wikipedia.org/wiki/Supercazzola and Movie Scene).

V:    *Lei ha clacsonato!*
P:    *Tu ha clacsonato?*
      . . .
M:    *Tarapia tapioco, prematurata l'interfaccia, o scherziamo?*
V:    *Prego?*
M:    *Scusi noi siamo in Monicelli, come fosse un linguaggio esoterico basato su C++ e utilizzante la toolchain di LLVM anche per Linux e Unix soltanto in due, oppure in quattro anche scribai con il file sorgente* `hm_bethe.mc`*?*
      *Come* `github.com/lcrippa/prematurata_la_dmft`*, per esempio?*
V:    *Ma che DMFT, mi faccia il piacere! Questi signori stavano programmando loro, non si intrometta!*
M:    *Ma no, dico, mi porga il file* `bagaglio.cpp`*. Le vede le funzioni? Lo vede che interfacciano gli array, non supportati da* `Monicelli`*, e prematurano anche! Ora io le direi, anche con il rispetto per l'autorità, anche solo le due parole come install* `Monicelli` *from* *github.com/esseks/monicelli.git e compila il file* `hm_bethe.mc`*, per esempio.*
V:    *Basta così! Mi seguano nel programma di test!*
P:    *No, no, no, attenzione! Il loop DMFT completo è supportato secondo la Ref. [7], abbia pazienza, sennò…. Plotta i dati, anche un pochino di Green's function e Self-energy in prefettura.*
M:    *Senza contare che* `prematurata_la_dmft` *ha perso i contatti con il tarapia tapioco, dopo.*
      . . .
V:    *Ho bello che capito. Si farà finta di passar da bischeri!*

We shall now present a fully functioning DMFT code for the solution of the Bethe lattice written in Monicelli. It requires a minimal C++ interface to handle arrays, complex numbers and properly interfacing the EDIpack functions. The first part of the DMFT code loads all the required functions from `bagaglio.cpp`, in particular it loads the interface to the EDIpack solver functions `ed_init_solver` and `ed_solve`:

```
bituma le funzioni ausiliarie che vengono dal bagaglio
...
blinda la supercazzola leggi o scherziamo?
blinda la supercazzola iniziailsolver o scherziamo?
blinda la supercazzola risolvi o scherziamo?
blinda la supercazzola prendilasigma o scherziamo?
blinda la supercazzola prendilbagno con l'elemento Necchi, il valore Sassaroli o
  scherziamo?
...
```

After a long initialization, we follow the structure already presented in Sec. 5.1 for the Fortran implementation. The DMFT code starts with the opening `Lei ha clacsonato` (see above). Next, we read the input file and set the dimensions of some array describing the bath, the self-energy and Green's functions. We start a DMFT iteration loop using the command `stuzzica`, which includes an internal frequency loop as `Monicelli` does not support array algebra. The loop contains a call to the EDIpack solver function, then proceeds by retrieving the self-energy function which is used to obtain the local Green's function. These enter the self-consistency equation which updates the Weiss field. Finally bath optimization is performed and if error condition is met the loop exit. The main step of the implementation reads:

```
Lei ha clacsonato
#Read the input file
    prematurata la supercazzola dimmilfile o scherziamo?
    prematurata la supercazzola leggi o scherziamo?
...
#Init the ED solver:
    voglio il sapone, Necchi come se fosse
    prematurata la supercazzola ilbagnoepronto o scherziamo?
    prematurata la supercazzola lavati con il sapone o scherziamo?

    prematurata la supercazzola prendilah o scherziamo?
    prematurata la supercazzola iniziailsolver o scherziamo?
...
#DMFT loop:
stuzzica
      prematurata la supercazzola risolvi o scherziamo?

      prematurata la supercazzola prendilag o scherziamo?
      prematurata la supercazzola prendilasigma o scherziamo?
#Get Gloc and update G^-1: frequency loop
      stuzzica
          il contatoredue come fosse 0
          prematurata la supercazzola prendi con 108, contatore, 0.0, 0.0, 0 o
  scherziamo?
...
      e brematura anche, se il contatore minore di frequenze
...
      prematurata la supercazzola lavatiancora o scherziamo?
#Fit the bath:
      prematurata la supercazzola spiaccica o scherziamo?
#Mix the bath
      prematurata la supercazzola failmischiotto con il sapone, il frullatore o
  scherziamo?
#Check error
      la fine come se fosse prematurata la supercazzola cisiamo o scherziamo?
 e brematura anche, se la fine minore di 1
```

**Usage.** The source code can be retrieved from [github.com/lcrippa/prematurata_la_dmft](github.com/lcrippa/prematurata_la_dmft). The `src` directory contains several files, including the source code `hm_bethe.mc` implementing the DMFT algorithm, an auxiliary file `bagaglio.cpp` which contains a number of functions implementing complex algebra, array construction and interfacing the EDIpack procedures. The directory also contains an example of input file and a converged bath parametrization for the Bethe lattice solution at $U = 2$.

The code is compiled using standard Make invocation in the source directory:

```
git clone https://github.com/lcrippa/prematurata_la_dmft
cd prematurata_la_dmft/src
make
```

A simple run using the provided input file `inputED.conf` will re-converge the solution within a few loops.

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
