# Peer review of "Next-generation EDIpack: A Lanczos-based package for quantum impurity models featuring general broken-symmetry phases, flexible bath topologies and multi-platform interoperability"

_SciPost Physics Codebases, doi:SciPost Phys. Codebases 58 (2025) , SciPost Phys. Codebases 58-r5.3 (2025)_

## Round 1 · Referee Report · Anonymous (Referee 1) · 2025-7-4

Report
Exact diagonalization (ED) plays a vital role in the study of quantum impurity problems. In this work, the authors present an ED library and describe its usage in considerable detail. While it builds upon the previous EDIpack library, the manuscript introduces sufficient new developments to warrant a separate publication. Notably, several significant extensions and improvements are included, such as a unified framework for zero- and finite-temperature calculations for a wide range of impurity models; access to the full complex frequency plane; access to the Fock space and the ability to evaluate quantities like reduced density matrices; support for inequivalent impurities; integration with other programming languages and scientific libraries; and compatibility with widely used software suites such as TRIQS and w2dynamics.
Overall, the paper is well written and highly detailed, allowing new users to install and utilize the library with ease. Furthermore, the comprehensive DMFT tests, demonstrated through a variety of examples and accompanied by corresponding scripts, provide excellent starting points for newcomers. I believe this work makes a valuable contribution to the community and am therefore pleased to recommend its publication in SciPost.
I have a few minor comments on the presentation or requests for clarification:
1) On page 3, a large portion of paragraph 3, which introduces DMFT, lacks citations. For example, the authors mention the success of DMFT in capturing key features of various materials, including Mott insulators, heavy fermion compounds, and unconventional superconductors. It would be helpful to include relevant references here to guide interested readers toward further reading.
2) Perhaps a sentence like “Section 6 provides the conclusions” was intended at the end of Section 1 but was inadvertently omitted?
3) Were the spin indices omitted in Eq. (3)? Or do the indices $i, j, k, l$ represent “flavors”—that is, combined spin and orbital indices—rather than orbitals alone? The same question applies to the display equation of $\hat{H}^{\rm int}$ on page 13. This could be confusing to readers, especially since the spin index is also not explicitly shown in the format of the $<$UMATRIX_FILE$>$.restart file described on the same page.
4) The last display equation on page 10 seems incorrect, since the creation and annihilation operators do not appear to result in any particle being created or annihilated.
5) In Sec. 3.3, it appears that users must manually select one of the three provided symmetry configurations to perform block-diagonalization. Is there a plan to extend the code to automatically detect all relevant symmetries—especially in cases where orbitals or sites possess additional symmetries that would allow further block-diagonalization of the matrix?
6) On page 43, it is mentioned that the spectrum obtained from ED is “inherently spiky”. What is the value of the broadening parameter used in these simulations? Including this information in the main text may improve the clarity and reproducibility of the paper.
Other small comments: $J_x" \rightarrowJ_X"$ and $J_p" \rightarrowJ_P"$ in the second paragraph of page 13; “store” $\rightarrow$ “stored” in the last line of page 13; As discussed" $\rightarrow$As will be discussed" at the beginning of Sec. 3.5.2; $\vec{B_\sigma}$'' $\rightarrow$$\vec{B}_\sigma$'' in the sentence The values are given by all integers $B_\sigma$ corresponding to...'' of page 14; for the display equation of $C(z)$ on page 16, summation index $m$ has lower and upper bounds, while index $n$ does not, although they are dummy variables for the same thing. Same for Eq.~(14); should be no indentation for the sentence right after the first display equation on page 17; $w(\mathcal{A})_{mn}^\nu" \rightarrow w_{mn}^\nu(\mathcal{A})"$ in the second line after Eq.~(14);This example two main goals'' $\rightarrow$ ``This example has two main goals'' on page 46.
Recommendation
Publish (surpasses expectations and criteria for this Journal; among top 10%)
We thank the referee for their time and work in reviewing our manuscript. In the following, we address point by point all the issues raised in their reports.
This paper presents the current status in an exact diagonalisation impurity solver EDIpack. It provides a detailed exposition of the multiple interfaces as well as a number of tutorial-style examples that provide details that will be appreciated to new users of the package. I believe that providing robust and well-tested computer codes, as well as high quality documentation, are a big service to the community. I wholeheartedly recommend this work for publication.
We sincerely thank the referee for their positive comments. We are very pleased to hear that the exposition, interfaces, and tutorial examples were found to be helpful and of value to new users. We are grateful for their recommendation for publication and their support for open and well-maintained scientific software.
On p. 8, function {\tt ed_set_Hloc} is introduced, but without much further information. Maybe the reader could be referred to some later section where its use is illustrated.
We acknowledge the referee’s suggestion. The function ${\tt ed_set_Hloc}$, introduced in Section 3.1.1, is employed to define the impurity Hamiltonian. At this stage of the manuscript, its purpose is solely as described: setting the local non-interacting component of the impurity Hamiltonian, without further elaboration.
To enhance clarity and facilitate reader comprehension of its practical application, we have revised the text at this location. Specifically, we have included a forward reference to Section 5, where ${\tt ed_set_Hloc}$ is explicitly utilized in various application examples. These examples encompass the Hubbard model on the Bethe lattice (Section 5.1), the interacting Bernevig-Hughes-Zhang model (Section 5.5), and the Kane-Mele-Hubbard model (Section 5.6). These instances aim to elucidate both the structure of the input arguments and the context in which this function is commonly employed.
On p. 9, the discussion about the phonon cutoff sounds potentially a bit misleading. The cutoff is surely problem dependent, and there are (generalized) cases where the shift A_m cannot be considered as a free parameter.
We thank the referee for this observation. Indeed, the cutoff on the number of phonon excitations is inherently problem dependent and should be carefully adjusted in light of the specific electron-phonon coupling regime, the phonon frequency, and the interplay with electronic correlations. Furthermore, we agree that the shift $A_m$, while often treated as a tunable parameter in simplified models, is not always free to choose. In more general contexts, such as in first-principles-derived Hamiltonians or systems constrained by symmetry or external potentials, $A_m$ may be fixed by physical considerations or derived from the underlying microscopic model.
To reflect this more nuanced detail, we have revised the paragraph on page 9 to acknowledge that the shift $A_m$ is not universally a free parameter. This ensures that our discussion does not overgeneralise while aligning with the range of physical contexts in which EDIpack may be applied.
I find the discussion at the end of page 11 somewhat unclear, especially the terminology. Is this a discussion of consecutive indexing vs. occupation number representation?
We thank the referee for pointing out this issue. Indeed, the original text aimed to explain how electronic Fock states within a given symmetry sector are indexed and related to their underlying occupation number (bitstring) representation. However, we acknowledge that the terminology and phrasing in the original submission may have been ambiguous.
To clarify this point, we have revised the final paragraph of page 11 in the current version of the manuscript. The updated text now reads:
“From a computational perspective, constructing a symmetry sector amounts to defining an injective map $\mathcal{M} : \mathcal{S}_{\vec{Q}} \rightarrow \mathcal{F}_e$ that relates the consecutive indexing of the electronic states $|i\rangle$ within a given symmetry sector $\mathcal{S}_{\vec{Q}}$ to the Fock states $|I_e\rangle$ in the electronic Fock space in occupation number representation. This map is typically implemented as a rank-1 integer array, whose size corresponds to the dimension of the sector $D^{\mathcal{S}}$.”
This new formulation replaces the previous wording and makes explicit the distinction between the consecutive indexing of the sector states $|i\rangle$ and the Fock states in the occupation number representation $|I_e\rangle$. In doing so, we clarify that the mapping $\mathcal{M}$ is what connects the two encodings of the same physical state. We believe the revised text should eliminate the ambiguity the referee noted.
The motivation of including the code on p. 12 is unclear. What is the intention here?
We thank the referee for raising this point. The purpose of the code snippet in question is to illustrate the concrete implementation of the mapping from Fock states to symmetry sectors for different values of ${\tt ed_mode}$. Our goal was to make explicit the internal logic used to construct the quantum number-conserving subspaces, bridging the gap between formal description and practical realization. We suggest this code snippet is useful to understand the structure of the ${\tt sparse_map}$ used to implement fast algorithm for the evaluation of the reduced impurity density matrix in Section 3.5.2.
In response to the referee’s comment, we have slightly revised the manuscript to clarify this intent. Specifically, we have added the following introductory sentence to the relevant section:
“The following code excerpts illustrate how the sector construction algorithms differ depending on the value of ed_mode, and serve as a concrete reference for understanding how quantum number constraints are enforced in practice.”
Top of p. 14: What is global share, what are istart, ishift, iend? I suppose this is Fortran-specific.
We thank the referee for their close reading and for raising this question concerning the variables ${\tt istart}$, ${\tt iend}$, and ${\tt ishift}$.
We recognise that the earlier reference to the global share was misleading and ultimately unnecessary concept that we have chosen to eliminate for clarity.
The issue at hand concerns the parallel construction of a matrix that is distributed across multiple threads or MPI processes. In such a setting, each thread is responsible for computing and storing only a subset—or "chunk" ${\tt Q}$—of the full matrix, typically defined as a range of contiguous rows. To maintain consistency with the global indexing of the matrix, while allowing each thread to work independently and with locally indexed arrays, a mapping must be established between global and local indices.
The variables ${\tt istart}$ and ${\tt iend}$ define the global index range of matrix rows assigned to a given thread or process. That is, a given thread will handle only the rows from ${\tt istart}$ to ${\tt iend}$ (inclusive), corresponding to its local share of the full matrix. The variable ${\tt ishift}$ is then used to translate global row indices into local ones, effectively setting ${\tt ishift = mpiRank*Q}$. This allows local arrays to be aligned correctly with the globally defined matrix structure.
While the code is written in Fortran, this construction is not specific to the language itself. Rather, it is a general approach for handling parallel matrix distribution in any programming environment where the global matrix is divided into row blocks across multiple processing units. To avoid potential confusion and to make the implementation choices more transparent to the reader in the revised, we now write:
"In a parallel setting, each thread is assigned a contiguous block of matrix rows $Q=D_{\mathcal S}/N_{cpu}$. We denote the locally stored rows on a given thread by $q = 1, \ldots, Q$, and relate them to their corresponding global indices $i = 1, \ldots, D_{\mathcal S}$ via the variables \texttt{istart}, \texttt{iend}, and \texttt{ishift}. Here, \texttt{istart} and \texttt{iend} define the global index range $[q_1, q_Q]$ assigned to the thread, corresponding to $q_1 = \texttt{istart}$ and $q_Q = \texttt{iend}$. The variable \texttt{ishift} $= \texttt{istart} - 1$ allows mapping between the local index $q$ and the global index $i$ through the simple relation $i = q + \texttt{ishift}$. This mapping ensures consistent global indexing while enabling efficient local computation."
p. 16: GFmatrix is said to be a critical component for high-speed execution. Maybe it could be describe in more detail? In what way is it efficient? What does it mean it is multi-layed? In passing, it would be nice if the capitalization would be uniform, e.g. GFmatrix vs. gfmatrix (I understand that Fortran compiler does not care, but the human reader perhaps does).
We thank the referee for pointing out the need for a clearer and more detailed description of the ${\tt gfmatrix}$ type, which indeed plays a central role in the performance of the code. In response, we have revised the corresponding section of the manuscript (page 16) to provide a more precise explanation of its structure and efficiency features.
The term multi-layered refers to the hierarchical organization of the ${\tt gfmatrix}$ object, which is essential for storing and accessing the analytic structure of correlation functions, i.e. the weights and the poles, in a flexible and scalable way. Specifically, the data are organized across three conceptual layers: (1) the many-body eigenstates of the Hamiltonian spectrum, (2) the amplitude channels associated to each such state, and (3) the set of poles and weights corresponding to physical excitations from this state. This structure is necessary to correctly represent the Lehmann decomposition of Green's functions and related objects, while still allowing dynamic access and manipulation during calculations.
The efficiency of ${\tt gfmatrix}$ stems from two main features. First, correlation functions do not need to be precomputed and stored on fixed frequency or time grids; instead, their values are computed on-the-fly at arbitrary complex arguments, allowing precise and flexible evaluation as required by different solvers or output formats. Second, memory usage is optimized by avoiding redundant storage. The internal design separates spectral data and evaluation logic, allowing minimal overhead during the object construction and easing post-processing analysis. We recognize that the use of the term "high-performance" could be misleading and rephrases the paragraph on page 16 in a more precise way as
"The module ${\tt ED_GFMATRIX}$ implements the class ${\tt gfmatrix}$, which provides an efficient and flexible representation of DCFs. It is structured as a hierarchical, or multilayered, data container that organizes spectral weights and poles across three levels: (i) the initial eigenstate $|n\rangle$ of the Hamiltonian spectrum, (ii) the amplitude channel associated with this state, corresponding to the applied operator, and (iii) the set of excitations from $|n\rangle$, each characterized by a pole and its corresponding weight.
This structure allows the ${\tt gfmatrix}$ class to simultaneously handle multiple initial states and operators, enabling on-the-fly evaluation of DCFs at arbitrary complex frequencies $z \in \mathbb{C}$. By avoiding precomputed grids and redundant storage this approach minimizes memory footprint and computational overhead. Through the use of ${\tt gfmatrix}$ EDIpack achieves efficient memory management, consistent storage of spectral data and high-performance post-processing."
We also thank the referee for pointing out inconsistencies in capitalization. We have now ensured that the identifier is uniformly written as gfmatrix throughout the manuscript.
A trick for computing off-diagonal functions is presented on p. 21. Doesn't this require switching to complex-valued floating points even in cases where the Hamiltonian is purely real?
We thank the referee for this observation. It is indeed correct that the general strategy presented for computing off-diagonal DCFs, as described on page 21, involves constructing linear combinations of operators, which in the most general case would require working with complex-valued floating point arithmetic—even if the Hamiltonian itself is real.
However, in the special case where the Hamiltonian is real and symmetric, the resulting Green's functions are symmetric under exchange of orbital indices, i.e., $G_{\alpha\beta}(z) = G_{\beta\alpha}(z)$. This property can be exploited to compute off-diagonal elements using only real-valued linear combinations of operators. In such cases, all required DCFs can be obtained exploiting memory advantages of real-valued computations.
In the revised manuscript we clarify this point by writing in the normal paragraph of the section 3.8:
"The sector Hamiltonian matrix in this mode is assumed
to be real symmetric, which simplifies the evaluation
of the off-diagonal terms by exploiting the following symmetry
under orbital exchange $G_{\alpha\beta\sigma\sigma}(z)=G_{\beta\alpha\sigma\sigma}(z)$.
In this case, the off-diagonal Green's function components can be obtained using just real-valued auxiliary operators such as
${\mathcal O}=c_{\alpha\sigma}+c_{\beta\sigma}$, along with the identity
$G_{\alpha\beta\sigma\sigma}=\tfrac{1}{2}(C_{\mathcal O} - G_{\alpha\alpha\sigma\sigma} - G_{\beta\beta\sigma\sigma})$,
where $C_{\mathcal O}$ denotes the DCF associated to ${\mathcal O}$. This approach, avoiding the need for complex arithmetic, preserves the efficiency of real-valued computation."
In Eq. (18), is Z the same on both sides of approximation sign?
We are indebted with the referee for pointing out this important subtlety, which in fact extends to both sections 3.8 and 3.9. In the original version of the manuscript, the partition function $Z$ appearing in different expressions (including the Eq.(18) indicated by the referee), may have misleadingly suggested that it was being evaluated exactly, even though the trace had been truncated.
In the revised manuscript we have corrected this issue. We now use the symbol $Z$ to indicate the exact partition function and reserve the symbol $\tilde{Z}$ to explicitly indicate the partition function approximated using only the finite set of low-energy eigenstates retained in the trace, i.e., those contributing significantly due to their large Boltzmann weights. This makes the approximation consistent on both sides of the involved equations, namely Eqs. (13), (14) and (18), properly reflecting the finite-state summation used in practice.
Are the code listings on p. 25 and p. 26 of sufficient interest to readers?
We acknowledge the referee observation. We believe that the code listings on pages 25 and 26, i.e. the implementation of the fast-algorithm for the evaluation of the impurity RDM, provide valuable insight into the practical aspects of the implementation discussed in the text, particularly with respect to the subtle issues related to state decomposition and the treatment of fermionic sign conventions where required.
These listings are intended to complement the discussion by offering concrete examples that clarify how the formal logic is realized in code. Given the potential for confusion when dealing with basis state reconstruction and operator ordering in fermionic systems, we believe that these examples can be of interest and practical benefit to users aiming to extend or interface with the library.
p. 28: "as the nonsu2 and superc diagonalizations entail nontrivial subtleties in optimizing the off-diagonal components of X". What are these subtleties?
We acknowledge that our brief comment reads a bit obscure while this point requires a more detailed, yet brief, discussion.
The bath optimization is a crucial step in the DMFT-ED algorithm. Hence, to ensure EDIpack has a large degree of versatility while preserving robustness, we introduced several control parameters for the conjugate-gradient fit. A key one is the definition of the norm, or distance, between matrix functions appearing in Eq. 30.
While the choice of metric plays a limited role for matrix-valued functions with particularly symmetric structures, it can significantly impact the quality of the optimization in more general cases. The subtleties in optimizing the off-diagonal components in either ${\tt nonsu2}$ and ${\tt superc}$ diagonalizations arise from the complex interplay between these components’ physical characteristics and the picked optimization metric. In either case, the coexistence of off-diagonal components having a very different physical origin, e.g. anomalous vs normal terms, and/or having different order of magnitude, represents a serious challenge to the optimization of the bath. This requires careful tuning of the control parameters to ensure that the optimization does not disproportionately favor certain matrix elements over others, thereby maintaining the overall fidelity of the bath representation. In this sense, introducing other norms which allow a fine tuning of the balance among the components, is a development line to improve EDIpack usage.
In the revised manuscript we amended this unclear statement writing: "We plan to expand the available options for the cg_norm parameter in future EDIpack updates, since the nonsu2 and superc diagonalization modes involve subtle challenges in optimizing the off-diagonal components of X. These difficulties are related to the coexistence of matrix elements with different physical origin, so that they can live on different orders of magnitude, hence requiring carefully balanced optimization metrics to ensure an accurate bath representation."
As a general coding comment: would it be possible to remove use of global variables?
We thank the referee for this general but important observation. In principle, it is indeed possible to eliminate the use of global variables by explicitly passing all data structures through subroutine arguments. However, in the context of Fortran, and similarly in other static languages, global variables (typically implemented via module-level variables) do not incur any performance or memory management penalties.
Their use in EDIpack is deliberate and serves to maintain clarity and modularity in the code, particularly for widely shared objects such as configuration parameters, system-wide operators, or state indexing data. Removing these global definitions would require passing a large number of arguments through deep call chains, which we believe would significantly increase code complexity, reduce readability, and introduce greater potential errors. All of this without offering any computational or architectural advantage that fits our current development plan.
However, we recognise the importance of careful design in managing global memory, and we have ensured that such variables are encapsulated within dedicated modules with well defined scopes. Should it prove beneficial for future code maintainability or extension by third-party contributors, we are open to review and refine this aspect of the implementation.
p. 40: I find the discussion of parallelism in Julia wrapper unnecessary. Such information does not necessarily age well. This belongs to a readme file.
We thank the referee for this suggestion. We agree that the detailed discussion of parallelism integration within the Julia wrapper is better suited for specialized documentation such as a README file or the on-line user manual rather than the main manuscript. Accordingly, we have removed the entire paragraph from the revised version to keep the presentation focused and ensure the manuscript’s longevity and clarity. Relevant implementation details regarding MPI usage in Julia will be provided in the package documentation to better serve users requiring this information.
p. 41: The code listing mentions Wband and de, but I don't see where this is coming from.
We thank the referee for carefully reviewing this section and pointing this out. Indeed, the variable ${\tt Wband}$ should not appear in the code listing, as it corresponds to ${\tt D}$, which is correctly defined earlier in the script. Additionally, the variable ${\tt de}$ represents the energy grid spacing, computed as ${\tt de = 2D / Le}$, where Le is the number of energy points discretizing the Bethe lattice density of states. We have corrected the code listing accordingly to eliminate Wband and clarify the definition of de.
p. 42: "provides access to well-tested functions". This is unclear. Functions doing what?
We thank the referee for pointing out the ambiguity in this sentence. Our intention was to clarify that, although the bath optimization problem is conceptually distinct from the core impurity diagonalization task, for which EDIpack was originally developed, nonetheless the package provides access to a well-tested implementation of the conjugate-gradient fitting procedure for the bath optimization. We have revised the sentence accordingly in the manuscript to make this connection and the role of these functions more explicit by writing:
"Given the critical importance of this step, EDIpack provides access to a well-tested implementation of the conjugate gradient method for performing the bath optimization, ensuring stability and reproducibility of the results. This task is conceptually distinct from the diagonalization of the impurity problem, which remains the core focus of the package."
p. 47: generate_kgrid is confusingly complex. There must be a simpler way to accomplish this.
We agree with the referee that the implementation of ${\tt generate_kgrid}$ could be simplified for clarity. The purpose of this function is merely to construct a uniform 2D grid in the Brillouin zone using the reciprocal lattice vectors. In the case of the square lattice discussed in this example, this implementation is indeed rather overcomplicated, given the orthogonality of the reciprocal lattice vectors. Being completely generic, however, this function can also be used for less simple 2d lattices (e.g. triangular). We further note that in the way it is written, relying entirely on numpy functions, it avoids making use of explicit python ${\tt for}$ loops, which cause performance degradation when the requested number of k-points is large.
More generally, we note that this step is not essential to the core functionalities provided by EDIpack, as it pertains only to the sampling of the Brillouin zone. For more involved tight-binding constructions, users may wish to rely on specialized libraries such as PythTB, which are designed for this purpose.
p. 57: Is the footnote necessary? Will it be of interest to the expected readers of this paper?
We appreciate the referee’s suggestion. Upon reconsideration, we agree that the footnote was not essential for the intended readership and have removed it from the revised version of the manuscript.
Parenthesis missing in the equation at the bottom of p. 12.
Bottom of p. 13: "are store" -> "are stored"
p. 41: "an comprehensive" -> "a", "fo manipulating" -> for
p. 46: This example two main goals: missing "has"?
We thank the referee for carefully pointing out these typographical errors. All the mentioned issues have been corrected in the revised version of the manuscript.

Anonymous on 2025-07-10 [id 5627]
Reconsider the title? "upgrading EDIpack" sounds a bit like a minor update to the predecessor that was presented in A. Amaricci, L. Crippa, A. Scazzola, F. Petocchi, G. Mazza, L. de Medici, and M. Capone, "EDIpack: A parallel exact diagonalization package for quantum impurity problems", Computer Physics Communications 273, 108261 (2022) [58]. Maybe "EDIpack 2.0"? Or specify what distinguishes it from the predecessor (and justifies the creation of a new repository on SciPost Physics Codebases).
Anonymous on 2025-07-19 [id 5650]
(in reply to Anonymous Comment on 2025-07-10 [id 5627])We acknowledge this insightful suggestion which led us to reconsider the title of our work.
In the resubmitted manuscript we changed the title to more clearly reflect the novelty and improvements introduced in this
version of EDIpack with respect to the previous work.

---

## Round 1 · Referee Report · Anonymous (Referee 2) · 2025-7-9

Report
This paper presents the current status in an exact diagonalisation impurity solver EDIpack. It provides a detailed exposition of the multiple interfaces as well as a number of tutorial-style examples that provide details that will be appreciated to new users of the package. I believe that providing robust and well-tested computer codes, as well as high quality documentation, are a big service to the community. I wholeheartedly recommend this work for publication.
Some minor comments and suggestions:
-
On p. 8, function ed_set_Hloc is introduced, but without much further information. Maybe the reader could be referred to some later section where its use is illusrated.
-
On p. 9, the discussion about the phonon cutoff sounds potentially a bit misleading. The cutoff is surely problem dependent, and there are (generalized) cases where the shift A_m cannot be considered as a free parameter.
-
I find the discussion at the end of page 11 somewhat unclear, especially the terminology. Is this a discussion of consecutive indexing vs. occupation number representation?
-
The motivation of including the code on p. 12 is unclear. What is the intention here?
-
Parenthesis missing in the equation at the bottom of p. 12.
-
Bottom of p. 13: "are store" -> "are stored"
-
Top of p. 14: What is global share, what are istart, ishift, iend? I suppose this is Fortran-specific.
-
p. 16: GFmatrix is said to be a critical component for high-speed execution. Maybe it could be describe in more detail? In what way is it efficient? What does it mean it is multi-layed? In passing, it would be nice if the capitalization would be uniform, e.g. GFmatrix vs. gfmatrix (I understand that Fortran compiler does not care, but the human reader perhaps does).
-
A trick for computing off-diagonal functions is presented on p. 21. Doesn't this require switching to complex-valued floating points even in cases where the Hamiltonian is purely real?
-
In Eq. (18), is Z the same on both sides of approximation sign?
-
Are the code listings on p. 25 and p. 26 of sufficient interest to readers?
-
p. 28: "as the nonsu2 and superc diagonalizations entail nontrivial subtleties in optimizing the off-diagonal components of X". What are these subtleties?
-
As a general coding comment: would it be possible to remove use of global variables?
-
p. 40: I find the discussion of parallelism in Julia wrapper unnecessary. Such information does not necessarily age well. This belongs to a readme file.
-
p. 41: "an comprehensive" -> "a", "fo manipulating" -> for
-
p. 41: The code listing mentions Wband and de, but I don't see where this is coming from.
-
p. 42: "provides access to well-tested functions". This is unclear. Functions doing what?
-
p. 46: This example two main goals: missing "has"?
-
p. 47: generate_kgrid is confusingly complex. There must be a simpler way to accomplish this.
-
p. 57: Is the footnote necessary? Will it be of interest to the expected readers of this paper?
Recommendation
Publish (surpasses expectations and criteria for this Journal; among top 10%)
We thank the referee for their time and work in reviewing our manuscript. In the following, we address point by point all the issues raised in their reports.
Overall, the paper is well written and highly detailed, allowing new users to install and utilize the library with ease. Furthermore, the comprehensive DMFT tests, demonstrated through a variety of examples and accompanied by corresponding scripts, provide excellent starting points for newcomers. I believe this work makes a valuable contribution to the community and am therefore pleased to recommend its publication in SciPost.
We thank the referee for their positive assessment of our work and for recognizing its value in a more general context. We deeply appreciate their recommendation.
On page 3, a large portion of paragraph 3, which introduces DMFT, lacks citations. For example, the authors mention the success of DMFT in capturing key features of various materials, including Mott insulators, heavy fermion compounds, and unconventional superconductors. It would be helpful to include relevant references here to guide interested readers toward further reading.
We agree with this comment by the referee. Although it is impossible to do justice to the wide literature accumulated over the years, we have made concerted efforts to incorporate key references that reflect the breadth and depth of research in this field. In the revised version, we have included several citations pertinent to DMFT across the discussed subjects, ensuring that foundational and more recent contributions are acknowledged. This approach aims to provide a balanced overview while addressing the referee's insightful comment.
Perhaps a sentence like “Section 6 provides the conclusions” was intended at the end of Section 1 but was inadvertently omitted?
We sincerely appreciate the referee's prompt attention to this matter. We indeed inadvertently overlooked a reference to Section 6 at the end of the introduction when discussing the manuscript's structure. We resolved this issue in the revised text by including the sentence:
"In Sec. 6 we present some concluding remarks and considerations."
Were the spin indices omitted in Eq. (3)? Or do the indices i,j,k,l represent “flavors” that is, combined spin and orbital indices—rather than orbitals alone? The same question applies to the display equation of $H^{int}$ on page 13. This could be confusing to readers, especially since the spin index is also not explicitly shown in the format of the <UMATRIX\_FILE>.restart file described on the same page.
Thanks for this insightful comment. Indeed, both in Eq.(3) and in the display equation of $H^{int}$ on page 13, the indices $i, j, k, l$ represent "flavors", corresponding to the combined spin and orbital indices rather than orbitals alone. We acknowledge that this could potentially be confusing, especially given the absence of explicit spin indices in the format of the <UMATRIX\_FILE>.restart file described on the same page.
To clarify this, we have revised the manuscript to explicitly state that the indices denote combined spin and orbital components. Additionally, we will add a note to the relevant sections to highlight this point, ensuring that the readers can interpret the equations and file formats correctly.
We appreciate the referee's attention to this detail and believe that their suggestion will enhance the clarity of our presentation.
The last display equation on page 10 seems incorrect, since the creation and annihilation operators do not appear to result in any particle being created or annihilated.
We express our gratitude to the referee for bringing to our attention the discrepancy in the equations for the bosonic operators. In the revised manuscript, we have rectified this error by modifying the expressions on the right-hand side of the equations to account for the change in the bosonic number.
In Sec. 3.3, it appears that users must manually select one of the three provided symmetry configurations to perform block-diagonalization. Is there a plan to extend the code to automatically detect all relevant symmetries—especially in cases where orbitals or sites possess additional symmetries that would allow further block-diagonalization of the matrix?
We acknowledge the referee’s observation. While the main code currently necessitates manual selection of one of the provided symmetry configurations, we recognize the value of automating the detection of all relevant symmetries. In fact, the Triqs interface ${\tt edipack2triqs}$ features an efficient algorithm to automatically recognize the symmetries of the problem and select the appropriate value for ${\tt ed_mode}$. In the actual version of the Fortran API this is not implemented yet, but we have plans to incorporate it in a future release. In the current setup, this requires to accurately analyse the model Hamiltonian and the effective bath decomposition, as discussed in section 3.6, at initialization time.
At the same time, we are actively exploring development avenues that aim to integrate more sophisticated conservation laws, which would further reduce the matrix dimensions. We are confident that these enhancements would improve code’s flexibility and efficiency in managing systems characterized by intricate symmetries.
On page 43, it is mentioned that the spectrum obtained from ED is “inherently spiky”. What is the value of the broadening parameter used in these simulations? Including this information in the main text may improve the clarity and reproducibility of the paper.
We appreciate the referee's observation. The width of the Lorentzian broadening of the spectral poles is controlled by the input parameter eps. It is important to note that the value of eps is not fixed universally but rather is chosen based on the specific requirements of each simulation. In the particular examples discussed on page 43 of our manuscript, we have employed a broadening parameter of eps = 0.01, expressed in units of the half-bandwidth D.
In response to the referee's suggestion, we have updated the manuscript explicitly specifying the value of this broadening parameter.
for the display equation of $C(z)$ on page 16, summation index $m$ has lower and upper bounds, while index $n$ does not, although they are dummy variables for the same thing. Same for Eq. (14)
We appreciate the referee's careful observation regarding the summation indices in the display equation of $C(z)$ on page 16 and Eq. (14). To maintain consistency and clarity, we have revised the manuscript to ensure that both indices $m$ and $n$ have clearly defined lower and upper bounds, as they represent dummy variables for the same summation.
In particular, we now explicitly indicate the upper summation bound for $n$ by using the number $N_{\rm state_list}$ of eigenstates contained in the state_list class and describing the lower end of the energy spectrum.
Following the referee's observation, we propagated this correction to all the expressions involving a trace over the Hamiltonian spectrum. We appreciate the referee's attention to this detail.
"$J_x \rightarrow J_X$" and "$J_p \rightarrow J_P$" in the second paragraph of page 13
“store” $\rightarrow$ “stored” in the last line of page 13
"As discussed" $\rightarrow$ "As will be discussed" at the beginning of Sec. 3.5.2
"$\vec{B_\sigma}$" $\rightarrow$ "$\vec{B}_\sigma$" in the sentence "The values are given by all integers $B_\sigma$ corresponding to..." of page 14
"$w(\mathcal{A})_{mn}^\nu$" $\rightarrow$ "$w_{mn}^\nu(\mathcal{A})$" in the second line after Eq.~(14)
"This example two main goals" $\rightarrow$ "This example has two main goals" on page 46.
should be no indentation for the sentence right after the first display equation on page 17
We sincerely appreciate the referee's careful review and constructive feedback. In the revised manuscript, we have addressed all the indicated typos and corrections as suggested. We are grateful for the referee’s meticulous attention to detail, which has significantly contributed to improving the clarity and accuracy of our manuscript.

---

## Round 2 · Author Response

Dear Editors,

We would like to thank you for considering our manuscript
“A flexible and interoperable high-performance Lanczos-based solver for generic quantum impurity problems: upgrading EDIpack” (SciPost submission ID: scipost_202506_00023v1), which we hereby resubmit for publication.

We are grateful to both referees for their insightful and constructive reviews, as well as for recognizing the value of our work. Both referees considered the manuscript worthy of publication and recommended it accordingly. In particular, Referee A described the paper as “well written and highly detailed”, highlighting that it “makes a valuable contribution to the community”. Likewise, Referee B emphasized that “providing robust and well-tested computer codes, as well as high quality documentation, are a big service to the community” and “wholeheartedly recommend[ed] this work for publication”. We deeply appreciate these encouraging evaluations and believe the feedback we received has helped us significantly improve the manuscript.

In our responses, enclosed with this resubmission, we have carefully addressed all comments and suggestions raised by the referees and the anonymous commenter. The manuscript has been revised accordingly. In particular, following the editor’s recommendation, we have updated the title to more clearly reflect the progress and novelty introduced in this version of EDIpack, without requiring reference to earlier works.

We are pleased to resubmit the revised version of our manuscript for your consideration, and we hope it now meets the criteria for publication in SciPost Physics Codebases.

Best regards,
Adriano Amaricci,
On behalf of all authors

---

## Round 2 · List of Changes

• Revised the title to more clearly reflect the novelty and improvements introduced in this version of the library.
  • Added references to DMFT results in the introduction.
  • Clarified the nature of the indices ${i,j,k,l}$ in the general interaction term.
  • Specified upper and lower bounds in Eq.~(14); the same clarification was propagated to all equations involving traces over Hamiltonian states.
  • Corrected the definition of bosonic creation and annihilation operators.
  • Added reference to the broadening parameters in the examples.
  • Introduced distinct notation to differentiate the exact partition function $Z$ from its truncated approximation $\tilde{Z}$.
  • Added details explaining subtle issues related to off-diagonal bath optimization.
  • Clarified the role of the integer parameters istart, iend, ishift used in parallel matrix construction. Removed misleading terminology and references to parallelism in the Julia interface.
  • Clarified the role of phonons' shift term $A_m$.
  • Improved the explanation of the symmetry sector indexing map.
  • Corrected various typographical errors throughout the manuscript.

---

## Editorial Decision

published